# Minimal Areas from Entangled Matrices

Jackson R. Fliss[b], Alexander Frenkel[♯], Sean A. Hartnoll[b] and Ronak M Soni[b,♮]

[b] *Department of Applied Mathematics and Theoretical Physics,*
*University of Cambridge, Cambridge CB3 0WA, UK*
[♯] *Department of Physics, Stanford University, Stanford, CA 94305-4060, USA*
[♮] *Chennai Mathematical Institute, H1, SIPCOT IT Park,*
*Siruseri, Kelambakkam 603103, India*

## Abstract

We define a relational notion of a subsystem in theories of matrix quantum mechanics and show how the corresponding entanglement entropy can be given as a minimisation, exhibiting many similarities to the Ryu-Takayanagi formula. Our construction brings together the physics of entanglement edge modes, noncommutative geometry and quantum internal reference frames, to define a subsystem whose reduced state is (approximately) an incoherent sum of density matrices, corresponding to distinct spatial subregions. We show that in states where geometry emerges from semiclassical matrices, this sum is dominated by the subregion with minimal boundary area. As in the Ryu-Takayanagi formula, it is the computation of the entanglement that determines the subregion. We find that coarse-graining is essential in our microscopic derivation, in order to control the proliferation of highly curved and disconnected non-geometric subregions in the sum.

# 1  Introduction

The Bekenstein-Hawking entropy, the area of a black hole event horizon in Planck units, shows that quantum gravity geometrises information [1–3]. The Ryu-Takayanagi formula for entanglement entropy extends the geometrisation of information away from horizons to more general minimal [4] or extremal [5] surfaces. The need to extremise is related to the diffeomorphism invariance of gravity, see [6–14] for recent explorations of this connection, and survives beyond the classical gravitational limit [15, 16]. This information-theoretic nature of gravity can be deduced from the semiclassical gravitational path integral [17–20] and therefore provides robust constraints on microscopic theories of quantum gravity.

Microscopic accounts of the Bekenstein-Hawking entropy involve a connection between the geometry of spacetime and the combinatorics of state counting, e.g. [21]. A microscopic account of the Ryu-Takayanagi formula must have, in addition to a combinatorial dimension, an explanation for the minimisation over area. In this paper we will show how combinatorics and minimisation arise, simultaneously, in considering entanglement in theories of gauged matrix quantum mechanics. We may recall that such gauged quantum mechanics underpin our best-understood microscopic formulations of gravity [22, 23].

Our first main result, in §2, is a definition of a subsystem in a general bosonic matrix quantum mechanics (MQM) such that computing the entanglement entropy leads to analogues of both a Bekenstein-Hawking-like area law and a Ryu-Takayanagi-like minimisation. Our second main result, in the remainder of the paper, is an explicit calculation of this entanglement in a simple MQM with an especially tractable 'fuzzy sphere' state.

In §3 we will review aspects of the fuzzy sphere matrix geometry [24]. This is a classical configuration of three $N \times N$ Hermitian matrices. Fuzzy spheres arise in a number of contexts, including as 'polarised' ground states of supersymmetric models [25–28]. For our purposes, it is one of the simplest instances of an emergent geometry from matrices. We will consider a simple bosonic matrix quantum mechanics that, in a certain limit, admits a semiclassical quantum state strongly supported on a fuzzy sphere. The following facts are important:

1. The low energy fluctuations about the fuzzy sphere state are described by a scalar field and a $U(1)$ Maxwell field with coupling $g_M$, on an $S^2 \times \mathbb{R}$ spacetime [29–31].

2. The $U(N)$ matrix gauge symmetry acts on the emergent fields as both the local $U(1)$ Maxwell symmetry and also as volume-preserving diffeomorphisms on $S^2$ [32–35, 31].[1] These are spatial diffeomorphisms on fixed time slices.

3. In particular, an $M \times M$ matrix sub-block corresponds to a subregion $\Sigma$ of the unit sphere with volume $|\Sigma| = 4\pi M/N$ [36].

4. The degrees of freedom in a sub-block corresponding to a subregion $\Sigma$ transform under an irreducible representation $\mu_\Sigma$ of $U(M)$. The dimension of this representation is given by the boundary area of the subregion, $\log d_{\mu_\Sigma} \equiv \log \dim \mu_\Sigma \propto |\partial \Sigma|$ [36, 37].

---

[1]Throughout, for generality, we will refer to 'volumes' that have 'areas' as boundaries. In our two dimensional examples, the 'volume' is more properly an area and the 'area' is more properly a length.

Our matrix subsystem, at low energies, corresponds not to specifying a fixed subregion of $S^2$, but specifying only its volume $|\Sigma|$. The entanglement entropy we find is

$$S_{|\Sigma|} = \frac{1}{\sqrt{3}\pi} \frac{\Lambda^{1/2}}{g_M} \min_{\Sigma} \left( \frac{|\partial\Sigma|}{1/\Lambda} \log\left[ \frac{g_M}{\Lambda^{1/2}} \frac{N|\Sigma|}{\Lambda|\partial\Sigma|} \right] \right) + \cdots . \tag{1.1}$$

The details of this formula will be explained throughout the paper. Here we may highlight that (1.1) shares the three basic properties of the Ryu-Takayanagi entanglement entropy:

1. It is proportional to the inverse of a small coupling $g_M$ in the continuum theory.

2. It is given as a minimisation over sub-regions $\Sigma$ satisfying a gauge-invariant condition; here the condition is a fixed volume $|\Sigma|$.

3. The quantity to be minimized is the boundary area $|\partial\Sigma|$ of the sub-region.

The astute reader will notice two other interesting aspects of (1.1). Firstly, the presence of a multiplicative logarithmic correction to the 'area law,' and secondly, the appearance of a smoothing scale $\Lambda$. The logarithmic correction is a fact of life of the representation-theoretic counting problem that we shall land on, whereas the smoothing scale is necessary to tame UV/IR mixing effects in the noncommutative theory [38–40, 36].

We may briefly give a sense of how the combinatorics and minimisation arise. While the mechanism for emergent geometry is specific to the class of simple MQM models we consider, the structure of combinatorics and minimisation should arise for general theories of MQM. Crucial to both of these is the $U(N)$ gauge invariance of the MQM, which at low energies in our models becomes the set of volume-preserving diffeomorphisms. Consider first a fixed subregion $\Sigma$, defined as a particular $M \times M$ sub-block, of every matrix, in a particular gauge [41, 42]. There are then two types of gauge transformations: the $U(M) \times U(N - M)$ subgroup of $U(N)$ that acts separately within $\Sigma$ and its complement $\overline{\Sigma}$, and the remaining gauge transformations that mix $\Sigma$ and $\overline{\Sigma}$. These are illustrated in Fig. 1. In previous

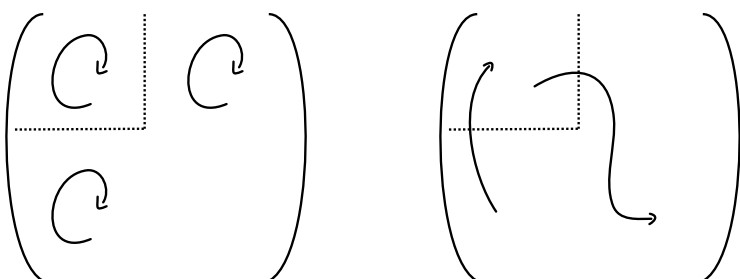

Figure 1: Left: A $U(M)$ transformation acts separately on an $M \times M$ sub-block and its complement. This action will lead to a combinatorial computation of edge mode entanglement. Right: $U(N)/[U(M) \times U(N - M)]$ transformations mix the sub-block with its complement. These actions will lead to a minimisation over 'wiggle modes,' defined below.

studies of entanglement of fixed subregions [43, 44, 36, 37], it was shown that the $U(M)$ transformations, in particular, gave rise to 'edge modes,' with entanglement equal to $\log d_{\mu_\Sigma}$ for an irrep $\mu_\Sigma$ of $U(M)$. The edge modes are similar to those dealt with in the context of

gauge theory [45–49]. Computing the irrep dimension is our combinatorial problem. In the semiclassical limit, this dimension produces an area law entanglement, giving (1.1) without the minimisation. There is no analogue in gauge theories, however, of the remaining 'wiggle mode' gauge transformations in $U(N)/[U(M) \times U(N-M)]$ that mix degrees of freedom between the region and its complement. These modes were gauge-fixed in the works above.

In this paper we will release the wiggle modes from their gauge-fixing. This leads to the picture shown in Fig. 2, as we now outline. Since the wiggle modes are gauge transformations

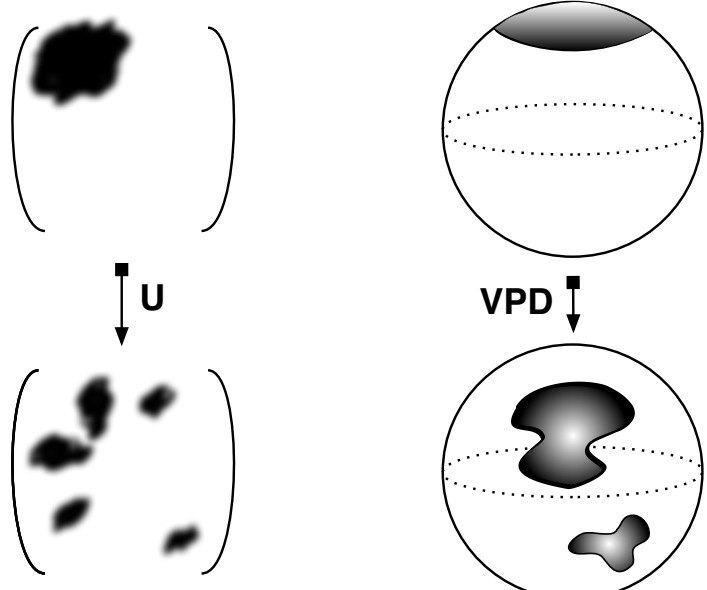

Figure 2: An $M \times M$ matrix sub-block, top left, corresponds to a subregion of the sphere with a volume $|\Sigma|$, top right. A general unitary transformation $U$ spreads an $M \times M$ sub-block around the matrix. The subregion is correspondingly mapped to another subregion with the same volume $|\Sigma|$. For the moment we loosely refer to these transformations as volume-preserving diffeomorphisms (VPDs), even while they may be discontinuous (as in the illustration). Of all subregions with fixed volume, the 'spherical cap' shown top right is the one with minimal boundary area $|\partial\Sigma|$.

that mix degrees of freedom inside and outside of the subregion, two reduced states that differ by one of these modes carry different information and are distinguishable.[2] As a result, we will find that defining the subsystem to be *any* $M \times M$ sub-block gives a density matrix that is a sum over sub-blocks

$$\rho_{\text{in}} \sim \oint_{\mathsf{F}} \mathrm{d}V \left[ \rho_V \otimes \frac{\mathbb{1}_{\mu_V}}{d_{\mu_V}} \right]. \tag{1.2}$$

Here $\mathsf{F}$ is the space of unitary wiggle modes, which we parametrise by $V$. Each $V$ maps the original fiducial sub-block to a new sub-block with $U(M)$ irrep $\mu_V$, while $\rho_V$ is the density

---

[2]Mathematically, the reduced state does not transform unitarily under these gauge transformations and so these transformations change the eigenvalues of the density matrix.

matrix of the $U(M)$ singlet degrees of freedom in the transformed sub-block. This singlet factor will be subleading compared to the maximally mixed edge mode term in (1.2), which is proportional to $\mathbb{1}_{\mu_V}$. The integral over $V$ in (1.2) makes clear that there is no preferred sub-block once the rank $M$ has been fixed.

The direct sum notation in (1.2) indicates that, as we will argue, the density matrices corresponding to the different, distinguishable subregions of the continuum theory are orthogonal to good approximation. With this assumption one obtains from (1.2), evaluating the integral by saddle-point using the fact that the irrep dimensions $d_{\mu_V}$ are large,

$$\operatorname{tr}\rho_{\mathrm{in}}^n \sim \int \mathrm{d}V\, d_{\mu_V}^{1-n} \approx \exp\left\{-(n-1)\min_V \log d_{\mu_V}\right\}. \tag{1.3}$$

The integral over wiggle modes is therefore the origin of our minimisation. The von Neumann entropy may then be obtained as $S_{\mathrm{E}} = (1 - n\partial_n)\log\operatorname{tr}\rho_{\mathrm{in}}^n\big|_{n\to 1}$ which, mapping the sum over wiggle modes to a sum over subregions with fixed volume as in Fig. 2, lands us on the advertised result (1.1). The dominant region with minimal boundary area is the 'spherical cap' shown in Fig. 2 and considered in [36]. There are well-known subtleties with the $n \to 1$ limit here that we will discuss in later sections.

The simple procedure outlined above runs into a major technical problem: most $U(N)$ transformations are highly non-geometric when interpreted as volume-preserving maps. That is to say, they lead to highly curved and disconnected subregions. There are so many of these non-geometric subregions that they dominate the integral over $V$ and overwhelm the saddle point contribution (1.3). Just as we had to coarse-grain the state to deal with UV/IR mixing effects, we will also coarse-grain the integral over the gauge transformations that change the subregion. We do this in a way that eliminates many of the non-geometric wiggle modes from the integral, but preserves more mild volume-preserving transformations. In this way we truly land on (1.1). We believe that the need to introduce these different coarse-grainings is not ad hoc, but will be generic in situations where continuum space is emergent from non-geometric microscopic degrees of freedom.

Our answer (1.1) is *prima facie* pleasing, but it leads to an urgent question: "What are the principles establishing this result?" While an integral over all gauge transformations serves to restore gauge invariance, an integral over coarse-grained gauge transformations does not seem especially gauge-invariant. We will interpret this procedure in the language of relational observables and internal quantum reference frames [50–57]. First, we will point out that the 'fixed $M \times M$ block in a given gauge' subsystem of [41–44,36,37] can be formulated in a gauge-invariant manner, as an algebra of relational observables. Unlike the simplest examples of relational observables, like 'the distance of this text from your eyes,' these observables are not relational to any specific object; rather they are relational to a system of 'rods' that define the gauge. This is also known as an internal quantum reference frame (QRF). We can then interpret our density matrix as that of a QRF-averaged subsystem.

The plan of the paper is as follows. In §2 we define a 'quantum reference frame averaged' entanglement in a general bosonic MQM. Then, in §3 we review a particular theory of MQM that has an especially tractable geometric 'fuzzy sphere' state. In the remainder, we apply the methods of §2 to this state: In §4 we deal with the problem of evaluating the entangle-

ment at a fixed $V$ from the fuzzy sphere state, and in §5 we derive (1.1) by introducing a coarse-graining that restricts the reference frame average to a subgroup $U(N') \subset U(N)$.

**Notation:** Tr is the trace on the $N \times N$ matrix space and tr is the Hilbert space trace.

# 2 Matrix entanglement and $U(N)$ gauge symmetry

In this section we will define the quantum reference frame averaged entanglement of a general theory of bosonic MQM. As we have discussed in the introduction, our notion of matrix entanglement builds on several previous works. These include the matrix partitions introduced in [41–43] and the construction of $U(M)$ matrix edge modes associated to a sub-block in [44, 36, 37]. We review the relevant aspects of previous work in the subsections below. The contribution of the present section is to interpret these partitions in a relational manner, as a subsystem of operators dressed to a particular internal quantum reference frame, and then to allow the reference frame itself to be uncertain.

The formalism developed in this section will be general. As also discussed in the introduction, in simple models of noncommutative geometry the averaging over reference frames leads to an integral over subregions with a fixed volume. We will see later that, in certain semiclassical regimes, the subregion with minimal boundary area dominates this integral.

## 2.1 Hilbert spaces: extended, invariant and physical

Our starting point is the quantum mechanics of $d$ $N \times N$ Hermitian matrices $X^1 \ldots X^d$, with a Lagrangian $L = \text{Tr}\, \mathsf{L}(X^a, \dot{X}^a)$. The matrices can often be associated to directions in an ambient space $\mathbb{R}^d$, which is distinct from the emergent noncommutative space we will be interested in. The theories we will consider have a $U(N)$ gauge symmetry[3] under which

$$X^a \to \mathsf{U} X^a \mathsf{U}^\dagger, \qquad \mathsf{U} \in U(N). \tag{2.1}$$

We define three Hilbert spaces: the extended, the invariant and the physical. The extended Hilbert space is just a copy of $L^2(\mathbb{R})$ for every real degree of freedom,

$$\mathcal{H}_{\text{ext}} \equiv \text{span}\left\{ |X^a\rangle \,\Big|\, (X^a)^\dagger = X^a, a = 1 \ldots d \right\}$$

$$= \bigotimes_{a=1}^{d} \left[ \bigotimes_{1 \le i < j \le N} \text{span}\left\{ |X^a_{ij}\rangle \,\Big|\, X^a_{ij} \in \mathbb{C} \right\} \otimes \bigotimes_{i=1}^{N} \text{span}\left\{ |X^a_{ii}\rangle \,\Big|\, X^a_{ii} \in \mathbb{R} \right\} \right]. \tag{2.2}$$

The cumbersome second line has the advantage of clarifying automatically the inner product in this Hilbert space: it is just a $\delta$-function in each of these factors. The invariant Hilbert space is the quotient of the extended Hilbert space by all gauge transformations (2.1),

$$\mathcal{H}_{\text{inv}} \equiv \mathcal{H}_{\text{ext}} \Big/ \left( |X^a\rangle \sim |\mathsf{U} X^a \mathsf{U}^\dagger\rangle \right). \tag{2.3}$$

---

[3]More properly, the gauge group is the projective unitary group $PU(N)$, which is $U(N)$ quotiented by the overall phase, $\mathsf{U} \sim e^{i\phi}\mathsf{U}$. Our considerations will mostly be at large $N$, where the distinction between $PU(N)$ and $U(N)$ is subleading. In §4 we will need to remember that the correct group is $PU(N)$.

An equivalent definition of the invariant Hilbert space is as follows. Let the matrices $\hat{\Pi}^a$ be, component-wise, the momenta canonically conjugate to $\hat{X}^a$. The generators of the $U(N)$ transformation (2.1) are the matrix of operators $\hat{G} = 2i \sum_a :[\hat{X}^a, \hat{\Pi}^a]:$, where the normal ordering symbol : : means that all $\hat{X}_{ij}$ are to the left of $\hat{\Pi}_{kl}$ in terms of operator ordering. Explicitly,

$$\hat{G}_{ij} = 2i \sum_{a=1}^{d} (\hat{X}_{ik}^a \hat{\Pi}_{kj}^a - \hat{X}_{kj}^a \hat{\Pi}_{ik}^a) . \tag{2.4}$$

Then, $\mathcal{H}_{\text{inv}}$ is the subspace of $\mathcal{H}_{\text{ext}}$ annihilated by $\hat{G}$:

$$\mathcal{H}_{\text{inv}} \equiv \left\{ |\Psi\rangle \in \mathcal{H}_{\text{ext}} \Big| \hat{G}_{ij} |\Psi\rangle = 0 \text{ for } 1 \le i, j \le N \right\} . \tag{2.5}$$

A point of notation: for the rest of §2 (and this section only), we denote Hilbert space operators with hats.

The physical Hilbert space $\mathcal{H}_{\text{phys}}$ of the theory is isomorphic to the invariant Hilbert space. It will be crucial for us, however, that (2.5) is just one of many ways of realising the physical Hilbert space within the extended Hilbert space. This fact is illustrated in Fig. 3 and elaborated on below. The essential point is that gauge-fixed states are an alternative

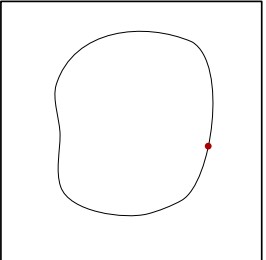 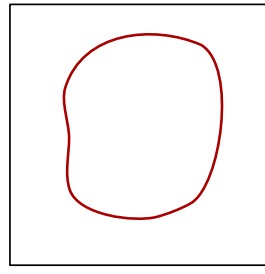 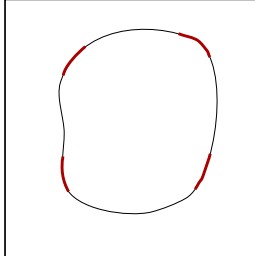

Figure 3: Three isomorphic embeddings of the physical Hilbert space $\mathcal{H}_{\text{phys}} \hookrightarrow \mathcal{H}_{\text{ext}}$. The curve in each box represents the gauge orbit of some classical matrix configuration. From left to right: (1) Wavefunctions are $\delta$-function localized to a particular gauge slice, represented by the highlighted point on the orbit, (2) $\mathcal{H}_{\text{inv}}$, consisting of wavefunctions that are smeared over the entire orbit, and so are annihilated by the gauge generators, (3) wavefunctions are not smeared over the entire gauge orbit, but are smeared over some subset of the orbit. This final possibility will arise below upon coarse-graining the gauge orbit.

way to realise $\mathcal{H}_{\text{phys}}$. We will see how this works in detail in the following §2.2. The first step is to decompose $\mathcal{H}_{\text{ext}}$ into the gauge orbits of a particular gauge-fixed slice. We will use the parametrisation

$$X^d = \mathsf{U}\lambda\mathsf{U}^\dagger, \quad \mathsf{U} \in U(N), \quad \lambda \text{ diagonal and ordered.} \tag{2.6}$$

Here $X^d$ is the last of the $d$ matrices. The remaining $d-1$ matrices are written as

$$X^a = \mathsf{U}X_{\text{gf}}^a\mathsf{U}^\dagger , \tag{2.7}$$

where the $X_{\text{gf}}^a$ are not necessarily diagonalised. Sometimes we will also write $X_{\text{gf}}^d = \lambda$ for convenience. The gauge-slice is parametrised by $\lambda$ and the $X_{\text{gf}}^a$, while the gauge orbits are

described by $\mathsf{U}$. Thus, we can write almost any vector $|X^a\rangle$ uniquely as

$$|X^a\rangle = \sqrt{\Delta(\lambda)}\,|\mathsf{U}\rangle\,|X_{\mathrm{gf}}^{1\ldots d-1},\lambda\rangle\,. \tag{2.8}$$

We have incorporated the Vandermonde measure factor

$$\Delta(\lambda) = \prod_{i<j}(\lambda_j - \lambda_i)^2\,, \tag{2.9}$$

with $\lambda_i$ the diagonal entries of $\lambda$, into the state (2.8). Of course, the specific choice of gauge slice (2.6) is not essential; our discussion generalises to any gauge in which a matrix function $F(\{X^a\})$, transforming in the adjoint of $U(N)$, is taken to be diagonal and ordered.

A subtlety is that the decomposition (2.6) fails to be unique when $X^d$ has coincident eigenvalues. This leads to a residual gauge symmetry that, as we discuss below, renders our relational observables ill-defined on states with support on coincident eigenvalues of $X^d$ [58]. The states of main interest to us, such as the fuzzy sphere state, have vanishing support on these configurations.

The decomposition (2.8) shows how $\mathcal{H}_{\mathrm{ext}}$ contains many equivalent copies of each physical state. Wavefunctions in $\mathcal{H}_{\mathrm{inv}}$ are obtained by integrating (2.8) over $\mathsf{U}$. This demonstrates that the physical data of the state is parametrised by $\{X_{\mathrm{gf}}^{1\ldots d-1},\lambda\}$. As evident from (2.8), this same data can also be accessed by making a choice of gauge-slice, such as setting $\mathsf{U} = \mathbb{1}$ in (2.8). This gives a different embedding of physical states into $\mathcal{H}_{\mathrm{ext}}$, as illustrated in Fig. 3. We emphasise that there is no intrinsic sense in which any of the representatives of a physical state in $\mathcal{H}_{\mathrm{ext}}$ is more physical than the others (as long as one deals correctly with residual gauge symmetries). The best choice depends on the problem one wishes to address.

## 2.2 Subsystem and factorisation map

In gauge theories there is a tension between Gauss's law on physical states and partitioning the degrees of freedom into local subsystems. In our case, different matrix components are related by the $U(N)$ Gauss's law, preventing a naïve partition of the matrix into blocks. It is important to emphasise that the tension between partition and Gauss's law is worse for matrix partitions than for spatial partitions of local gauge theories. In gauge theories, Gauss's law results in a lack of local tensor factors of the physical Hilbert space but the theory still admits local subalgebras of physical operators [47]. These local subalgebras act on tensor factors of the analogue of our extended space $\mathcal{H}_{\mathrm{ext}}$ [45, 46, 48], but there are nevertheless ambiguities with defining entanglement entropy [59, 60]. This last observation was formalised into the statement that the definition of entropy requires a choice of *factorisation map* $J : \mathcal{H}_{\mathrm{phys}} \hookrightarrow \mathcal{H}_{\mathrm{fact}}$, where $\mathcal{H}_{\mathrm{fact}}$ admits a local tensor factorisation [61, 62]. The previous literature can then be interpreted as choosing $\mathcal{H}_{\mathrm{fact}} \cong \mathcal{H}_{\mathrm{ext}}$.

Since $U(N)$ gauge transformations move a given sub-block around the matrix, see Fig. 2, we do not obviously have gauge-invariant subalgebras that are local in the emergent space. We will thus bypass an algebraic description and specify the partition as a factorisation map: an isometric map from $\mathcal{H}_{\mathrm{phys}}$ to a subspace of $\mathcal{H}_{\mathrm{ext}}$. Recall from the discussion around Fig. 3 that $\mathcal{H}_{\mathrm{ext}}$ contains many equivalent copies of each physical state. The factorisation map will

involve choosing a particular representative $|\psi_G\rangle \in \mathcal{H}_{\text{ext}}$ of a physical state $|\psi\rangle \in \mathcal{H}_{\text{phys}}$, and then using the tensor product structure of $\mathcal{H}_{\text{ext}}$ to define a Hilbert space of the subsystem. We will now describe the embedding in more detail. In the following subsections we will interpret our construction as a choice of measure over quantum reference frames.

Firstly, we define a gauge-fixed state in $\mathcal{H}_{\text{ext}}$, using the decomposition (2.8), as

$$|\psi_{\mathbb{1}}\rangle \equiv |\mathbb{1}\rangle \, |\psi_{\text{gf}}\rangle \, , \tag{2.10}$$

where $|\psi_{\text{gf}}\rangle$ is any normalisable superposition of the $|X_{\text{gf}}^{1\cdots d-1}, \lambda\rangle$ states in (2.8). The state $|\psi_{\mathbb{1}}\rangle$ itself is only $\delta$-function normalisable because the unitary $\mathsf{U}$ in (2.8) has been fixed to be the identity. As in textbook quantum mechanics, we could make the state normalisable by constructing a wavepacket over unitaries $\mathsf{U}$ that is strongly peaked on $\mathsf{U} = \mathbb{1}$.[4] The details of this wavepacket will contribute to the entanglement at subleading orders in the semiclassical limit, and we will not keep track of these corrections. We take $|\psi_{\mathbb{1}}\rangle$ to be invariant under all residual gauge symmetries, and similarly for the other representatives we consider below.

The state $|\psi_{\mathbb{1}}\rangle$ in (2.10) is fully gauge-fixed. An invariant state is recovered by integrating $|\psi_{\mathbb{1}}\rangle$ over the gauge orbits of the unitary group. However, we will see later that such an integration introduces too much microscopic information. In particular, in the fuzzy sphere model of interest below, it includes discontinuous volume-preserving transformations in which a region can be broken up into a large number of 'Planck-sized' disconnected regions. For this reason, we will wish to build a state in which the gauge transformations are coarse-grained. Thus we consider an intermediate case of partially gauge-fixed states, in which we integrate $|\psi_{\mathbb{1}}\rangle$ over a subgroup $G \subseteq U(N)$. This is the third case illustrated in Fig. 3 above. That is, we consider

$$\left|\psi_G'\right\rangle = \int_G \mathrm{d}g \, \hat{\pi}(g) \, |\psi_{\mathbb{1}}\rangle = \int_G \mathrm{d}g \, |g\rangle \, |\psi_{\text{gf}}\rangle \, . \tag{2.11}$$

Here, $\hat{\pi}$ is the representation of $U(N)$ on $\mathcal{H}_{\text{ext}}$. For reasons to be made clear shortly, we call $G$ the frame transformation group. The state $|\psi_G'\rangle$ has $U(N)/G$ gauge-fixed. The prime in $\psi_G'$ indicates that we have one more step to perform.

Finally, there is one part of $U(N)$ that we will want to fully integrate over. We will be interested in $M \times M$ sub-blocks of the matrix and we wish to consider states that are fully invariant under the $U(M)$ associated to this sub-block, and the $U(N-M)$ associated to the complementary sub-block. This $U(M) \times U(N-M)$ does not move the sub-block around the matrix, as illustrated in Fig. 1 above. Correspondingly, in the fuzzy sphere model it does not move the subregion around the sphere and does not suffer from the same kind of microscopic sensitivity as the general $U(N)$ transformations discussed above. The $U(M)$ transformations, but not the $U(N-M)$ transformations, will instead induce 'edge modes' due to the fact that in the state $|\psi_{\text{gf}}\rangle$ the degrees of freedom inside the subsystem carry a non-trivial $U(M)$ charge. These edge modes will be the source of our boundary area law.

---

[4]It is also possible to take care of this by modifying the inner product using group-averaging [63] or the Hamiltonian BRST formalism [64, 65]. We take the simplest approach available to us, expecting that different approaches differ only at subleading order.

Thus we set

$$|\psi_G\rangle \equiv \int_{U(M)} \mathrm{d}U \int_{U(N-M)} \mathrm{d}\widetilde{U}\,\hat{\pi}(U\widetilde{U})\,|\psi'_G\rangle = \int \mathrm{d}U\,\mathrm{d}\widetilde{U} \int \mathrm{d}g\,|U\widetilde{U}g\rangle\,|\psi_{\mathrm{gf}}\rangle\,. \qquad (2.12)$$

Here, we are abusing notation by using $U$ to denote both an element of $U(M)$ as well as $U \oplus \mathbb{1}_{N-M} \in U(N)$, and similarly with $\widetilde{U}$. We will continue with this practice henceforth. As we discuss further below, there may be a redundancy between the $U(M) \times U(N-M)$ integral and the $G$ integral in (2.12); this does not cause any difficulties. We will not need keep track of the normalisation of our states very carefully, as the normalisation will be accounted for automatically when we compute the von Neumann entropy.

Equation (2.12) defines a map $\mathcal{H}_{\mathrm{phys}} \to \mathcal{H}_{G,M} \subset \mathcal{H}_{\mathrm{ext}}$, where $\mathcal{H}_{G,M}$ is the Hilbert space spanned by states of the form given above. This is our factorisation map and these will be the states we work with. As we will explain in detail, there are two qualitatively different types of integral in (2.12): the integrals over $U, \widetilde{U} \in U(M) \times U(N-M)$ and the integral over $g \in G$. The latter integral is the key ingredient that is new relative to [36, 37]. We will return to the physical interpretation of these steps in the next subsection.

For the class of states in (2.12), we define the subsystem as a tensor factor of $\mathcal{H}_{\mathrm{ext}}$. Recall from (2.2) that $\mathcal{H}_{\mathrm{ext}}$ admits a tensor factorisation into $N^2$ copies of $L^2(\mathbb{R})$, each one corresponding to an element of the matrix. This allows us to define our subsystem as a matrix sub-block, in the spirit of [41, 42]. To specify the sub-blocks of the matrices we introduce the projector and its complement

$$\Theta_\Sigma \equiv \Theta \equiv \begin{pmatrix} \mathbb{1}_{M \times M} & 0_{M \times (N-M)} \\ 0_{(N-M) \times M} & 0_{(N-M) \times (N-M)} \end{pmatrix}, \qquad \Theta_{\overline{\Sigma}} \equiv \overline{\Theta} \equiv 1 - \Theta_\Sigma\,. \qquad (2.13)$$

We can then break up each matrix into four blocks,

$$X^a_{\Sigma\Sigma} \equiv \Theta X^a \Theta, \qquad X^a_{\Sigma\overline{\Sigma}} \equiv \Theta X^a \overline{\Theta}, \qquad X^a_{\overline{\Sigma}\Sigma} = \left(X^a_{\Sigma\overline{\Sigma}}\right)^\dagger, \qquad X^a_{\overline{\Sigma}\overline{\Sigma}} \equiv \overline{\Theta} X^a \overline{\Theta}. \qquad (2.14)$$

The extended Hilbert space thus decomposes as

$$\mathcal{H}_{\mathrm{ext}} = \mathcal{H}_{\Sigma\Sigma} \bigotimes \left(\mathcal{H}_{\Sigma\overline{\Sigma}} \otimes \mathcal{H}_{\overline{\Sigma}\overline{\Sigma}}\right) \equiv \mathcal{H}_{\mathrm{in}} \bigotimes \mathcal{H}_{\mathrm{out}}. \qquad (2.15)$$

Here $\mathcal{H}_{\mathrm{in,out}}$ are defined as the Hilbert space on the corresponding side of the $\bigotimes$.

When $G = \{\mathbb{1}\}$, and in regimes where $|\psi_{\mathrm{gf}}\rangle$ describes a semiclassical noncommutative geometry, the factorisation (2.15) of the Hilbert space corresponds to a geometric partition of the fuzzy geometry. The projector $\Theta$ in (2.13) is written in the same basis in which $\lambda$ was diagonalised in (2.6). When $\mathsf{U} = \mathbb{1}$ in (2.6), then $\lambda = X^d$ and $\Sigma$ is therefore a geometric subregion with coordinates $x^d \leq \lambda_M$ (the $M$th diagonal entry of the matrix $X^d$) and $\overline{\Sigma}$ is its complement [41, 42]. A transformation by $\hat{\pi}(g)$ as in (2.11) effectively conjugates the projector with a volume-preserving diffeomorphism. Because we are integrating over all $g \in G$ in (2.11), the subsystem is then not a single subregion but an average over subregions (when $G \neq \{\mathbb{1}\}$). This was illustrated in Fig. 2 above.

## 2.3 Algebra of relational observables and quantum reference frames

In this subsection and the following §2.4, we will interpret the construction above in the language of internal quantum reference frames [52–57], which is closely related to the Page-Wootters formalism [50, 51].[5] The factorisation map (2.12) will be motivated using this framework. The reader that is happy with (2.12) as a technical starting point may wish to skim these sections. Technical developments towards our main result continue in §2.5.

The basic point is that what one might call gauge-fixed observables can be promoted to gauge-invariant relational observables: they are relational to a *quantum reference frame.* Our partition, from this point of view, corresponds to fixing some gauge-invariant data (the size of the matrix sub-block or the volume of a subregion) but being uncertain about the QRF.[6] This is why we call $G$ the frame transformation group.

A common approach to obtain gauge-invariant operators is to write down combinations of the basic operators $\hat{X}_{ij}^a, \hat{\Pi}_{ij}^a$ that are invariant under gauge transformations, such as $\mathrm{Tr}\, \hat{X}^a \hat{X}^a$. An alternative approach is to define relational operators, as we now describe.

Consider a gauge, such as the choice above where the matrix $X^d$ is diagonal and ordered. Then, working in the basis (2.8), define a class of (generalised) projectors

$$\hat{P}_{\mathsf{U}} \equiv |\mathsf{U}\rangle \langle \mathsf{U}| \,. \tag{2.16}$$

We may now define the (matrix-valued) reference frame operator

$$\hat{\mathsf{U}} \equiv \int_{U(N)} \mathrm{d}\mathsf{U} \, \mathsf{U} \, \hat{P}_{\mathsf{U}} \,, \tag{2.17}$$

and then consider

$$\hat{X}_{\mathbb{1},ij}^a \equiv \left( \hat{\mathsf{U}}^\dagger \hat{X}^a \hat{\mathsf{U}} \right)_{ij} \,. \tag{2.18}$$

To understand what these operators do we may act on a general basis state (2.8) of $\mathcal{H}_{\mathrm{ext}}$. From the definitions above, and using $\hat{X}^a |\mathsf{U}\rangle |X_{\mathrm{gf}}\rangle = \mathsf{U} X_{\mathrm{gf}}^a \mathsf{U}^\dagger |\mathsf{U}\rangle |X_{\mathrm{gf}}\rangle$, one has

$$\hat{X}_{\mathbb{1},ij}^a |\mathsf{U}\rangle |X_{\mathrm{gf}}\rangle = X_{\mathrm{gf},ij}^a |\mathsf{U}\rangle |X_{\mathrm{gf}}\rangle \,. \tag{2.19}$$

That is to say, the relational operators pick out the values of the matrices on a particular gauge-slice. Gauge invariance is the fact that for $\mathsf{U}' \in U(N)$,

$$\hat{\pi}(\mathsf{U}')\hat{X}_{\mathbb{1},ij}^a = \hat{X}_{\mathbb{1},ij}^a \hat{\pi}(\mathsf{U}') \,. \tag{2.20}$$

This may be verified by acting on basis vectors, similarly to (2.19).

The steps above demonstrate that the matrix elements of a fully gauge-fixed operator are gauge invariant data. We may think of this fact relationally: the gauge-fixed matrix elements are values taken by the operator *given* that $X^d$ is diagonal and ordered. Conditioning on the gauge-fixing provides an internal reference frame or, more pictorially, a series of 'rods'

---

[5]We thank Elliot Gesteau for alerting us to this language and Phillipp Höhn for detailed comments.

[6]We use a slightly different language than the QRF literature. What we call different QRFs below are referred to as different orientations of the same QRF in the above cited works.

given by the diagonalised $X^d$.

The $\hat{X}^a_{\mathbb{1},ij}$ operators, however, are not truly well-defined due to the possibility of coincident eigenvalues and the attendant residual gauge symmetries. If $X^d$ has coincident eigenvalues, then there are non-trivial unitaries $\mathsf{U}_d$ that commute with $\lambda$, so that $\lambda = \mathsf{U}_d \lambda \mathsf{U}_d^\dagger$. Defining $\widetilde{X}^a \equiv \mathsf{U}_d X^a \mathsf{U}_d^\dagger$ for $a = 1 \ldots d-1$, the states $|X^{1\ldots d-1}, \lambda\rangle$ and $|\widetilde{X}^{1\ldots d-1}, \lambda\rangle$ lie on the same gauge orbit; this is a residual gauge symmetry. The value of, say, $\hat{X}^1_{\mathbb{1},ij}$ is therefore ambiguous on this gauge orbit, and the operator is not well-defined. It is possible that there is a more sophisticated gauge that does not leave these residual gauge symmetries; if so, the corresponding relational observables will be well-defined.[7] We ignore this subtlety, assuming that our wavefunctions do not have any support on orbits with coincident eigenvalues. Further discussion of this sort of issue can be found in [58].

Let us now construct relational momentum operators, which will again be well-defined only in the absence of residual gauge symmetry. In language similar to (2.18), they are

$$\hat{\Pi}^a_{\mathbb{1},ij} \equiv \int_{U(N)} d\mathsf{U} \left( \mathsf{U}^\dagger \hat{P}_\mathsf{U} \hat{\Pi}^a \hat{P}_\mathsf{U} \mathsf{U} \right)_{ij} . \tag{2.21}$$

It is important that $\hat{\Pi}^a$ is flanked on both sides by the same projector $\hat{P}_\mathsf{U}$. This ensures that the components of the momentum along the gauge orbits are projected out. Explicitly, on a basis state of $\mathcal{H}_{\text{ext}}$:

$$e^{i\,\mathrm{Tr}\left(A^a \hat{\Pi}^a_{\mathbb{1}}\right)} |\mathsf{U} X^a_{\text{gf}} \mathsf{U}^\dagger\rangle = \hat{P}_\mathsf{U} |\mathsf{U}(X^a_{\text{gf}} + A^a)\mathsf{U}^\dagger\rangle = |\mathsf{U}\rangle \, |X^a_{\text{gf}} + A^a_{\text{gf}}\rangle . \tag{2.22}$$

The last equality defines $A^a_{\text{gf}}$; it is the part of $A^a$ that doesn't change the gauge. In particular, all the off-diagonal terms of $A^d_{\text{gf}}$ are zero. Equivalently, the full momentum operator $\hat{\Pi} = -i\partial_X$ defines a vector field on configuration space, and the relational operator is the projection of the vector field onto the gauge slices.

The ordering of eigenvalues leads to another subtlety for the relational momentum operator. Consider the translation $\lambda_1 \to \lambda_1 + a$, generated by $\exp\left(ia\hat{\Pi}^d_{\mathbb{1},11}\right)$. If $a > \lambda_2 - \lambda_1$ then the first eigenvalue is moved beyond the second eigenvalue, which is not allowed. This shift should instead be thought of as a shift of the first eigenvalue by $\lambda_2 - \lambda_1$ and of the second eigenvalue by $a - \lambda_2$, which preserves the ordering. For our purposes, as shown in [66], one may treat the $\hat{\Pi}$ as standard translation operators, with the ordering imposed by the wavefunction. This is the perspective implicit in our factorisation map.

The gauge-invariant operators (2.18) and (2.21) allow the construction of a gauge-invariant algebra of observables associated to the sub-block

$$\widetilde{\mathcal{A}}_{\mathbb{1}} = \left\{ \hat{X}^a_{\mathbb{1},ij}, \hat{\Pi}^a_{\mathbb{1},ij} \middle| i,j = 1 \ldots M \right\}'' , \tag{2.23}$$

---

[7]The main difficulty is that an enlarged gauge orbit at certain points in configuration space implies a reduced number of independent relational operators. The number of such operators therefore has to vary along the gauge slice. This issue doesn't arise when the gauge-invariant data is parametrised via traces. We should also note that there is no difficulty when all gauge orbits are enlarged. In particular, there is always a $U(1)^N$ worth of residual gauge symmetry, because if $\Phi$ is a diagonal matrix of phases, $\mathsf{U}\lambda\mathsf{U}^\dagger = \mathsf{U}\Phi\lambda\Phi^\dagger\mathsf{U}^\dagger$. But since this $U(1)^N$ exists for every configuration, it can be dealt with simply by restricting $\mathsf{U}$ in $|\mathsf{U}\rangle$ to lie in $U(N)/U(1)^N$.

where the $''$ involves taking all sums and products. The algebra is not entirely well-defined due to the issues of residual gauge invariance described above. The tilde denotes that we have one more step left to go. We refer to this (or rather, the algebra $\mathcal{A}_{\mathbb{1}}$ to be introduced shortly) as the algebra 'in the QRF $X^d$.'

To the algebra $\widetilde{\mathcal{A}}_{\mathbb{1}}$ we can associate a factorisation map, the embedding $\mathcal{H}_{\mathrm{phys}} \hookrightarrow \mathcal{H}_{\mathrm{in}} \otimes \mathcal{H}_{\mathrm{out}}$ given by $J_{\mathbb{1}} : |\psi\rangle \mapsto |\psi_{\mathbb{1}}\rangle$ of (2.10). Recall that the tensor factors $\mathcal{H}_{\mathrm{in/out}}$ were defined in (2.15), they are respectively the top-left block and the remaining matrix elements. The operators in the algebra $\widetilde{\mathcal{A}}_{\mathbb{1}}$ are conjugated by $J_{\mathbb{1}}$ under this map. The image of $\widetilde{\mathcal{A}}_{\mathbb{1}}$ acts on $\mathcal{H}_{\mathrm{in}}$ while the image of its commutant $\widetilde{\mathcal{A}}_{\mathbb{1}}'$ acts on $\mathcal{H}_{\mathrm{out}}$. The conjugation is an important technicality and arises because $|\psi_{\mathbb{1}}\rangle$ is a gauge-fixed state in $\mathcal{H}_{\mathrm{ext}}$.

We can now define the algebra $\mathcal{A}_{\mathbb{1}}$, without the tilde. Since we are interested in partitioning the emergent space into two parts, specifying the whole QRF is too fine-grained: we don't need to fix all the rods, we just need to know which region is covered by the first $M$ of them. This means that we want to construct operators that are 'less relational' than those in $\widetilde{\mathcal{A}}_{\mathbb{1}}$. The rods inside the region are moved around by $U(M)$, while those outside are moved around by $U(N - M)$. These are transformations we do not wish to keep track of as they do not affect the partition; we want our QRFs to be incomplete [56]. Therefore, we average the projector $\hat{P}_{\mathsf{U}}$ over $U(M) \times U(N - M)$. The averaged projectors are labelled by $V \in U(N)/[U(M) \times U(N - M)]$:

$$\hat{P}_V \equiv \int \mathrm{d}U \, \mathrm{d}\widetilde{U} \, \hat{P}_{VU\widetilde{U}} \, . \tag{2.24}$$

And the averaged reference frame operator is now

$$\hat{V} \equiv \int \mathrm{d}V \, V \, \hat{P}_V = \int \mathrm{d}\mathsf{U} \, V \, \hat{P}_{\mathsf{U}}. \tag{2.25}$$

To define this operator, we must define a representative of $U(N)/[U(M) \times U(N - M)]$ in $U(N)$: we may choose it to be the space obtained by the action of the exponential map on the subspace $\mathfrak{u}(N)/[\mathfrak{u}(M) \oplus \mathfrak{u}(N - M)]$ of the Lie algebra $\mathfrak{u}(N)$.

In complete analogy to (2.18) and (2.21) above we may introduce the relational operators

$$\hat{X}^a_{M,ij} \equiv \left( \hat{V}^\dagger \hat{X}^a \hat{V} \right)_{ij} \, , \qquad \hat{\Pi}^a_{M,ij} \equiv \int_{U(N)} \mathrm{d}V \left( V^\dagger \hat{P}_V \hat{\Pi}^a \hat{P}_V V \right)_{ij} \, . \tag{2.26}$$

The action of $\hat{X}^a_{M,ij}$ on vectors in the extended Hilbert space is then, cf. (2.19) above,

$$\hat{X}^a_{M,ij} |\mathsf{U}\rangle |X_{\mathrm{gf}}\rangle = (U\widetilde{U} X^a_{\mathrm{gf}} \widetilde{U}^\dagger U^\dagger)_{ij} |\mathsf{U}\rangle |X_{\mathrm{gf}}\rangle \, . \tag{2.27}$$

The operators are 'less relational,' since they relate the matrix element not to a specific gauge (or system of rods) but to a $U(M) \times U(N - M)$ equivalence class of gauges.

The operators in (2.27) remember the location in $U(M) \times U(N - M)$. This means that, unlike for $\widetilde{\mathcal{A}}_{\mathbb{1}}$ in (2.23), we must trace the matrices over the $M \times M$ sub-block to obtain a

gauge-invariant algebra [56]. Thus we define

$$\mathcal{A}_{\mathbb{1}} = \left\{ \operatorname{Tr} F \left( \Theta \hat{X}_M^a \Theta, \Theta \hat{\Pi}_M^b \Theta \right) \right\}, \tag{2.28}$$

for functions $F$ such that $F(U\hat{X}^a U^\dagger, U\hat{\Pi}^b U^\dagger) = U F(\hat{X}^a, \hat{\Pi}^b) U^\dagger$, such as polynomials. This is the same as the sub-block algebra written down in [42], at leading order.

The $\mathcal{A}_{\mathbb{1}}$ algebra commutes with the generators of $U(M)$ gauge transformations — this is the algebraic counterpart of the central observation in [36] that the reduced density matrix carries $U(M)$ edge modes. This fact also means that it is natural to represent $\mathcal{A}_{\mathbb{1}}$ on a Hilbert space annihilated by the $U(M)$ generators. Thus, the factorisation map we associate to this algebra is (2.12) with $G = \{\mathbb{1}\}$:

$$J_M : |\psi\rangle \mapsto \int dU \, d\widetilde{U} \, |U\widetilde{U}\rangle \, |\psi_{\mathrm{gf}}\rangle \ . \tag{2.29}$$

The crucial point is that $U(M)$ acts separately within $\mathcal{H}_{\mathrm{in}}$ and $\mathcal{H}_{\mathrm{out}}$ and does not mix degrees of freedom within these two spaces. Then, analogously to before, this embedding leads to a representation of the image under $J_M$ of $\mathcal{A}_{\mathbb{1}}$ on $\mathcal{H}_{\mathrm{in}}$ and of the image of $\mathcal{A}_{\mathbb{1}}'$ on $\mathcal{H}_{\mathrm{out}}$. However, there will be additional entanglement because the action of $U(M)$ in the two factors is correlated. This is precisely the edge mode entanglement computed in [36, 37].

## 2.4   Coarse-grained frame averaging

There was nothing preferred about the particular set of gauge-fixed states (2.10). We could have instead considered, for example, the states

$$|\psi_{\mathsf{U}}\rangle \equiv |\mathsf{U}\rangle \, |\psi_{\mathrm{gf}}\rangle \ , \tag{2.30}$$

for some fixed $\mathsf{U} \in U(N)$. This amounts to a different gauge-slice where $X^d = \mathsf{U}\lambda\mathsf{U}^\dagger$ is no longer diagonal. The construction of the previous subsection can be repeated, leading to a distinct algebra $\mathcal{A}_{\mathsf{U}}$ of an $M \times M$ sub-block of operators in the QRF $\mathsf{U}^\dagger X^d \mathsf{U}$. Of course, changing both the QRF and the sub-block by a $\mathsf{U}$ transformation will cancel each other, leading to the same subsystem. So, we change the QRF but *not* the sub-block to define $\mathcal{A}_{\mathsf{U}}$.

The fact that every QRF leads to a respectable algebra and subsystem — democratically so, from the perspective of the symmetries of the microscopic theory — means that choosing which frames to consider depends on the physical question one wishes to address. We will now make the argument, already outlined in the introduction and previously in this section, that it is natural to consider a coarse-grained average over QRFs.

In the introduction we recalled that, in semiclassical states supported on noncommutative geometries, a matrix sub-block corresponds to a subregion in the geometry. Reference frame transformations, i.e. unitary gauge transformations of the matrices, correspond to volume-preserving diffeomorphisms that move the subregion around. Our problem is thus seen to be similar to the difficulties encountered in defining gauge-invariant subregions in theories of gravity. It is instructive, then, to make an analogy to subregion-subregion duality in the AdS/CFT correspondence, see e.g. [67] and references therein. The gauge-invariant input

that is fixed in that case is a boundary subregion $B$. The corresponding bulk subregion $b$ is then determined dynamically as the region bounded by the minimal extremal surface $X$ homologous to $B$, such that $\partial X = \partial B$.

In our setup, where there is no AdS/CFT-like asymptotic boundary, the natural gauge-invariant data associated to a subregion is its volume. This is determined by the size $M$ of the matrix sub-block — we recall the proof of this statement in (3.7) below.[8] The analogous procedure to AdS/CFT is then to fix the volume and let the theory determine the actual subregion. To this end it is natural to consider all subregions with a given fixed volume. As we have explained above, this amounts to considering a fixed size $M$ sub-block in all QRFs. Thus we are led to think about integrating over all QRFs. This will lead to an interesting sum over physically inequivalent subsystems, as the reduced state transforms non-linearly under changing QRFs [57]. Before we can set up the computation, however, there is one more issue to deal with: integrating over all QRFs introduces too much uncertainty.

We will show below that, in semiclassical models, the $U(N)$ transformations become volume preserving 'diffeomorphisms' — in quotes because the corresponding maps are typically not continuous, let alone differentiable. We illustrated this fact in Fig. 2 in the introduction. In the relational language, we can say that most transformations in $U(N)$ map a continuous system of rods into a highly discontinuous one. Suppose, for example, that the QRF $X^d$ is associated to a coordinate that increases along a certain direction of the emergent space. The QRF $\mathsf{U}^\dagger X^d \mathsf{U}$ is then instead organized in terms of the direction of increasing $\mathsf{U}^\dagger X^d \mathsf{U}$. This latter 'direction' will generically not have a simple geometric interpretation, as illustrated in Fig. 4. The proliferation of non-geometric QRFs will have an undesirable technical

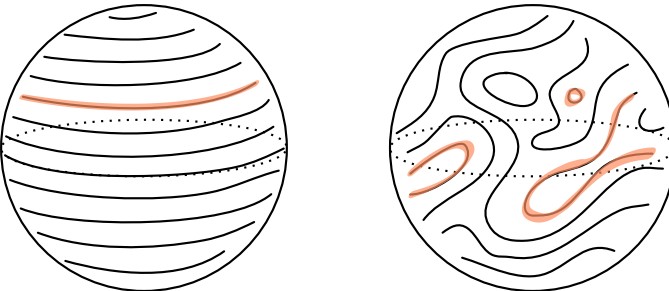

Figure 4: Left, a geometric QRF on the fuzzy sphere. The matrix $X^3$ has been diagonalised and its eigenvectors are associated to level sets that separate regions of unit volume. The highlighted level set is the boundary of the subregion. Right, a less geometric QRF on the fuzzy sphere. The eigenvectors of $\mathsf{U}^\dagger X^3 \mathsf{U}$ are associated to a more complicated set of level sets. The boundary of the subregion has now been mapped to a more curved and disconnected level set. A generic, fully non-geometric QRF cannot be usefully drawn but can be imagined as an even more curvy version of the right hand illustration.

consequence below: the many discontinuous QRFs dominate the average entanglement and, in particular, overwhelm the contribution of the minimal entropy surface. That is, they obscure the connection between entanglement and geometry.

In addition to the technical consequences of a sum over all QRFs, outlined above, one

---

[8]At subleading semiclassical order, fixing the volume and fixing $M$ will give rise to different subsystems.

might expect that extracting sensible low-energy, semiclassical quantities should involve a coarse-graining of the microscopic QRF data. For instance, sufficiently short distance wiggles of the QRF should not change the effective state that a semiclassical observer (measuring with the help of the system of rods whose positions are also fluctuating) sees. This may suggest that the mathematical formalism of non-ideal QRFs, see e.g. [52, 68, 56], could be used to develop a consistent framework for a coarse-grained integral over QRFs. In this work we pursue a more low-tech solution, by considering a simple model for coarse-graining. Specifically, we restrict the frame transformations to a subgroup $G \subseteq U(N)$. This restriction clearly reduces the number of QRFs that will appear in the average. Within the extended Hilbert space formalism developed in §2.2, $G$ is the group introduced in (2.11) above.

One may ask whether there is an algebraic description of the average over frames. It is natural to consider a direct sum over the algebras for each of the possible subsystems

$$\mathcal{A}_{\text{tot}} = \left\{ |g\rangle\langle g| \otimes a_g \,\Big|\, a_g \in \mathcal{A}_g \right\}. \tag{2.31}$$

The $|g\rangle\langle g|$ factor is necessary to label the different elements. In §2.7 below we will argue that (2.31) can't be quite right because it excludes replica symmetry-breaking effects in the entanglement. More physically, the observables associated to distinct — but in general overlapping — subregions should not be fully independent. It would be incredibly interesting to define the frame-averaged partition algebraically. Doing so could ultimately put our extended Hilbert space computations below on a firmer footing or perhaps refine the prescription. We leave this to future work, however.

## 2.5 Entanglement from gauge charge

In this subsection, we review how the matrix edge modes arise. Recall that we are interested in algebras invariant under $U(M) \times U(N-M)$, because these are the operators that depend only on the partition and not the entire QRF. This was implemented in our factorisation map by the explicit integral over this subgroup in (2.12).[9] It is important to emphasise here that the two integrals in the factorisation map (2.12) have distinct raisons d'être: the $U(M)$ integral ensures that the subregion is not over-specified by a choice of interior coordinates while the $G$ integral implements a coarse-grained averaging over subregions.

In this subsection we describe the effects of the $U(M)$ integral in (2.12). The $U(M) \times U(N-M)$ subgroup of the gauge symmetry acts on the different blocks (2.14) of the matrices as, with $U \in U(M)$ and $\widetilde{U} \in U(N-M)$,

$$X^a_{\Sigma\Sigma} \to U X^a_{\Sigma\Sigma} U^\dagger, \qquad X^a_{\overline{\Sigma}\overline{\Sigma}} \to \widetilde{U} X^a_{\overline{\Sigma}\overline{\Sigma}} \widetilde{U}^\dagger, \qquad X^a_{\Sigma\overline{\Sigma}} \to U X^a_{\Sigma\overline{\Sigma}} \widetilde{U}^\dagger. \tag{2.32}$$

The transformation of each block in (2.32) involves only itself. This action therefore doesn't mix the three Hilbert spaces in (2.15). The $U(N-M)$ subgroup acts entirely on $\mathcal{H}_{\text{out}}$, and hence has no consequence for the reduced density matrix on $\mathcal{H}_{\text{in}}$. The $U(M)$ subgroup, however, acts on both $\mathcal{H}_{\Sigma\Sigma}$ and $\mathcal{H}_{\Sigma\overline{\Sigma}}$. This was illustrated in Fig. 1. Thus, when we average

---

[9]Mathematically, this is equivalent to making the reduced state invariant under the subgroup of the gauge symmetry that leaves the subsystem unchanged via a depolarisation channel, as in [57].

over $U(M)$, we are adding back in entanglement that was destroyed by the gauge-fixing [47], while taking care not to disturb the physical region. This is the edge mode entanglement computed in [44, 36], that we will now recover.

The $U(M)$ generators in (2.4) can be written as a sum

$$\hat{G}_{\Sigma\Sigma} = 2i \sum_{a=1}^{d} : [\hat{X}^a_{\Sigma\Sigma}, \hat{\Pi}^a_{\Sigma\Sigma}] : +2i \sum_{a=1}^{d} : [\hat{X}^a_{\Sigma\overline{\Sigma}}, \hat{\Pi}^a_{\Sigma\overline{\Sigma}}] : \equiv \hat{Q}_\Sigma + \hat{Q}_{\overline{\Sigma}}, \tag{2.33}$$

where the first term acts on $\mathcal{H}_{\Sigma\Sigma}$ and the second acts on $\mathcal{H}_{\Sigma\overline{\Sigma}}$. Furthermore, these two terms commute as quantum operators because they act on orthogonal parts of the extended Hilbert space. Given that $\hat{Q}_\Sigma$ and $\hat{G}_{\Sigma\Sigma}$ manifestly generate $U(M)$ actions, so does $\hat{Q}_{\overline{\Sigma}}$. That is, each of $\hat{Q}_\Sigma$ and $\hat{Q}_{\overline{\Sigma}}$ generate a $U(M)$ group that we may call $U(M)_\Sigma$ and $U(M)_{\overline{\Sigma}}$, respectively. The gauge symmetry is the 'diagonal' part of this $U(M)_\Sigma \times U(M)_{\overline{\Sigma}}$.

Any physical state in $\mathcal{H}_{\mathrm{phys}}$ can be decomposed into branches labelled by the irreps $\mu$ of $U(M)_\Sigma$. The integral over $U(M)$ in the factorisation map (2.12) ensures that the corresponding state $|\psi\rangle \in \mathcal{H}_{\mathrm{ext}}$ is invariant under the diagonal $U(M)$ generated by $\hat{G}_{\Sigma\Sigma}$, so that $[\hat{Q}_\Sigma + \hat{Q}_{\overline{\Sigma}}] |\psi\rangle = 0$. This condition implies that each branch also carries the same irrep $\mu$ of $U(M)_{\overline{\Sigma}}$. Because $\hat{Q}_\Sigma$ and $\hat{Q}_{\overline{\Sigma}}$ act on separate degrees of freedom, the state is forced to take the form [46, 59]

$$|\psi\rangle = \sum_{\mu \in \mathrm{irr}\, U(M)} \sqrt{p_\mu} |\psi_\mu\rangle \otimes \frac{1}{\sqrt{d_\mu}} \sum_{m=1}^{d_\mu} |m\rangle_{\Sigma\,\mu} \otimes |\overline{m}\rangle_{\overline{\Sigma}\,\mu} , \tag{2.34}$$

where $p_\mu$ and $d_\mu$ are the probability of finding the irrep $\mu$ and the dimension of the irrep, respectively. The states $|m\rangle_{\Sigma\,\mu}, |\overline{m}\rangle_{\overline{\Sigma}\,\mu}$ are bases for the $\mu$ irrep of $U(M)_\Sigma, U(M)_{\overline{\Sigma}}$, such that the superposition above is a singlet under $U(M)_\Sigma \times U(M)_{\overline{\Sigma}}$.[10] The states $|\psi_\mu\rangle$ are singlets under $U(M)_\Sigma$ and $U(M)_{\overline{\Sigma}}$, one for each irrep. These singlet states live in the full Hilbert space and in particular also contain entanglement between $\mathcal{H}_{\mathrm{in}}$ and $\mathcal{H}_{\mathrm{out}}$. The decomposition (2.34) implies that the reduced density matrix is

$$\hat{\rho}_{\mathrm{in}} = \bigoplus_\mu p_\mu \left[ \hat{\rho}_{\mu,\mathrm{in}} \otimes \frac{\mathbb{1}_\mu}{d_\mu} \right] , \tag{2.35}$$

with $\mathbb{1}_\mu = \sum_m |m\rangle_{\Sigma\,\mu} \langle m|_{\Sigma\,\mu}$ and $\rho_{\mu,\mathrm{in}}$ the reduced density matrix obtained from $|\psi_\mu\rangle$. The entanglement entropy is therefore

$$S_{\mathrm{E}} = \sum_\mu p_\mu \left\{ \log \frac{d_\mu}{p_\mu} + S_{\mathrm{vN}}\left(\rho_{\mu,\mathrm{in}}\right) \right\} . \tag{2.36}$$

The above treatment of the $U(M)$ part of the matrix gauge symmetry — in particular equations (2.34) and (2.36) — is the same as in lattice gauge theories, e.g. [46]. Equations (2.34) and (2.36) have also appeared in interpretations of holography as a quantum error-

---

[10]As an example, the (unnormalised) singlet in the spin-$j$ irrep of $SU(2)$ is $\sum_{m=-j}^{j} (-1)^j |j, m\rangle |j, -m\rangle$. So, in this case, $|\overline{j, m}\rangle = (-1)^j |j, -m\rangle$.

correcting code [69], where the $\sum_\mu p_\mu \log d_\mu$ term was identified as the source of the Ryu-Takayanagi area term. In matrix quantum mechanics too, an area law entanglement was obtained from the same term in [44, 36, 37]. The essential points of that computation are as follows. Firstly, semiclassical matrix states are strongly localised on the gauge orbit of a particular classical configuration, which we write as $X^a_{\mathrm{gf}} = X^a_{\mathrm{cl}}$. Secondly, in such states the reduced density matrix transforms under a single dominant irrep $\mu_\star$. The dimension $d_{\mu_\star}$ of this irrep can be computed from $X^a_{\mathrm{cl}}$, and the entanglement entropy is

$$S_{\mathrm{E}} \approx \log d_{\mu_\star} \,. \tag{2.37}$$

The area law is then obtained by evaluating $d_{\mu_\star}$, as we will recall below. The final term in (2.36) is the von Neumann entropy of the fluctuating fields in the emergent geometry, and is subleading in the semiclassical matrix limit.

The result (2.37) gives the gauge-theoretic entanglement entropy of the state (2.34). This is also the entanglement entropy of the state $|\psi_G\rangle$ in (2.12) when $G = \{\mathbb{1}_N\}$. We must now incorporate the integral over a non-trivial $G$.

## 2.6 Towards an emergent Ryu-Takayanagi formula

In this subsection we will average over QRFs by taking $G$ in (2.12) to be a non-trivial group. We will need to separate out the $G$ transformations that are also in $U(M) \times U(N-M)$, and those that are not. To that end, we can define two new subgroups and a quotient set

$$H \equiv G \cap U(M), \qquad H' \equiv G \cap U(N-M), \qquad \mathsf{F} \equiv H \times H' \backslash G. \tag{2.38}$$

It is easy to check that $H$ and $H'$ are subgroups. $\mathsf{F}$ is known as a generalised flag manifold; physically, it indexes the QRFs that lead to distinct subregions. Denote by $U, \widetilde{U}$ and $V$ elements of $U(M), U(N-M)$ and $\mathsf{F}$, respectively.

As we have noted, we will be focused on cases where the gauge-fixed wavefunction is sharply peaked on a certain classical configuration $X^a_{\mathrm{cl}}$. To leading semiclassical order we have $\langle X^a_{\mathrm{gf}} | \psi_{\mathrm{gf}} \rangle \propto \delta(X^a_{\mathrm{gf}} - X^a_{\mathrm{cl}})$, so that the extended Hilbert space state (2.12) can be approximated by the gauge orbit

$$|\psi_G\rangle \approx \int \mathrm{d}U \, \mathrm{d}\widetilde{U} \int_G \mathrm{d}g \, |U\widetilde{U}g X^a_{\mathrm{cl}} g^\dagger \widetilde{U}^\dagger U^\dagger\rangle \equiv \int_{\mathsf{F}} \mathrm{d}V \, |\psi_V\rangle \,. \tag{2.39}$$

We are not keeping track of the normalisation of the state at this point. In the final step we have defined the partially gauge-fixed state $|\psi_V\rangle$. The state $|\psi_V\rangle$ is integrated over, and hence invariant under, $U(M)$, $H$ and $H'$, but not $\mathsf{F}$. The most important of these invariances will be $U(M)$, as this connects to the earlier discussion in §2.5. Clearly there is a redundancy in integrating over both $U(M)$ and $H$ and over both $U(N-M)$ and $H'$; this can be subsumed into the normalisation of the state.

The analysis in §2.5 above is valid for each $|\psi_V\rangle$ separately. In particular, as $|\psi_V\rangle$ is $U(M)$ invariant it can be decomposed into representations of $U(M)_\Sigma$ as in (2.34). We will argue in §5 that $|\psi_V\rangle$ is dominated by a single irrep $\mu_V$ in two instances that together cover

all cases of interest: $(i)$ generic $V$ and $(ii)$ $V$ corresponding to weakly curved and connected subregions. Thus the decomposition (2.34) becomes

$$|\psi_V\rangle \approx |\psi_{\mu_V}\rangle \otimes \frac{1}{\sqrt{d_{\mu_V}}} \sum_{m=1}^{d_{\mu_V}} |m\rangle_{\Sigma\,\mu_V} \,|\overline{m}\rangle_{\overline{\Sigma}\,\mu_V}\,. \tag{2.40}$$

Approximating the state by a single irrep in this way is stronger than making the approximation in the entanglement entropy, as we did in (2.37). We will discuss the effects of the (here neglected) variance over irreps at the end of this subsection.

The reduced density matrix of $|\psi_G\rangle$ in $\mathcal{H}_{\text{ext}}$ is, from (2.39),

$$\hat{\rho}_{\text{in}} = \int dV\, dV'\, \text{tr}_{\text{out}} \left[ |\psi_V\rangle\langle\psi_{V'}| \right]. \tag{2.41}$$

In Appendix A we argue that we can approximate the integrand by something proportional to $\delta(V - V')$ at leading semiclassical order. This is a non-trivial approximation with important physics content, as we discuss in §2.7 below. It means that the density matrix has no mixing between different subregions and hence two QRFs that lead to distinct subregions can be perfectly distinguished from the outside alone. With this approximation and using (2.40) in (2.41), we obtain the density matrix

$$\hat{\rho}_{\text{in}} \approx \mathcal{N} \oint dV \left[ \hat{\rho}_{\mu_V,\text{in}} \otimes \frac{\mathbb{1}_{\mu_V}}{d_{\mu_V}} \right]. \tag{2.42}$$

The (unimportant) normalisation $\mathcal{N}$ will be determined shortly. From (2.42):

$$\text{tr}\,\hat{\rho}_{\text{in}}^n \approx \delta(0)\,\mathcal{N}^n \int dV\, d_{\mu_V}^{1-n}\, \text{tr}\,\hat{\rho}_{\mu_V,\text{in}}^n\,. \tag{2.43}$$

The prefactor of $\delta(0)$ here is not raised to the $n$th power and therefore cannot be absorbed into the normalisation of $\hat{\rho}_{\text{in}}$; it leads to a $\log\delta(0)$ contribution to the entropy. In the following paragraph we elaborate on the meaning of this term.

The density matrix in (2.42) can also be written as $\hat{\rho}_{\text{in}} \approx \mathcal{N} \int dV \left[ \hat{\rho}_{V,\text{in}} \otimes |V\rangle\langle V| \right]$, where $|V\rangle$ is just the position vector for the matrix $V$. The orthogonality of the terms with different $V$ is now explicit from $\langle V|V'\rangle = \delta(V - V')$. In computing $\text{tr}\,\hat{\rho}_{\text{in}}^n$ one then obtains the factor of $\delta(0) = \langle V|V\rangle$ in (2.43). This fact also determines the normalisation $\mathcal{N} = 1/\delta(0)$. These delta functions will be smeared out by subleading corrections in which the $|V\rangle$ in the average over frames are replaced by states with an uncertainty $\Delta V$ in $V$. The prefactor $\delta(0) \sim \frac{1}{\Delta V}$. We will estimate $\Delta V$ in §5. The appearance of such prefactors is generic in continuum limits of probability distributions, as discussed recently in e.g. [70].[11]

---

[11]Consider a set of probabilities $p_i$ with $i \in \mathbb{Z}$. Let $x = \Delta x\, i$ and take the continuum limit $\Delta x \to 0$. In this limit $p_i \to \Delta x\, p(x)$ and the entropy

$$s = -\sum_i p_i \log p_i \to -\int dx\, p(x) \log[\Delta x\, p(x)] = \widetilde{s} - \log\Delta x\,. \tag{2.44}$$

Here $\widetilde{s}$ is the naïve continuum entropy, that can be negative, and $-\log\Delta x$ corresponds to the offset $\log\delta(0)$ in the text. Similarly in correspondence to the text: $\sum_i p_i^n \to \frac{1}{\Delta x}\mathcal{N}^n \int dx\, p(x)^n$, with $\mathcal{N} = \Delta x$.

We will evaluate the integral in (2.43) by saddle point. This may not be a valid approximation in all theories — the remainder of this section is concerned with cases where it is. In §5 we explicitly demonstrate the dominance of the saddle point for the semiclassical fuzzy sphere state of the 'bosonic mini-BMN' model, studied in [36], with a suitable choice of frame transformation group $G$. Furthermore, in our computation the contribution to the entanglement from $\hat{\rho}_{\mu_V,\text{in}}$ in (2.43) will be subleading compared to the contribution from the dimension $d_{\mu_V}$. Assuming this fact, as well as the saddle-point approximation, the leading-order Rényi entropy at $n > 1$ is

$$S_n = \frac{1}{1-n} \log \operatorname{tr} \hat{\rho}_{\text{in}}^n \approx \min_V \log d_{\mu_V}. \tag{2.45}$$

It is important to appreciate that the minimum rather than the maximum appears in (2.45) because the exponent in $e^{(1-n)\log d_{\mu_V}}$ is negative for $n > 1$ and $d_{\mu_V}$ large. It has been assumed in (2.45) that factor of $\delta(0)\mathcal{N}^n$ in (2.43) is also subleading to the value of the saddle point exponent. This will be addressed explicitly in §5.

If we were to take (2.45) at face value it would imply that all the Rényi entropies are equal. The von Neumann entropy would then be obtained trivially from (2.45) in the limit $n \to 1$ as

$$S_{\text{E}} = \min_V \log d_{\mu_V}. \tag{2.46}$$

This is our minimisation formula for the entropy. In fact, we claim that while the von Neumann entropy (2.46) is correct, the formula (2.45) only holds for

$$\frac{1}{\log d_{\mu_V}} \ll n - 1 \ll 1. \tag{2.47}$$

We will discuss the lower and upper bounds on $n - 1$ in turn. These considerations closely mirror issues arising in deriving the Ryu-Takayanagi formula.

The von Neumann entropy is obtained from (2.43) in the limit $n \to 1$. A well-known subtlety for this kind of limit is that the saddle-point approximation breaks down when $n - 1$ becomes sufficiently small that it cancels out the large factors that justify the saddle point approximation in the first place, see e.g. [71, 72]. In our case a valid saddle point approximation requires $1 \lesssim (n-1)\log d_{\mu_V}$. This is the lower bound on $n - 1$ in (2.47). We give explicit formulae for $\log d_{\mu_V}$ in the fuzzy sphere model below. This lower bound may seem to obstruct the $n \to 1$ von Neumann limit. However, the rule for calculating the von Neumann entropy is to calculate the Rényi entropy at *integer* $n > 1$ and then analytically continue the result. This means that the semiclassical limit (large $d_{\mu_V}$) must be taken before the $n \to 1$ limit. In practice, we cannot compute the Rényi entropies at integer $n > 1$ because of the upper bound in (2.47). Nonetheless, within the same order of limits we can evaluate $S_n$ in the window allowed by (2.47). This is sufficient to subsequently take the $n \to 1$ limit and obtain (2.46).

The upper bound in (2.47) is due to variance in the irrep that has been neglected in (2.40), as we now show. Analogous effects are well-known in gravitational entanglement [73]. We

can re-instate the variance by taking a few steps back to (2.35) and writing

$$\hat{\rho}_{\text{in}} = \bigoplus_{\mu} \frac{p_\mu}{d_\mu} \mathbb{1}_\mu \, . \tag{2.48}$$

Here we have dropped the subleading factors of $\hat{\rho}_{\mu,\text{in}}$. Since we are interested in irreps with parametrically large dimension, we can approximate $\sum_\mu$ as $\int d\mathsf{D} \, d\Omega$, where $\mathsf{D}$ is the dimension and $\Omega$ is an abstract variable including all of the other details about the irrep. All Jacobian factors have been absorbed into the measure $d\Omega$, and so have the normalisation factors that ensure $\text{tr} \, \hat{\rho}_{\text{in}} = 1$. Thus we find

$$\text{tr} \, \hat{\rho}_{\text{in}}^n = \sum_\mu p_\mu^n d_\mu^{1-n} = \int d\mathsf{D} \, d\Omega \, p(\mathsf{D}, \Omega)^n \mathsf{D}^{1-n}. \tag{2.49}$$

We now evaluate the $\mathsf{D}$ integral by saddle point. As previously below (2.45) we assume that the saddle point exponent dominates the result so that

$$\text{tr} \, \hat{\rho}_{\text{in}}^n \approx d_\star(n)^{1-n} \, , \tag{2.50}$$

where

$$\partial_{\mathsf{D}} \log \left( \int d\Omega \, p(\mathsf{D}, \Omega)^n \mathsf{D}^{1-n} \right) \Big|_{\mathsf{D}=d_\star(n)} = 0 \, . \tag{2.51}$$

We wish to quantify the backreaction of the non-trivial function of $\mathsf{D}$ in (2.51), $\int d\Omega \, p(\mathsf{D}, \Omega)^n$, on the saddle point $d_\star(n)$ relative to our previous answer, in which $d_\star(n) = d_\star(1) = d_{\mu_\star}$. For $n$ close to 1, we find

$$- \log \text{tr} \, \hat{\rho}_{\text{in}}^n \approx (n-1) \log d_\star(1) + (n-1)^2 \frac{d'_\star(1)}{d_\star(1)} + \mathcal{O}\left((n-1)^3\right). \tag{2.52}$$

We have therefore recovered (2.45), up to a correction that can be neglected when $n-1$ is small enough. More precisely, the corrections are small when $(n-1)|\partial(\log \log d_\star)/\partial n| \ll 1$. It is plausible that the factor appearing here is an $\mathcal{O}(1)$ quantity, and this is what we have written in the upper bound in (2.47), but it is hard to estimate without a more detailed analysis that we do not carry out in this work. These steps are a Hamiltonian version of a famous argument in [17] that shows why the 'derivation' of the RT formula in [74] gives the right answer for the von Neumann entropy despite the correct objections in [75].

The logarithm of the dimension of the representation in (2.46) is the same as was found previously in gauge-fixed computations [36, 37]. We will recall below why, in semiclassical models of noncommutative geometry, this logarithm is proportional to the area bounding the subregion in the emergent space. The crucial new ingredient in (2.46) is the minimisation over gauge transformations. As we also recall below, in semiclassical regimes these gauge transformations are volume-preserving transformations that move the subregion around. Therefore, from the geometric perspective, we may re-write (2.46) as

$$S_{\text{E}} \sim \min_{|\Sigma|=M} |\partial \Sigma|, \tag{2.53}$$

where we minimise the boundary area $|\partial\Sigma|$ over all subregions of the target space with a fixed volume $|\Sigma|$. Equation (2.53) is our emergent 'Ryu-Takayanagi' formula, that we have obtained from a microscopic theory. We may repeat that the volume is gauge-invariant data in our theory, and is therefore analogous to a boundary subregion in AdS/CFT setups. The proportionality factors in (2.53) depend on the theory; our result for the fuzzy sphere model was advertised in (1.1) above.

## 2.7 Further comparison to the Ryu-Takayanagi formula

We have emphasised that that the minimal area formula (2.53) is similar to the Ryu-Takayanagi (RT) formula. Derivations of the RT formula within the AdS/CFT correspondence [17,15,18,19] begin with defining boundary conditions for a semiclassical gravitational path integral that correspond to a replica trick in the boundary CFT. That path integral is then performed by saddle point. We have instead taken an entirely Hamiltonian route. However, we obtained the traces of our density matrix in terms of an integral over quantum reference frames that we have also evaluated using a saddle point approximation. We will now compare various aspects of our story with the holographic RT formula in more detail.

We have already argued in §2.4 that there is a strong analogy with holographic entanglement in how we define the subregion: in both cases one specifies a gauge-invariant property of the subregion and then allows the actual subregion to fluctuate.

A second analogy to holographic entanglement concerns the role of replica symmetry. An important simplification for us was the absence of cross-terms among the different $\mu_V$ irreps in the reduced density matrix (2.42). Recall that each $V$ corresponds to a distinct location of the entanglement cut. The diagonal density matrix (2.42) then says that the choice of bulk subregion is a *classical* choice — there is no quantum mixing between different choices. This has a parallel in the derivation of the RT formula, which contains an assumption of replica symmetry. It was emphasised in [76,77,71] that violations of this assumption involve cross-terms between different choices of bulk subregion. Thus, our approximation (2.42) is closely related to replica symmetry, as we further elaborate in Appendix A. As we noted above, connecting to physics beyond replica symmetry motivates extending the naïve algebra (2.31) to contain 'frame off-diagonal' elements $|g\rangle \langle g'|$.

We have emphasised above that once the replica index $n$ becomes significantly greater than one, then it is essential to include the backreaction of the variance over irreps in the integral. In a gravitational setting an entirely analogous backreaction is crucial to obtain smooth bulk saddles, and hence the correct Rényi entropies, for general $n$. It is also known that, in simple cases, the correct answer for the gravitational von Neumann entropy can be found by neglecting the backreaction and allowing the bulk saddle to look like a bulk replica trick [74,17]. This is precisely the situation we described around (2.52) above. It would be interesting to improve our procedure to capture the backreaction explicitly — this would be the analogue of a 'smooth bulk' in the calculation of $\operatorname{tr} \hat{\rho}_{\text{in}}^n$.

# 3 The fuzzy sphere state

## 3.1 Classical configuration

In the remainder of the paper we will apply the construction of §2 to a particular quantum state of matrices. This state will be strongly supported on a classical configuration of three $N \times N$ Hermitian matrices $\{X^a\}_{a=1}^3$, known as a fuzzy sphere [24]. The current §3 will collect various known facts about this state that we will then use in the following sections.

On the classical fuzzy sphere, the $\{X^a\}_{a=1}^3$ matrices furnish an $N$ dimensional irreducible representation of the $\mathfrak{su}(2)$ algebra. Explicitly, the classical configuration is

$$X_{\text{cl}}^a = \nu J^a \,, \tag{3.1}$$

where the number $\nu \gg 1$ controls the radius of the fuzzy sphere and the matrices on the RHS are

$$J^3 = \sum_{i=0}^{N-1} \left( \frac{N-1}{2} - i \right) |i)(i| \,, \quad J^+ = \sum_{i=1}^{N-1} \sqrt{i(N-i)} \, |i-1)(i| \,, \tag{3.2}$$

as well as $J^- = (J^+)^\dagger$, and where $J^\pm = J^1 \pm iJ^2$. We are using $|i)(j|$ to denote a basis of matrix entries, to distinguish them from states of a quantum Hilbert space. The matrices in (3.2) have been written in the familiar basis for $\mathfrak{su}(2)$, where $J^3$ is diagonal with entries running in integer steps from $(N-1)/2$ to $-(N-1)/2$ and the raising and lowering matrices $J^\pm$ have non-zero entries just above and below the diagonal.

An example of a potential that has the configuration (3.1) as a minimum is

$$V(X) = \frac{1}{4} \text{Tr} \left[ \left( \nu \epsilon^{abc} X^c + i[X^a, X^b] \right)^2 \right] \,. \tag{3.3}$$

This potential arises in the bosonic sector of the so-called mini-BMN model [28, 78].

## 3.2 $U(N)$ and volume-preserving diffeomorphisms

The emergence of a smooth two-sphere as the large $N$ limit of the fuzzy sphere may be seen from the properties of the Moyal map, as we now briefly review.[12] The matrices $X_{\text{cl}}^a$ may be mapped to coordinates $x^a$ endowed with a noncommutative $\star$ product such that

$$X_{\text{cl}}^a X_{\text{cl}}^b \to x^a \star x^b \,. \tag{3.4}$$

In the large $N$ limit the $\star$ product is found to tend towards the ordinary commutative multiplication of functions. This is completely analogous to the familiar emergence of ordinary multiplication from operator multiplication in the $\hbar \to 0$ limit of quantum mechanics. In the large $N$ limit, then, the $SU(2)$ Casimir constraint $\sum_a X_{\text{cl}}^a X_{\text{cl}}^a = \frac{1}{4}\nu^2(N^2 - 1)$ becomes the constraint $\sum_a x^a x^a = \frac{1}{4}\nu^2 N^2$. This shows that the $x^a$ coordinates are constrained to a two-sphere of radius $\frac{1}{2}\nu N$. Any function $f(X_{\text{cl}})$ of the matrices therefore maps to a function

---

[12]For a recent explicit construction of the Moyal map for the fuzzy sphere see Appendix C of [31]. The construction uses matrix spherical harmonics as a convenient basis of matrices. The large $N$ correspondence of these matrices to ordinary spherical harmonics has been shown very explicitly in Appendix A of [36].

$f(x)$ on the sphere. At finite $N$ the function $f(x)$ is defined via its Taylor series expansion, with the $x^a$ multiplied using the $\star$ product and with the same ordering as the $X_{\mathrm{cl}}$ matrices. We will mostly be interested in the commutative large $N$ limit.

The matrices transform naturally under $U(N)$, with $\widetilde{X}_{\mathrm{cl}}^a = U X_{\mathrm{cl}}^a U^\dagger$. Under the Moyal map this transformation corresponds, in the large $N$ limit, to a diffeomorphism on the two-sphere: $\widetilde{x}^a = y^a(x)$. These are in fact volume-preserving diffeomorphisms (see footnote 1 for use of 'area' $vs$ 'volume'), as has been known for a long time [32, 33]. The preservation of volume may be understood from a further property of the Moyal map, which is that the trace of a matrix is mapped to the integral of the corresponding function over a sphere:

$$\frac{1}{N}\operatorname{Tr}[f(X_{\mathrm{cl}})] \to \frac{1}{4\pi}\int_{S^2}\mathrm{d}\Omega_x f(x)\,. \tag{3.5}$$

In the analogy with familiar quantum mechanics, (3.5) is the statement that a trace over the Hilbert space becomes an integral over phase space in the $\hbar \to 0$ limit. It then follows from $\operatorname{Tr}[f(X_{\mathrm{cl}})] = \operatorname{Tr}\big[f(U X_{\mathrm{cl}} U^\dagger)\big]$ that

$$\int_{S^2}\mathrm{d}\Omega_x f(x) = \int_{S^2}\mathrm{d}\Omega_x f(y(x)) = \int_{S^2}\mathrm{d}\Omega_y |\partial_y x| f(y)\,, \tag{3.6}$$

for all functions $f$. In the second step we changed variables in the integral to $y$. The equality of the first and final expressions requires the Jacobian determinant to be the identity everywhere, and hence $y(x)$ must be a volume-preserving diffeomorphism.

The projection matrices introduced in §2.5 obey $\Theta_\Sigma^2 = \Theta_\Sigma$, so their natural images under the Moyal map, at large $N$, are characteristic functions $\chi_\Sigma(x)$ that return 1 inside $\Sigma$ and 0 outside of $\Sigma$. This connection shows us that the size $M$ of the projection matrix is simply the volume $|\Sigma|$ of the subregion $\Sigma$. From (3.5):

$$M = \operatorname{Tr}\Theta_\Sigma = \frac{N}{4\pi}\int_{S^2}\mathrm{d}\Omega_x \chi_\Sigma(x) = \frac{N}{4\pi}|\Sigma|\,. \tag{3.7}$$

This is in units where the emergent sphere has unit radius.

### 3.3 Quantum mechanical state

In this subsection we review a quantum mechanical state that describes Gaussian fluctuations about the classical fuzzy sphere. A Hamiltonian that will generate such fluctuations is

$$H = \tfrac{1}{2}\operatorname{Tr}\left(\Pi^a \Pi^a\right) + V(X)\,, \tag{3.8}$$

with the potential $V$ as in (3.3). The large $\nu$ limit is the semiclassical limit in which the potential term dominates and the quantum fluctuations are small.

The fluctuations of the matrices can be expanded in normal modes $Y_{lm}^a$ as

$$X^a = X_{\mathrm{cl}}^a + \sum_{lm}\delta x_{lm}Y_{lm}^a\,, \tag{3.9}$$

where $\delta x_{lm}$ are coefficients and $\{l,m\}$ are the usual angular momentum quantum numbers,

but with $l \leq N$. This truncation is a symptom of the underlying 'fuzziness'. The normal modes $Y_{lm}^a$ have corresponding normal frequencies $\omega_{lm}$, such that the semiclassical fuzzy sphere state is

$$\psi_{\text{fs}}(\delta x) = \exp\left\{-\frac{\nu}{2}\sum_{lm}|\omega_{lm}|\delta x_{lm}^2\right\}. \tag{3.10}$$

At large $\nu$ this state is indeed strongly supported on the classical fuzzy sphere.[13] The connected two-point functions in this Gaussian state scale as $\langle XX\rangle_c \sim \nu^{-1}$ and $\langle \Pi\Pi\rangle_c \sim \nu$. The explicit form of the $Y_{lm}^a$ modes and $\omega_{lm}$ frequencies for the fuzzy sphere state of the Hamiltonian (3.8) has recently been discussed in [31, 36], building on [29, 30]. A summary is given in Appendix B. The fuzzy sphere state (3.10) is written in the gauge (2.6). Infinitesimal $U(N)$ transformations of $X_{\text{cl}}^a$, that would rotate the state out of this gauge, correspond to zero modes with $\omega_{lm} = 0$ and therefore do not appear in (3.10) [31, 36].

It has been found in previous works that to obtain a geometric edge mode entanglement it is necessary to coarse-grain the quantum state [36]. This is logically distinct from the coarse-graining over QRFs that we have discussed previously. One may think of it as the statement that a local partition of the degrees of freedom should only be defined within the low energy theory, which has an emergent approximate locality. Because the state (3.10) is a product of angular momentum modes, one can easily coarse-grain the state by introducing a cutoff $\Lambda \ll N$ on the sum over angular momentum. The coarse-grained modes are not entangled with the modes that are projected out in this way. Indeed, $\Lambda$ functions as a cutoff on the momentum of the physical modes of the emergent noncommutative field theory on the fuzzy sphere [38]. The truncation leaves us with a Gaussian wavefunction that depends on $2\Lambda^2 + \Lambda$ independent modes $\delta x_{lm}$, instead of $2N^2 + N$ independent modes. Explicitly, going forward all expectation values are calculated with respect to the truncated wavefunction:

$$\psi_{\text{fs}}(\delta x) = \exp\left\{-\frac{\nu}{2}\sum_{l=0}^{\Lambda}\sum_{m=-l}^{l}|\omega_{lm}|\delta x_{lm}^2\right\}. \tag{3.11}$$

## 3.4 Geometric partition and boundary area

We have seen in (3.7) that the volume $|\Sigma|$ of a subregion is given by the rank $M$ of the corresponding matrix sub-block. In this subsection we will recall how the boundary area $|\partial\Sigma|$ of the 'spherical cap' subregion is encoded in the classical fuzzy sphere matrices [36]. This subregion, illustrated in the top right of Fig. 2, has the minimal boundary area among subregions of fixed volume. The projection matrix for this subregion is simply (2.13), in the gauge (3.1) where $X_{\text{cl}}^3$ is diagonal.

Using the projector (2.13), the 'off-diagonal' blocks of the classical matrices (3.1) may be obtained as in (2.14):

$$X_{\text{cl},\Sigma\overline{\Sigma}}^3 = 0, \qquad X_{\text{cl},\Sigma\overline{\Sigma}}^+ = \nu\sqrt{M(N-M)}\,|M-1)(M|, \qquad X_{\text{cl},\Sigma\overline{\Sigma}}^- = 0. \tag{3.12}$$

We see that there a single non-zero off-diagonal entry in (3.12). This coefficient has a

---

[13]For the bosonic Hamiltonian (3.8) the fuzzy sphere state is metastable [31].

geometric interpretation

$$\sqrt{M(N-M)} = \frac{N}{4\pi}|\partial\Sigma|\,, \tag{3.13}$$

where $|\partial\Sigma|$ is the boundary area of the cap. This can be seen as follows. The relation (3.7) between the volume of the cap and $M$ can be written as $M = N\sin^2\frac{\theta}{2}$, where $\theta$ is the polar angle of the cap. It then follows that $\sqrt{M(N-M)} = \frac{N}{2}\sin\theta$. The boundary area of the cap is $|\partial\Sigma| = 2\pi\sin\theta$.

The fact that the off-diagonal blocks of the classical matrices are low-rank, with entries that specify the boundary area of the corresponding subregion, holds for general weakly curved, connected subregions [37]. However, as we have repeatedly emphasised, this is not the case for generic highly curved and disconnected subregions. In particular, for a generic unitary $V$ the off-diagonal blocks $(VX^a_{\text{cl}}V^\dagger)_{\Sigma\overline{\Sigma}}$ will typically no longer be low rank. An important objective in the remainder of this paper will be to understand the contribution of such non-geometric subsystems to the average over reference frames, and to show how this contribution can be controlled by coarse-graining.

The off-diagonal entry (3.13) determines the irrep dimension $d_{\mu_\star}$ in (2.37) as [36]

$$\log d_{\mu_\star} = 2\ell\log\frac{eM}{\ell}\,, \qquad \ell = \frac{\sqrt{\nu^3\Lambda^3 N}}{4\sqrt{3}\pi}|\partial\Sigma|\,. \tag{3.14}$$

Here $\Lambda$ is the cutoff introduced in (3.11). We may motivate the result (3.14) as follows. As discussed around (2.33) above, the $U(M)_\Sigma$ charge carried by the sub-block must equal the $U(M)_{\overline{\Sigma}}$ charge carried by the off-diagonal block. Furthermore, the off-diagonal generators $\hat{Q}_{\overline{\Sigma}}$ in (2.33) are proportional to the off-diagonal matrix operators $\hat{X}_{\Sigma\overline{\Sigma}}$. In the semiclassical limit, the generators are thus dominated by the classically non-zero entry in (3.12).

We will revisit and extend the computation leading to the area law entanglement (3.14) below, to incorporate the average over frames.

# 4 Entanglement in a fixed reference frame

To quantify the relative contribution of geometric and non-geometric subregions to the average over frames in (2.43) we will need to compute the dimension of the irrep associated to the different types of subregion. In this section we will explain how the irrep for a given fixed frame (i.e. a fixed subregion) can be extracted from the reduced density matrix by evaluating the expectation values of the $U(M)_\Sigma$ Casimirs. We will assume in this section that in each frame a single irrep dominates the reduced density matrix. This will be proven in the following §5, in which we shall also explicitly evaluate the Casimirs.

## 4.1 The dominant representation

The entanglement entropy for a fixed $V \in \mathsf{F}$ is given by the dimension of the $U(M)_\Sigma$ irrep that dominates $|\psi_V\rangle$, as in (2.37). In the following §4.2 we will explain how, for the cases of interest, the dimension can be calculated from the row lengths of the Young diagram (YD) for the representation. In the present subsection, following [37], we will obtain these row

lengths from the expectation values of the Casimirs $\operatorname{Tr} Q_\Sigma^p$, for $p = 2 \ldots M$, in the state $|\psi_V\rangle$. Because the group is in fact $PU(M)$ the irreps are indexed by two-sided YDs, as discussed in [79]. We will, however, work with conventional Young diagrams and note any extra subtleties where relevant. Note also that $\operatorname{Tr} Q_\Sigma = 0$ in $PU(M)$.

By definition, a Casimir is proportional to the identity on each irrep. It is therefore enough to obtain the expectation value of $\operatorname{Tr} Q_\Sigma^p$ in any single state in the irrep. A convenient choice is the highest weight state $|\mu; \mathrm{hw}\rangle$, which is annihilated by all of the generators $Q_{\Sigma ij}$ with $i < j$ [80, 81]. In the Young tableau description this state is represented by a tableau where the label in each box is equal to its row number,[14]

$$
|\mu; \mathrm{hw}\rangle = \left|
\begin{array}{|c|c|c|c|c|c|}
\hline
1 & 1 & 1 & \ldots & \ldots & 1 \\
\hline
\end{array}
\right.
\left.
\begin{array}{|c|c|c|c|}
\hline
2 & 2 & \ldots & 2 \\
\hline
\end{array}
\right.
\left.
\begin{array}{}
3 \ldots 3 \\
\vdots
\end{array}
\right\rangle , \tag{4.1}
$$

and where the irrep $\mu$ is encoded in the shape of the Young diagram. We find

$$
\left\langle \operatorname{Tr} Q_\Sigma^p \right\rangle_{\rho_{\mathrm{in}}} = \sum_\mu p_\mu \left\langle \mu; \mathrm{hw} \middle| \operatorname{Tr} Q_\Sigma^p \middle| \mu; \mathrm{hw} \right\rangle \approx \left\langle \mu_V; \mathrm{hw} \middle| \operatorname{Tr} Q_\Sigma^p \middle| \mu_V; \mathrm{hw} \right\rangle . \tag{4.2}
$$

In the first step we used the fact that the Casimir is proportional to the identity within each irrep to replace all the states within each irrep with the highest weight state. In the second step we have assumed that there is a single dominant irrep $\mu_V$. This will be proven in §5.

To evaluate $\operatorname{Tr} Q_\Sigma^p |\mathrm{hw}\rangle$ in (4.2), we may commute all of the entries of $Q_\Sigma$ that annihilate the highest weight state to the right. The $U(M)$ quantum commutators for the components $Q_{\Sigma ij}$ have the schematic form $[Q_\Sigma, Q_\Sigma] \sim Q_\Sigma$. In particular, each commutator reduces the power of $Q_\Sigma$. The only term with $p$ factors of $Q_\Sigma$ that survives will consist of entirely diagonal entries. Iterating this process one obtains

$$
\langle \mathrm{hw} | \operatorname{Tr}\left[Q_\Sigma^p\right] |\mathrm{hw}\rangle = \left\langle \mathrm{hw} \middle| \sum_i Q_{\Sigma ii}^p + \mathcal{O}\left(Q_\Sigma^{p-1}\right) + \mathcal{O}\left(Q_\Sigma^{p-2}\right) + \cdots \middle| \mathrm{hw} \right\rangle . \tag{4.3}
$$

Here we have suppressed $p$-dependent combinatorial prefactors. We may now estimate the terms with lower powers of $p$ and show that they become subleading at large $\nu$.

To leading order in the semiclassical approximation we can replace $X^a \to X_V^a \equiv V X_{\mathrm{cl}}^a V^\dagger$ in the generators (2.33), to obtain

$$
Q_\Sigma \approx Q_{\Sigma \,\mathrm{cl}} \equiv -2i \sum_{lm} \delta\pi_{lm} \left( X_{V\,\Sigma\overline{\Sigma}}^a Y_{lm\,\overline{\Sigma}\Sigma}^a - Y_{lm\,\Sigma\overline{\Sigma}}^a X_{V\,\overline{\Sigma}\Sigma}^a \right) . \tag{4.4}
$$

Because the momenta $\Pi^a$ vanish in the classical ground state, these must be considered to first non-trivial order in the quantum fluctuations (3.9). Here the $\delta\pi_{lm}$ are the momenta conjugate to the $\delta x_{lm}$ in (3.9) and the $Y_{lm}^a$ are the normal modes in (3.9). We have used the Gauss law $Q_\Sigma = -Q_{\overline{\Sigma}}$, see e.g. (2.33), to express the generator in terms of the off-diagonal

---

[14]Two-sided Young diagrams have both boxes and anti-boxes. The latter have to be numbered in descending order, with $M$ in the first row, $M - 1$ in the second row, etc.

blocks of the matrices.

In the semiclassical generators (4.4) we have typical matrix elements $X_V \sim \nu N$, from the classical matrices (3.1), while $\langle \delta \pi^2 \rangle \sim \nu$, from the Gaussian wavefunction (3.10). For even values of $p = 2s$ there are an even number of $\delta \pi$ terms in the trace and hence $\langle \operatorname{Tr} Q_\Sigma^{2s} \rangle \sim \nu^{3s} N^{2s}$. For odd values of $p = 2s+1$ one of the $\delta \pi$ terms must be contracted in the Gaussian with a fluctuation $\delta x$ about the classical matrices. Using the fact that $\langle \delta \pi \delta x \rangle \sim 1$ we obtain in this case that $\langle \operatorname{Tr} Q_\Sigma^{2s+1} \rangle \sim \nu^{3s} N^{2s}$. Thus the trace of an odd power scales in the same way as the trace of the preceding even power. This effectively means that the odd traces are down in powers of $\nu$ and $N$ relative to a naïve power counting and may therefore be thought of as vanishing to leading order. Using these scalings in (4.3) we obtain

$$
\langle \text{hw} | \operatorname{Tr} Q_\Sigma^p | \text{hw} \rangle = \left\langle \text{hw} \left| \sum_i Q_{\Sigma\,ii}^p \left[ 1 + \mathcal{O}\left( \frac{1}{\nu^3 N^2} \right) \right] \right| \text{hw} \right\rangle
$$

$$
\approx \left\langle \text{hw} \left| \sum_i Q_{\Sigma\,ii}^p \right| \text{hw} \right\rangle = \sum_r \ell_r^p \,. \tag{4.5}
$$

In going to the second line we have taken the large $\nu$ semiclassical limit. In the final step we have used the fact that the action of $Q_{\Sigma\,ii}$ on any state is to count the number of boxes that have label $i$. On the highest weight state (4.1), this is equal to the number of boxes $\ell_i$ in the $i$th row. We have let the label $i \to r$ in the final step for later convenience.

As (4.5) holds for any even power $p$, the spectrum of this classicalised charge must be

$$
\operatorname{spec} Q_{\Sigma\,\text{cl}} = \{ \ell_1, -\ell_1, \dots \ell_{M/2}, -\ell_{M/2} \} \,. \tag{4.6}
$$

That is to say, the semiclassical eigenvalues of $Q_\Sigma$ are precisely the row lengths of the Young diagram labelling the irrep. The negative eigenvalues correspond to the reflected side of the YD that we are suppressing (and we see here that, to leading order, the full two-sided YD is reflection-symmetric). From (4.6) we see that in order to extract the irrep of a subregion, we need to understand the eigenvalues of $Q_\Sigma$.

Note that the entries of $Q_{\Sigma\,\text{cl}}$ in (4.4) quantum commute amongst themselves and hence no longer obey the $U(M)$ commutation relations. This is consistent with the commutator terms becoming subleading in the semiclassical limit (4.5). The smallness of quantum fluctuations is a first indication that the $\operatorname{Tr} Q_\Sigma^p$ observables are strongly peaked on their expectation values, as we will establish in §5. In this way, the $\ell_r$ will define a particular shape of Young diagram that dominates the reduced density matrix of each subregion.

## 4.2 Entropy from the Young diagram

In this section we recall how the dimension of an irrep $\mu$ of $U(M)$ can be obtained from the row lengths $\ell_r$, and thereby obtain the entropy. We have seen in (4.6) that the two-sided Young diagrams are symmetric — we deal with these by adjoining a factor of 2 to the analysis for single-sided YDs below. Indexing the rows of the Young diagram by $r$ and the

columns by $c$, the dimension of an irrep $\mu$ of $PU(M)$ is given by (see e.g. chapter 5 of [82])

$$\log \dim \mu = 2 \sum_{r,c} \Big[ \log(M + c - r) - \log(\mathrm{hook}(r,c)) \Big]. \tag{4.7}$$

For a given position $(r, c)$, $\mathrm{hook}(r, c)$ is the hook length, defined as $1 +$ the number of boxes to the right of $(r, c) +$ the number of boxes below $(r, c)$. That is

$$\log(\mathrm{hook}(r,c)) = \log(1 + \ell_r - c) + \log\Big(1 + \frac{d_{r,c}}{1 + \ell_r - c}\Big), \tag{4.8}$$

where $d_{r,c}$ counts the number of boxes below the box at position $(r, c)$. We may now perform the sum row by row

$$\log \dim \mu = 2 \sum_{r} \left[ \log \binom{M + \ell_r - r}{\ell_r} - \sum_{c} \log\Big(1 + \frac{d_{r,c}}{1 + \ell_r - c}\Big) \right]. \tag{4.9}$$

There are two cases of note to consider in (4.9), the 'flat' and 'tall' diagrams illustrated in Fig. 5. The 'flat' diagrams, denoted by $\mu_f$, have a single row of length $\ell_1 \gg 1$. These give the dominant representation for the cap subregion considered in [36] and will describe the saddle point configuration in §5. The second term in (4.9) vanishes because $d_{r,c} = 0$ for these diagrams. The regime of interest to us will be $M \gg \ell_1 \gg 1$, wherein the first term in (4.9) gives

$$\log \dim \mu_f \approx 2\ell_1 \log \frac{eM}{\ell_1}. \tag{4.10}$$

We have already recalled in (3.14) that in the regularised fuzzy sphere state (3.11) the row length $\ell_1 \sim |\partial \Sigma|$, the boundary area of the cap subregion.

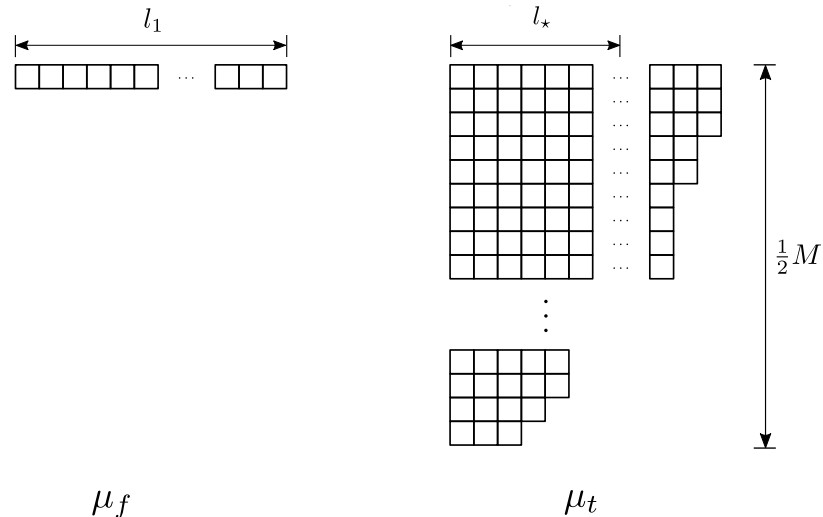

Figure 5: Two classes of YD for which we explicitly evaluate (4.9). Left: a 'flat' YD of only one row — the saddle point configuration is of this form. Right: a 'tall' YD with $\mathcal{O}(M/2)$ rows. A generic partition of the space (considered in §5.2) will be of this form.

The 'tall' diagrams, denoted by $\mu_t$, have $\mathcal{O}(M/2)$ rows, which is the most allowed for an irrep built from degrees of freedom transforming in the adjoint of $U(M)$. Furthermore, the diagrams will be thinner than they are tall, so that the row lengths obey $1 \ll \ell_r \ll M/2$. For these cases it will be sufficient to bound the order of magnitude of the dimension to leading order in $M$. We will obtain both an upper and a lower bound. An upper bound on the dimension follows from the fact that the second term in (4.9) is negative so that

$$\log \dim \mu_t \leq 2 \sum_r \log \binom{M + \ell_r - r}{\ell_r} \approx 2 \sum_r \ell_r \log \frac{e(M - r)}{\ell_r}$$

$$\leq 2 \sum_r \ell_r \log \frac{eM}{\ell_r} \sim M\ell_\star \log \frac{eM}{\ell_\star}, \tag{4.11}$$

where in the last line $\ell_\star$ is a typical row length. A lower bound is obtained by noting that $d_{r,c} \leq M/2$, the maximum depth of the diagram. Using this inequality to perform the sum over $c$ in (4.9), and with the same asymptotic formula for the binomial as in (4.11),

$$\log \dim \mu_t \gtrsim 2 \sum_r \left[ \ell_r \log \frac{e(M - r)}{\ell_r} - \ell_r \log \frac{eM}{2\ell_r} \right]$$

$$= 2 \sum_r \ell_r \log \frac{2(M - r)}{M} \gtrsim M\ell_\star. \tag{4.12}$$

For the final estimate we may note that all the terms in the final sum are positive as $r \leq M/2$. Therefore we may restrict to $r \leq M/4$, say, to get a lower bound.

The two bounds above show that in terms of parametric scaling

$$\log \dim \mu_t \sim M\ell_\star, \tag{4.13}$$

up to at most a logarithmic correction.

## 5  The minimal area formula

In this section we will prove the minimal area formula (1.1) for the fuzzy sphere state (3.11) of the bosonic mini-BMN model (3.8). We use the prescription introduced in §2, the model discussed in §3 and the representation theoretic facts from §4.

We firstly make a convenient choice of frame transformation group $G$ for the factorisation map (2.12). Recall that the role of this group is to coarse-grain the integral over $U(N)$ and hence coarse-grain the emergent volume-preserving diffeomorphisms. We will coarse-grain $U(N)$ to $U(N')$ where

$$N' \equiv \frac{N}{p}. \tag{5.1}$$

We will have $N \gg p$, so that both $N$ and $N'$ can be taken to be integers. There are many ways to embed $U(N')$ into $U(N)$, corresponding to different choices of basis. We take

$$G = U(N') \otimes \mathbb{1}_p \subset U(N), \tag{5.2}$$

in the basis where $X_{\rm cl}^3$ is diagonal with ordered eigenvalues. In this basis $G$ commutes with the block matrices

$$
\mathbb{1}_{N'} \otimes A =
\begin{bmatrix}
A & 0 & 0 & 0 & \ldots & 0 \\
0 & A & 0 & 0 & \ldots & 0 \\
0 & 0 & A & 0 & \ldots & 0 \\
0 & 0 & 0 & A & \ldots & 0 \\
\vdots & \vdots & \vdots & \vdots & \ddots & \vdots \\
0 & 0 & 0 & 0 & \ldots & A
\end{bmatrix},
\tag{5.3}
$$

for any $p \times p$ matrix $A$. Similar blockings were considered before in [83, 84]. We set $M' \equiv M/p$ and, for the same reason as above, take $M'$ to be an integer; so there is a natural $U(M') \times U(N' - M') \subset U(N')$. This may be seen from the fact that $\Theta_M$ may be written as $\Theta'_{M/p} \otimes \mathbb{1}_p$ for an $N' \times N'$ projector $\Theta'_{M/p}$. In particular, this means that in (2.38),

$$
H = U(M') \otimes \mathbb{1}_p, \qquad H' = U(N' - M') \otimes \mathbb{1}_p.
\tag{5.4}
$$

As there are various parameters involved in the discussion, let us be clear about their $N$ scalings. In the remainder we are assuming the following scalings:

$$
N \to \infty, \qquad M = \mathcal{O}(N), \qquad 1 \ll \nu, \Lambda = \mathcal{O}(N^0), \qquad p = \mathcal{O}(N^\gamma), \text{ with } \gamma \in [0, 1). \tag{5.5}
$$

More general scalings may be possible, but the above are sufficient for our purposes. The upper bound on $\gamma$ ensures that $N'$ and $M'$ remain large in the large $N$ limit.

## 5.1 The structure of the frame average integral

We saw in (2.42) that the reduced density matrix can be approximated as a sum over orthogonal terms labelled by the transformations $V$. As in the paragraph below (2.43), we write this as

$$
\rho_{\rm in} = \oint_{\mathsf{F}} dV \, \rho_{{\rm in},V} \, |V\rangle \langle V| .
\tag{5.6}
$$

The flag manifold $\mathsf{F}$ was defined in §2.6. We do not normalise $\rho_{\rm in}$, as this is unnecessary in computing the entanglement using (5.9) below. The $n$-purity is thus approximately

$$
Z_n \equiv {\rm tr}\, \rho_{\rm in}^n = \delta(0) \int_{\mathsf{F}} dV \, {\rm tr}\, \rho_{{\rm in},V}^n \equiv \delta(0) \int_{\mathsf{F}} dV \, e^{-I_n(V)}.
\tag{5.7}
$$

The factor of $\delta(0)$ was discussed in §2.6. In §5.4 below we will see that $\delta(0)$ scales like a power of $N$ and is therefore subleading to the integral, which is exponential in $N$. We drop this factor henceforth. Our objective in the remainder is to evaluate the integral in (5.7). Because the exponent is large, one may hope to perform the integral by a saddle point analysis. However, the dimension of $\mathsf{F}$ is also large and so the 'energetic' dominance of the saddle point may be challenged by the large 'entropic' contribution of all the different possible frames. We have illustrated the situation schematically in Fig. 6. With the choice of basis in (3.2) the maximum will be seen to be at $V = \mathbb{1}$, corresponding to the spherical cap subregion. Although the integrand becomes very small away from the maximum, the

width of the saddle point region is also very small.

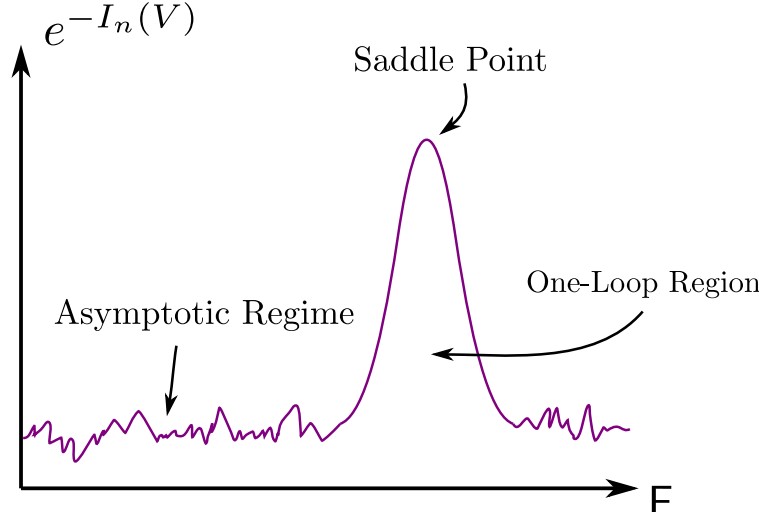

Figure 6: Schematic plot of the integrand $e^{-I_n(V)}$ in (5.7). The integrand peaks at the saddle point, and then drops to a regime of strong fluctuations about a much smaller average value. The width of the saddle point peak has been exaggerated in the figure. Our analysis will decompose the integral into two contributions: a one-loop integral about the saddle point and the average value of the fluctuations (recalling that we have normalized vol $\mathsf{F} = 1$).

We approximate (5.7) by splitting it into two pieces with distinct parametric scaling. The first is a 'one-loop' integral capturing the peak around the maximum and the second is a term proportional to the volume of $\mathsf{F}$ weighted by the generic value of the integrand:

$$Z_n = Z_{\text{loop}} e^{-I_n(\mathbb{1})} + e^{-\overline{I_n}}, \tag{5.8}$$

where $\overline{I_n}$ is the average of $I_n$ over $\mathsf{F}$. In order to be consistent with the computations in §2, we must work with a measure such that vol $\mathsf{F} = 1$. This is why there is no explicit prefactor of the volume of $\mathsf{F}$ in (5.8). It will be important to keep track of this measure when we compute the one-loop contribution, which is the volume of the saddle point region. The reader will notice that in (5.8) the average over $\mathsf{F}$ has been moved into the exponent. This step will be justified below, to an extent, when we show that the variance in $I_n$ is small compared to the average $\overline{I_n}$.

In the following subsections we will evaluate the three terms in (5.8) and show that for sufficiently large values of the coarse-graining parameter $p$ in (5.1), the saddle point value dominates over both the one-loop and the generic contributions. Ensuring the saddle point dominance was, as we described in the introduction and §2.4, our primary motivation for coarse-graining the average over frames in the first place. With the $n$-purities at hand, we will evaluate the entanglement entropy as

$$S_E = \lim_{n \to 1} (1 - n \partial_n) \log Z_n \,. \tag{5.9}$$

We may recall that it is not necessary to normalise the density matrix in this expression.

Normalisation proceeds by $Z_n \to Z_n/Z_1^n$, but the denominator then drops out of (5.9).

**Summary of results of this section**

As some of the discussion below is technically involved, let us now summarise the results of the rest of this section and show how the minimal area formula arises. As a function of $N, M$ and $p$, we will find the following results for the various objects in (5.8):

$$
\begin{aligned}
\S5.2: \qquad & \overline{I_n} \gtrsim (n-1)(\nu\Lambda)^{3/2}M\sqrt{\frac{M(N-M)}{N}}\,, \\[2mm]
\S5.3: \qquad & I_n(\mathbb{1}) = \frac{n-1}{\sqrt{3}}(\nu\Lambda)^{3/2}\sqrt{\frac{M(N-M)}{N}}\log\left(\frac{4\pi\,M\,N}{(\nu\Lambda)^3\,(N-M)}\right)\,, \\[2mm]
\S5.4: \quad \log Z_{\text{loop}} \approx & -\frac{\alpha}{p^2}M(N-M)\log N\,, \qquad \text{with } \alpha \geq \frac{1}{2}\,.
\end{aligned}
\tag{5.10}
$$

While the saddle point value $I_n(\mathbb{1}) \ll \overline{I_n}$, as is of course expected, the small volume of the saddle point region — the one-loop contribution — means that the generic contribution dominates the integral unless $p \gg N^{1/4}$. In addition, the stronger condition $p \gg N^{3/4}$ is needed in order for the saddle point value itself to determine the value of the integral, rather than the one-loop term. Finally, recall from (5.5) that $p \ll N$. All told, for

$$
N^{3/4} \ll p \ll N,
\tag{5.11}
$$

and for $n$ in the regime (2.47), we find that (5.8) becomes

$$
Z_n \approx e^{-I_n(\mathbb{1})}\,.
\tag{5.12}
$$

Using this result to obtain the entropy (5.9) gives (1.1), advertised in the introduction.

In (5.10) the technical consequence of coarse-graining is to introduce a relative factor of $p^2$ between the values of the action, $I_n(\mathbb{1})$ and $\overline{I_n}$, and the measure on phase space, as manifest in the one-loop contribution. An alternative way to achieve this same effect is to integrate over all of $U(N)$ but consider $N_{\text{f}}$ separate flavours of matrices on which this $U(N)$ symmetry acts simultaneously. With $N_{\text{f}}$ flavours the values of the action get enhanced by a factor of $N_{\text{f}}$ while the one-loop term is unchanged, up to a logarithmic factor.

The three following subsections will derive the results listed in (5.10).

## 5.2   The generic term

In this subsection we will calculate the average action

$$
\overline{I_n} = -\int_{\mathsf{F}} \mathrm{d}V \log \operatorname{tr} \rho_{\text{in},V}^n = -\int_G \mathrm{d}g \log \operatorname{tr} \rho_{\text{in},g}^n\,.
\tag{5.13}
$$

The second equality increases the integration range from $\mathsf{F} = U(M') \times U(N'-M')\backslash U(N')$ to $U(N')$, using the facts that transformations in $U(M') \times U(N'-M')$ do not change the density matrix, and that the volume of $U(M') \times U(N'-M')$ does not depend on $V$. Recall

that $\mathrm{vol}\, \mathsf{F} = 1$. The Haar integral over $G = U(N')$ is also normalised so that $\mathrm{vol}\, G = 1$.

The key technical computations are of multi-point functions of the Casimirs. To reduce clutter in the rest of this section we will drop the $\Sigma$ subscript from $Q_\Sigma$, and define the Casimirs as

$$C_{2s} \equiv \mathrm{Tr}\, Q^{2s} \,. \tag{5.14}$$

Our calculations will establish the following:

1. The expectation value of the second Casimir is self-averaging over $G$:

$$\frac{\overline{\langle C_2\rangle^2} - \overline{\langle C_2\rangle}^2}{\overline{\langle C_2\rangle}^2} \sim \frac{1}{N'^2} \ll 1 \,. \tag{5.15}$$

2. The quantum variance $\Delta^2_{C_2} \equiv \langle C_2^2\rangle - \langle C_2\rangle^2$ of the second Casimir is also self-averaging:

$$\frac{\overline{\left(\Delta^2_{C_2}\right)^2} - \left(\overline{\Delta^2_{C_2}}\right)^2}{\overline{\Delta^2_{C_2}}^2} \sim \frac{1}{N'^2} \ll 1 \,. \tag{5.16}$$

3. The quantum variance of the Casimir is small compared to its expectation value

$$\frac{\overline{\Delta^2_{C_2}}}{\overline{\langle C_2\rangle}^2} \sim \frac{1}{\Lambda^2} \ll 1 \,. \tag{5.17}$$

Recall that $\Lambda$ is the angular momentum cutoff in the regularised state (3.11).

The above statements extend to the general Casimir $C_{2s}$. As we saw in §4, the Casimirs determine the representation. The first two statements therefore indicate that at large $N'$ the quantum distribution over irreps has the same average and width for all generic $V \in \mathsf{F}$. The final statement then shows that at large $\Lambda$ the quantum state of a generic $V$ is dominated by a single irrep $\mu_*$ of $U(M)$. We will estimate the shape of its Young diagram by calculating $\overline{\langle C_{2s}\rangle}$. From the Young diagram we can use the results in §4 to calculate the dimension $d_{\mu_*}$ of the irrep. Recalling from e.g. (2.42) that the density matrix is maximally mixed over the representation, we obtain

$$\overline{I_n} \approx (n-1)\log d_{\mu_*} \,. \tag{5.18}$$

**Calculation of the second Casimir**

We will first calculate $\overline{\langle \mathrm{Tr}\, Q^2\rangle}$, since it is the simplest quantity and therefore a good setting to introduce the technique. The semiclassical $U(M)_\Sigma$ generators are given by (4.4). The second Casimir is then, in the QRF corresponding to some $g \in G$,

$$\langle \mathrm{Tr}\, Q_g^2\rangle = -4 \sum_{lm;l'm'} \langle \delta\pi_{lm}\delta\pi_{l'm'}\rangle$$

$$\mathrm{Tr}\left[\left(\Theta_g X^a_{\mathrm{cl}}\overline{\Theta}_g Y^a_{lm}\Theta_g - \Theta_g Y^a_{lm}\overline{\Theta}_g X^a_{\mathrm{cl}}\Theta_g\right)\left(Y^a_{lm} \to Y^a_{l'm'}\right)\right] \,, \tag{5.19}$$

where $\Theta_g \equiv g^\dagger \Theta g$, and similarly for $\overline{\Theta}$. The quantum expectation value is, from (3.11),

$$\langle \delta\pi_{lm}\delta\pi_{l'm'}\rangle = \frac{\nu}{2}|\omega_{lm}|\,\delta_{ll'}\delta_{mm'}. \tag{5.20}$$

We can now Haar integrate (5.19) over $G = U(N')$ to obtain its average. Recall that the measure is normalised, so that $\int_G \mathrm{d}g = 1$. To leading order in $N'$, Haar averages simplify via Wick contractions [85]. For example,

$$\overline{g^\dagger_{ij}g_{kl}g^\dagger_{pq}g_{rs}} = \overline{g^\dagger_{ij}g_{kl}g^\dagger_{pq}g_{rs}} + \overline{g^\dagger_{ij}g_{kl}g^\dagger_{pq}g_{rs}} + \dots$$
$$= \frac{\delta_{il}\delta_{kj}\delta_{ps}\delta_{qr} + \delta_{is}\delta_{jr}\delta_{kq}\delta_{pl}}{N'^2} + O(N'^{-3})\,. \tag{5.21}$$

The general rule, as usual, is to contract all pairs of $g^\dagger$ and $g$. A power of $1/N'$ appears for every contraction. We introduce a graphical notation for these calculations, denoting a $g^\dagger$ by a downward arrow and a $g$ by an upward one, so that (5.21) becomes

$$\tag{5.22}$$

The quantum expectation value $\langle \operatorname{Tr} Q^2 \rangle$ contains 4 terms, each of which we have to Haar integrate. To proceed systematically we introduce the auxiliary quantities,

$$\mathsf{Q}(A,B) \equiv g^\dagger\Theta g A g^\dagger\overline{\Theta}g B g^\dagger\Theta g, \qquad \mathsf{Q}[A,B] \equiv \mathsf{Q}(A,B) - \mathsf{Q}(B,A)\,, \tag{5.23}$$

such that

$$Q = -2i\sum_a \mathsf{Q}[X_{\mathrm{cl}}^a, \Pi^a]\,. \tag{5.24}$$

We need to calculate Haar averages of traces made out of $\mathsf{Q}[A,B]$.

We will illustrate the technology first in the simpler case $p = 1$ so that $N' = N$. We will discuss $p \neq 1$ at the end of this computation. To start with, consider $\operatorname{Tr}\mathsf{Q}(A,B) = \operatorname{Tr}\left(g^\dagger\Theta g A g^\dagger\overline{\Theta}g B\right)$, where we used the fact that the projector $\Theta_g^2 = \Theta_g$. We then have

$$= \frac{1}{N^2}\left[\operatorname{Tr}A\operatorname{Tr}B\operatorname{Tr}\Theta\overline{\Theta} + \operatorname{Tr}AB\operatorname{Tr}\Theta\operatorname{Tr}\overline{\Theta}\right]$$
$$= \frac{M(N-M)}{N^2}\operatorname{Tr}AB. \tag{5.25}$$

In the last equality, we have used the fact that $\operatorname{Tr}\Theta\overline{\Theta} = 0$ because they are orthogonal projectors. Since this particular simplification will recur in our work, we will colour-code

our diagrammatics such that every line entering or emerging from a $\Theta$ ($\overline{\Theta}$) will be coloured blue (green). We then take as a rule that only like colour lines can contract.

Now consider $\text{Tr}[\mathsf{Q}(A,B)\mathsf{Q}(C,D)] = \text{Tr}\left(g^\dagger\Theta g A g^\dagger\overline{\Theta} g B g^\dagger\Theta g C g^\dagger\overline{\Theta} g D\right)$,

$$\overline{\text{Tr}\,\mathsf{Q}(A,B)\mathsf{Q}(C,D)} = \qquad\qquad\qquad\qquad\qquad . \tag{5.26}$$

There are four possible ways to contract:

$$= \frac{M^2(N-M)^2}{N^4}\,\text{Tr}\,ABCD \sim N\mathcal{O}(ABCD)$$

$$= \frac{M(N-M)^2}{N^4}\,\text{Tr}\,AB\,\text{Tr}\,CD \sim N\mathcal{O}(AB)\mathcal{O}(CD)$$

$$\tag{5.27}$$

$$= \frac{M^2(N-M)}{N^4}\,\text{Tr}\,BC\,\text{Tr}\,AD \sim N\mathcal{O}(BC)\mathcal{O}(AD)$$

$$= \frac{M(N-M)}{N^4}\,\text{Tr}\,ADCB \sim \frac{1}{N}\mathcal{O}(ADCB).$$

The final term in each line, following the "$\sim$," indicates the overall $N$ scaling, in which $M \sim N - M \sim N$ and $\text{Tr} \sim N$. The $\mathcal{O}(ABCD)$ terms indicate any explicit factors of $N$ that might occur in the matrix elements or in their overlaps. This $N$ counting highlights that the final diagram in (5.27) is subleading; this will be true of all non-planar diagrams (i.e. those with crossing lines) and we will drop them in subsequent calculations.

The expressions above give the following leading order result for $\overline{\langle\text{Tr}\,Q^2\rangle}$:

$$\overline{\langle\text{Tr}\,Q^2\rangle} = -2\nu\sum_{lm}|\omega_{lm}|\frac{M(N-M)}{N^4}\left\{M(N-M)\,\text{Tr}\left([Y^a_{lm},X^a_{\text{cl}}][Y^b_{lm},X^b_{\text{cl}}]\right)\right.$$
$$\left. + 2M\left(\text{Tr}\left[Y^a_{lm}X^b_{\text{cl}}\right]\text{Tr}\left[Y^b_{lm}X^a_{\text{cl}}\right] - \text{Tr}\left[Y^a_{lm}Y^b_{lm}\right]\text{Tr}\left[X^a_{\text{cl}}X^b_{\text{cl}}\right]\right)\right\}\,. \tag{5.28}$$

The $N$ scaling of each term in (5.28) may be obtained using the following facts. Since $\text{Tr}\,X^2_{\text{cl}} \sim N^3$, the matrix elements of $X_{\text{cl}}$ scale as $X_{\text{cl},ij} \sim N$. Since $\text{Tr}\,Y^2 \sim N^0$, see

Appendix B, the $Y$ matrix elements scale as $Y_{lm,ij} \sim N^{-1/2}$. These scalings are sufficient to estimate the last term in (5.28). For the middle term we must note that since $X_{\rm cl}^a \propto Y_{1,0}^a$ is a spin-1 matrix, the trace $\mathrm{Tr}[Y_{lm}X_{\rm cl}]$ can only be non-zero for $l \leq 1$, by the Wigner-Eckart theorem. Finally, the commutators appearing in the first term must also be treated separately. Within noncommutative geometry, see (5.49) below for a precise formula, the commutator is equivalent to a derivative, so that $[X_{\rm cl}, Y_{lm}]_{ij} \sim l N^{-1/2}$.

Using the scalings described in the previous paragraph, and recalling that $X_{\rm cl}$ is proportional to $\nu$ and that $\omega_{lm} \sim l$ at large $l$, the three terms in (5.28) respectively give

$$\overline{\langle \mathrm{Tr}\, Q^2 \rangle} \sim \nu^3 \left( \Lambda^5 + N^2 + \Lambda^3 N^2 \right) \sim \nu^3 \Lambda^3 \frac{M^2(N-M)}{N}. \tag{5.29}$$

As always, $\Lambda$ is the cutoff on $l$ from the regularised wavefunction (3.11). In the final step we have recalled from (5.5) that $\Lambda \sim N^0$ and we have also re-instated the detailed dependence of the dominant (final) term on $M$.

The case with $N' = N/p \ll N$ proceeds similarly to above. As in (5.2), we decompose the $N$-dimensional colour space $\mathbb{C}^N$ as $\mathbb{C}^{N'} \otimes \mathbb{C}^p$, so that we Haar integrate over $U(N') \otimes \mathbb{1}_p$. Any matrix $M \in \mathrm{Mat}(\mathbb{C}^N)$ may be decomposed as

$$M = \sum_\alpha M_\alpha \otimes \mathfrak{b}_\alpha, \qquad \mathrm{Tr}[\mathfrak{b}_\alpha \mathfrak{b}_{\alpha'}] = \delta_{\alpha\alpha'}, \tag{5.30}$$

where the $\mathfrak{b}_\alpha$'s form a basis for the $p^2$-dimensional space $\mathrm{Mat}(\mathbb{C}^p)$. We perform the calculation of the second Casimir in detail in Appendix C and find the same result (5.29), which was derived for $p = 1$. As we explain in that appendix, this Casimir does not depend on $p$ to leading order.

**Haar variances of expectation values**

We now argue that all expectation values of powers of Casimirs are self-averaging. This means that they take the same value, at leading order, for almost all $V \in \mathsf{F}$. Specifically, given $O = \mathrm{Tr}\, Q^{2s_1} \mathrm{Tr}\, Q^{2s_2} \ldots$ and $O' = \mathrm{Tr}\, Q^{2s_1'} \mathrm{Tr}\, Q^{2s_2'} \ldots$, we argue that

$$\frac{\overline{\langle O \rangle \langle O' \rangle} - \overline{\langle O \rangle}\, \overline{\langle O' \rangle}}{\overline{\langle O \rangle}\, \overline{\langle O' \rangle}} \sim \frac{1}{N'^2}. \tag{5.31}$$

In particular, this includes the results (5.15) and (5.16) quoted above.

The diagrams that contribute to the Haar variance (5.31) are those where the contractions connect the two operators. These diagrams are subleading. We will illustrate this with the simple example $\mathrm{Tr}\, \mathsf{Q}(A,B)$, that we considered above. We can rearrange (5.25), and indeed any diagram corresponding to a single trace, into a circle:

$$\overline{\mathrm{Tr}\big(g^\dagger \Theta g A g^\dagger \overline{\Theta} g B\big)} = \quad  \quad \supset \quad  \quad \sim \quad N' \mathcal{O}(AB). \tag{5.32}$$

This is equivalent to (5.25), now written as a familiar double-line diagram of disk topology. The boundary of the disk corresponds to the original trace. We also have that $N \to N'$ as we are now working with general $p > 1$.

Now consider two traces, $\overline{\text{Tr}\big(g^\dagger \Theta g A g^\dagger \overline{\Theta} g B\big) \text{Tr}\big(g^\dagger \Theta g A g^\dagger \overline{\Theta} g B\big)}$. This is represented by a diagram whose boundary contains two circles. The Haar variance is given by diagrams that connect the two boundaries, like

$$= \quad \frac{M'^2(N' - M')}{N'^4} \, \text{Tr}\, A^2 B^2 \sim \mathcal{O}(A^2 B^2) \,. \quad (5.33)$$

These 'wormhole' diagrams have cylinder topology. The Euler characteristic of a cylinder is $\chi = 0$, whereas that of two disks is $\chi = 2$. This results in the familiar $N'^{-2}$ suppression of the wormhole [86]. Specifically, the result (5.33) for the variance is down compared to square of the result (5.32) for the expectation value by a factor of $N'^2$.

The same argument works for general products of traces, giving (5.31). The conclusion is that to leading order at large $N'$ the Haar average can be taken on each trace separately.

**Quantum variance of the second Casimir**

We now show that for a generic $g$ the variance of the second Casimir is small. Based on this fact we argue that the state for generic $g$ is dominated by a single $U(M)$ irrep. We wish to compute the Haar averaged quantum variance

$$\overline{\Delta^2_{C_2}} \equiv \overline{\langle \text{Tr}\, Q^2 \, \text{Tr}\, Q^2 \rangle - \langle \text{Tr}\, Q^2 \rangle \langle \text{Tr}\, Q^2 \rangle}$$
$$\approx \overline{\langle \text{Tr}\, Q^2 \, \text{Tr}\, Q^2 \rangle} - \overline{\langle \text{Tr}\, Q^2 \rangle}^2 \,. \quad (5.34)$$

The second line used (5.31), at large $N'$. The terms that contribute to (5.34) at leading order are those where the Haar average has the topology of two disks, but the quantum Wick contractions of the $\delta\pi$ operators connect the two Casimirs. The leading term is then, cf. the final term in (5.28),

$$\overline{\Delta^2_{C_2}} \sim \frac{M'^4(N' - M')^2}{N'^8} \, \text{Tr}\Big[X^a_{\text{cl}} X^b_{\text{cl}}\Big] \text{Tr}\Big[X^c_{\text{cl}} X^d_{\text{cl}}\Big]$$
$$\times \sum_{lm, l'm'} \nu^2 |\omega_{lm} \omega_{l'm'}| \, \text{Tr}\Big[Y^a_{lm} Y^b_{l'm'}\Big] \text{Tr}\Big[Y^c_{lm} Y^d_{l'm'}\Big]$$
$$\sim \nu^6 N^4 \sum_{lm} l^2 \sim \nu^6 N^4 \Lambda^4 \,. \quad (5.35)$$

Here, we have used the fact that $\text{Tr}[Y_{lm} Y_{l'm'}]$ vanishes unless $l = l'$ and $m = m'$. We also used the facts, from Appendix C, that after decomposing the matrices according to (5.30) one has (i) $\text{Tr}\big[Y^2_\alpha\big] \sim 1/p$, (ii) $\text{Tr}\big[X^2_\alpha\big] \sim N'N^2$ and (iii) an additional factor of $p^2$ from two traces over the $\mathfrak{b}_\alpha$ elements.

The result (5.35) is down by a factor of $\Lambda^2$ compared to the square of the expectation

value of $C_2$ in (5.29). Thus we recover the advertised result (5.17). The quantum variance is small, consistent with the state being supported on a single irrep, up to $\mathcal{O}(1/\Lambda)$ corrections.

### The higher Casimirs and $\overline{I_n}$

To characterise the irrep of the reduced density matrix, we need to know the higher Casimirs too. We will explicitly discuss the fourth Casimir, $\overline{\langle \operatorname{Tr} Q^4 \rangle}$, before noting how the logic generalises to higher cases. For $\overline{\langle \operatorname{Tr} Q^4 \rangle}$ the most important diagram is analogous to the previously dominant diagrams we have discussed, which here gives $(\operatorname{Tr} X^2 \operatorname{Tr} Y^2)^2$:

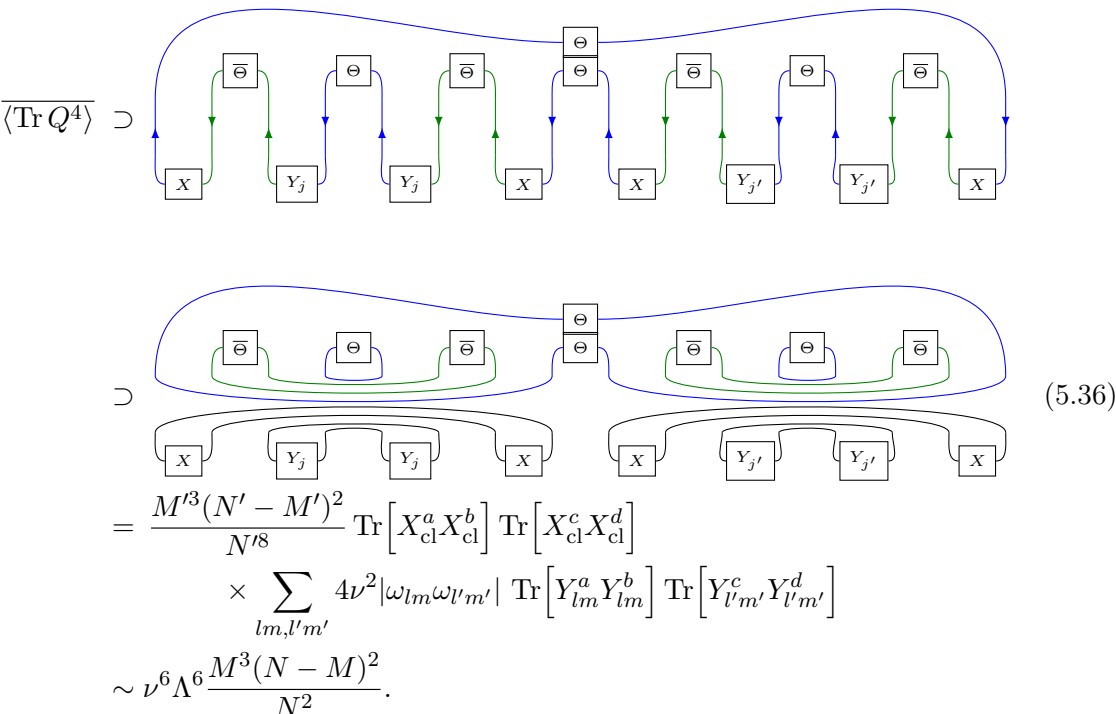

$$\begin{aligned}
&= \frac{M'^3 (N' - M')^2}{N'^8} \operatorname{Tr}\left[X_{\mathrm{cl}}^a X_{\mathrm{cl}}^b\right] \operatorname{Tr}\left[X_{\mathrm{cl}}^c X_{\mathrm{cl}}^d\right] \\
&\quad \times \sum_{lm, l'm'} 4\nu^2 |\omega_{lm} \omega_{l'm'}| \operatorname{Tr}\left[Y_{lm}^a Y_{lm}^b\right] \operatorname{Tr}\left[Y_{l'm'}^c Y_{l'm'}^d\right] \\
&\sim \nu^6 \Lambda^6 \frac{M^3 (N - M)^2}{N^2}.
\end{aligned}$$

(5.36)

We can argue that there are no diagrams that dominate over this one as follows. The full expression for $\operatorname{Tr} Q^4$ contains four $X$s, four $Y$s, and the sums over $l$ and $l'$. Let us naïvely calculate the $N$ scaling; we find $X^4 \sim N^4, Y^4 \sim N^{-2}$ and the trace gives another power of $N$, and so $\operatorname{Tr} Q^4 \sim N^3$. The $l, l'$ sums give a $\Lambda^6$. This naïve scaling can only possibly upper bound the true scaling of any term: commutators $[X, Y]$ reduce the power of $N$ and factors like $\operatorname{Tr} Y X$ truncate the sum over $l, m$ to values much lower than $\Lambda$. We met both of these effects in our discussion below (5.28) above. We have then also shown in (5.36) that that the naïve scaling is achieved in one term in which none of these reductions happen.

The argument given in the previous paragraph easily generalises to show that $\overline{\langle \operatorname{Tr} Q^{2s} \rangle} \sim \nu^{3s} \Lambda^{3s} M[M(N - M)/N]^s$; the relevant diagram is always the one that gives $\left(\operatorname{Tr} X^2 \operatorname{Tr} Y^2\right)^s$. This is, of course, also consistent with the previous result (5.29) for the second Casimir.

These Casimirs are consistent with $Q$ having $\mathcal{O}(M)$ singular values of typical magnitude

$$\ell_i \sim \sqrt{(\nu\Lambda)^3 \frac{M(N-M)}{N}}\,, \tag{5.37}$$

since with these values $\sum_i \ell_i^{2s} \sim \overline{\langle \mathrm{Tr}\, Q^{2s} \rangle}$. This indicates that the dominant representation in a QRF of a typical unitary is one whose Young diagram is 'tall' in the language of §4. The row lengths in (5.37) are symmetric under $M \leftrightarrow N-M$; it is only the number of rows that keeps track of whether the off-diagonal blocks are in the same subsystem as $\Sigma$ or $\overline{\Sigma}$.

We may use the typical singular value (5.37) in the formula (4.13) to obtain the dimension of the irrep, and hence the entanglement, of a generic partition:

$$\overline{I_n} \approx (n-1)\log d_{\mu_*} \gtrsim (n-1)(\nu\Lambda)^{3/2} M \sqrt{\frac{M(N-M)}{N}}\,. \tag{5.38}$$

This is our main result regarding the generic contribution, quoted above in (5.10).

## 5.3   The saddle point

In this subsection and the following we will calculate the saddle point contribution to the integral (5.7) over reference frames. This saddle point will be at $V = \mathbb{1}$. At this point the projector is simply $\Theta$ in (2.13), written in the QRF where $X^3$ is diagonal and ordered. The corresponding subregion is thus the spherical cap of the fuzzy sphere discussed in [36] and §3.4 above, containing the 'north pole' and with a boundary at fixed latitude.

Later in this subsection we will give an analysis of the perturbations about the cap subregion, which will establish that it is a local minimum of the 'action' in the integral (5.7). A short argument to reach this conclusion goes as follows: we have explained how, under the Moyal map, (5.7) is an integral over subregions of $S^2$ with fixed volume. It was argued in [37] that connected subregions with low-curvature boundaries correspond to irreps with 'flat' Young diagrams in the sense of §4. These irreps have a much lower action than the 'tall' irreps corresponding to generic subregions and are therefore candidate minima. Furthermore, the entanglement entropy for the low-curvature regions is proportional to the boundary area (up to a log correction) [37]. The region minimising this boundary area, given a fixed volume, is the one with an $S^1$ boundary, i.e. a 'cap' on the fuzzy sphere.[15]

In [36], it was found that for this cap region

$$I_n(\mathbb{1}) = \frac{n-1}{\sqrt{3}}(\nu\Lambda)^{3/2}\sqrt{\frac{M(N-M)}{N}}\log\left(\frac{4\pi\, M\, N}{(\nu\Lambda)^3\,(N-M)}\right)\,. \tag{5.39}$$

It is clear that this saddle point action is less than the generic action (5.38) by a factor of $1/M$. This is the formula quoted in (5.10). As in [36], it is instructive to write the answer in

---

[15]Clearly there is in fact an $SO(3)$ family of saddle points, given by rotating the cap around the sphere. These different saddle points correspond to QRFs where some matrix $R^3{}_b X^b$ is diagonalised, for $R \in SO(3)$. There are no cross-terms from these different saddles in the reduced density matrix (2.41), since different matrices are diagonalised in the different components. Thus, including these saddle points merely gives a subleading $\mathcal{O}(\log \mathrm{vol}\, SO(3))$ additive contribution to the entanglement entropy, which we will neglect.

terms of emergent geometric quantities: the coupling $g_M$ of the IR noncommutative Maxwell theory that describes the low-energy effective dynamics about the fuzzy sphere, as well as the area $|\Sigma_{\text{cap}}|$ and the boundary length $|\partial\Sigma_{\text{cap}}|$ of the cap in units where $S^2$ has surface area 1. These are given explicitly by:

$$g_M = \sqrt{\frac{4\pi}{N\nu^3}} , \qquad |\Sigma_{\text{cap}}| = 4\pi\frac{M}{N} , \qquad |\partial\Sigma_{\text{cap}}| = 4\pi\frac{\sqrt{M(N-M)}}{N} . \qquad (5.40)$$

The expression (5.39) becomes

$$I_n(\mathbb{1}) = \frac{n-1}{\sqrt{3\pi}}\frac{\sqrt{\Lambda}}{g_M}\frac{|\partial\Sigma_{\text{cap}}|}{1/\Lambda}\log\frac{g_M N|\Sigma_{\text{cap}}|}{\Lambda^{3/2}|\partial\Sigma_{\text{cap}}|} . \qquad (5.41)$$

We have given the answer in this form in (1.1). As we noted there, $\Lambda$ is the short distance cutoff that makes up the units of areas and volumes. The 2+1 dimensional Maxwell coupling $g_M$ is also dimensionful, and the dimensionless coupling at the scale $\Lambda$ is $\sqrt{\Lambda}/g_M$.

We now turn to proving that the spherical cap subregion corresponds to a local minimum of $I_n(V)$. Perturbations are given by the adjoint action of infinitesimal $V \in \mathsf{F}$ on the projector $\Theta$. We can write $V = e^{i\epsilon H}$ and expand in small $\epsilon$,

$$\Theta_\Sigma = V^\dagger\Theta V \approx \Theta - i\epsilon[H,\Theta] - \frac{\epsilon^2}{2}[H,[H,\Theta]] \equiv \Theta + \delta_H\Theta . \qquad (5.42)$$

This perturbation of the projector then gives a perturbation to the generators $Q_\Sigma$ in (4.4), and hence to the eigenvalues of $Q_\Sigma$. Recall from (4.6) that these eigenvalues are (plus and minus) the Young diagram row lengths, so we label them by $\ell_r \geq 0$. The perturbations to the eigenvalues are then

$$\ell_r \to \ell_r + \delta_H(\ell_r) . \qquad (5.43)$$

Now, the state $|\psi_{\mathbb{1}}\rangle$ is dominated by a single irrep consisting of a Young diagram with a single row [36], so that there are two non-zero eigenvalues $\ell_1$ and $-\ell_1$ while the remaining $\ell_r$ vanish. Thus, the change in the entanglement can be written as

$$\delta_H\left(\frac{I_n}{2(n-1)}\right) \approx \delta_H(\ell_1) + \sum_{r\neq 1}\delta_H(\ell_r) . \qquad (5.44)$$

Here we have used the fact, from §4.2, that the log of the dimension of the irrep is given, up to a subleading multiplicative logarithmic correction, by the sum of the lengths of rows of the Young diagram. For reasons to be explained shortly, we have separated out the perturbation to the first, non-zero, eigenvalue from the rest.

The two terms in (5.44) correspond to two different types of perturbations: those that change the length of the row and those that add new rows. Semiclassically, each eigenvalue corresponds to the boundary area of a distinct connected component of the subregion [37]. Therefore, the first perturbation in (5.44) can be thought of as changing the geometry of the boundary while preserving topology, while the second perturbation changes the topology of the boundary. These are illustrated in Fig. 7.

Adding new rows increases the dimension of the irrep. This is seen in (5.44) in the fact

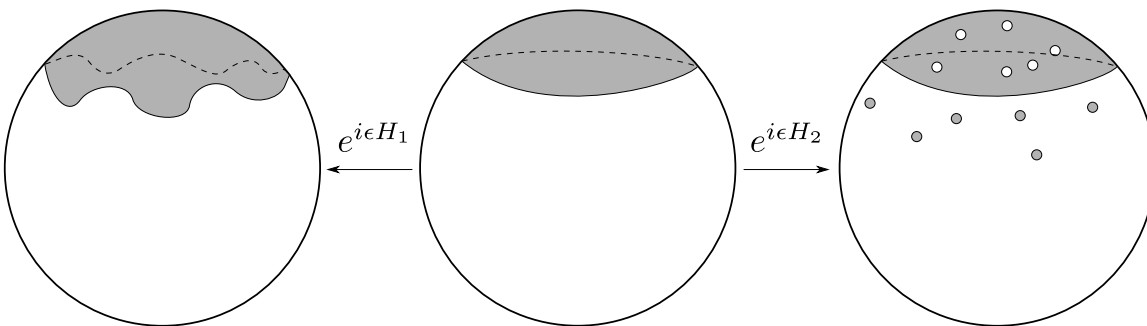

Figure 7: Two different types of perturbative deformations of the cap subregion. $H_1$ deforms the cap by a continuous VPD, by wiggling the boundary. This transformation perturbs the only non-zero eigenvalue of $Q_\Sigma^2$ and contributes to the first term on the right-hand side of (5.44). $H_2$ generates a transformation that takes small disconnected pieces of the cap and moves them into the exterior region. This transformation perturbs the initially zero eigenvalues of $Q_\Sigma^2$ and contributes to the second term on the right-hand side of (5.44).

that $\delta_H(\ell_r) \geq 0$ for $r \neq 1$, because the row length vanishes initially but must be positive. Thus, for the purposes of showing that the spherical cap is a minimum, in the remainder of this subsection we may restrict to a subset of perturbations $H_{\text{VPD}}$ that do not introduce new eigenvalues. These perturbations preserve the fact that $Q_\Sigma$ has only a single pair $\pm \ell_1$ of eigenvalues, which may therefore be extracted from $\operatorname{Tr} Q_\Sigma^2 = 2\ell_1^2$.

We will discuss $\operatorname{Tr} Q_\Sigma^2$ in two steps. A direct analytic connection to the continuum theory can be made for $H_{\text{VPD}}$ perturbations that generate subregions with a weakly curved boundary (not too wiggly). This means that the spherical harmonics involved in the VPD have angular momenta well below the cutoff, $l \ll \Lambda$. We will then return to the general case via numerics. For weakly curved subregions, the following approximations can be made

$$
\begin{aligned}
\langle \operatorname{Tr} Q_\Sigma^2 \rangle &= -4 \left\langle \operatorname{Tr} \left( X_{\text{cl},\Sigma\overline{\Sigma}}^a \Pi_{\overline{\Sigma}\Sigma}^a - \Pi_{\Sigma\overline{\Sigma}}^a X_{\text{cl},\overline{\Sigma}\Sigma}^a \right) \left( X_{\text{cl},\Sigma\overline{\Sigma}}^b \Pi_{\overline{\Sigma}\Sigma}^b - \Pi_{\Sigma\overline{\Sigma}}^b X_{\text{cl},\overline{\Sigma}\Sigma}^b \right) \right\rangle \\
&\approx 8 \left\langle \operatorname{Tr} \left( X_{\text{cl},\overline{\Sigma}\Sigma}^a X_{\text{cl},\Sigma\overline{\Sigma}}^b \Pi_{\overline{\Sigma}\Sigma}^b \Pi_{\Sigma\overline{\Sigma}}^a \right) \right\rangle & (5.45) \\
&\approx \frac{\nu \Lambda^3}{6N} \operatorname{Tr} \left( \Theta_\Sigma X_{\text{cl}}^a \Theta_{\overline{\Sigma}} X_{\text{cl}}^a \Theta_\Sigma \right) = \frac{\nu \Lambda^3}{12N} \operatorname{Tr} \left( \Theta_\Sigma [X_{\text{cl}}^a, [X_{\text{cl}}^a, \Theta_\Sigma]] \right). & (5.46)
\end{aligned}
$$

The second line (5.45) above follows, as argued in [36], from the fact that $X_{\text{cl},\Sigma\overline{\Sigma}}$ is a rank one matrix at the saddle point. The other ordering of the $X$ and $\Pi$ matrices is subleading at large $N$. We may note that the matrix in the trace in the second line is a rank 1 matrix (of operators). This is a signature of the fact that the dominant irrep has an approximately flat Young diagram in the language of Fig. 5. The third line (5.46) above, which is the fuzzy Laplacian $[X_{\text{cl}}^a, [X_{\text{cl}}^a, \cdot]]$ acting on the projection matrix $\Theta_\Sigma$, was obtained in [37] and holds for general weakly curved subregions.

Given the approximation (5.46) for weakly curved subregions, we may use the variation of the projection in (5.42) to evaluate the shift in the trace. To first order we obtain:

$$
\delta_H^{(1)} \operatorname{Tr} (\Theta_\Sigma [X_{\text{cl}}^a, [X_{\text{cl}}^a, \Theta_\Sigma]]) = -2i\epsilon \operatorname{Tr} (H[\Theta, [X_{\text{cl}}^a, [X_{\text{cl}}^a, \Theta]]]). \tag{5.47}
$$

The saddle point equations are then

$$[\Theta, [X_{\text{cl}}^a, [X_{\text{cl}}^a, \Theta]]] = 0. \tag{5.48}$$

It is straightforwardly verified that $X_{\text{cl}}^a$ in (3.2) solve these equations. The spherical cap is therefore indeed a saddle point with respect to these weakly curved matrix VPDs.

To see the positivity of the $H_{\text{VPD}}$ perturbations about the saddle it is convenient to use the Moyal map on (5.46). The Moyal map was discussed briefly in §3.2. A basic fact about the Moyal map is that commutators become derivatives:

$$i[X_{\text{cl}}^a, F] \rightarrow -\nu \epsilon^{abc} x^b \partial_c f(x). \tag{5.49}$$

Therefore, using (3.5) and (5.49),

$$\text{Tr}(\Theta_\Sigma[X_{\text{cl}}^a, [X_{\text{cl}}^a, \Theta_\Sigma]]) = -\text{Tr}\left([X_{\text{cl}}^a, \Theta_\Sigma]^2\right) = \frac{\nu^2 N}{4\pi} \int_{S^2} d\Omega \, (\nabla \times \chi_\Sigma)_\star^2. \tag{5.50}$$

In the large $N$ limit one might hope to replace $\star$ with ordinary multiplication. However, this does not work here because the derivative of the characteristic function $\chi_\Sigma(x)$ is a delta function supported on the boundary of the subregion. Noncommutative delta functions obey the possibly counterintuitive relation $\delta(x) \star \delta(x) \propto L \, \delta(x)$ where $L$ is the size of the region localised to. See [87] or §5.1 of [44] for a discussion, while a detailed computation on the fuzzy sphere can be found in [37]. Thus one finds [37]

$$\text{Tr}(\Theta_\Sigma[X_{\text{cl}}^a, [X_{\text{cl}}^a, \Theta_\Sigma]]) = \frac{\nu^2 N^2}{2\pi^2} |\partial\Sigma|^2. \tag{5.51}$$

It follows that the non-zero eigenvalue $\ell_1 \propto |\partial\Sigma|$, the boundary area.

For weakly curved subregions, then, the perturbation of the eigenvalue $\ell_1$ is given by the change of the boundary area of the subregion. A circle has the minimum boundary area for fixed volume and hence these perturbations necessarily increase the action. It is instructive to verify this directly. Working locally on a plane rather than the sphere, for convenience, we may compute the change in the circumference of a circle under a VPD. A general infinitesimal VPD on the plane $\{x^i\} \mapsto \{\tilde{x}^i\}$ can be written to second order as

$$\tilde{x} = x - \partial_y \left(\epsilon h + \frac{\epsilon^2}{2} \partial_y h \partial_x h\right) + \epsilon^2 \partial_x \left(\frac{1}{2}(\partial_y h)^2\right) + \cdots, \tag{5.52}$$

$$\tilde{y} = y + \partial_x \left(\epsilon h - \frac{\epsilon^2}{2} \partial_y h \partial_x h\right) + \epsilon^2 \partial_y \left(\frac{1}{2}(\partial_x h)^2\right) + \cdots.$$

Here $h(x, y)$ is an arbitrary function. This map preserves the volume element $dx \wedge dy$. The VPD (5.52) is the image of the matrix transformation

$$V X_{\text{cl}}^a V^\dagger \approx X_{\text{cl}}^a + i\epsilon[H, X_{\text{cl}}^a] - \frac{\epsilon^2}{2}[H, [H, X_{\text{cl}}^a]] + \cdots, \tag{5.53}$$

under the Moyal map for a plane, $i[X_{\text{cl}}^a, H] \rightarrow \epsilon^{ab} \partial_b h.$

Under the VPD (5.52), a circle of radius $R$ parametrised by $(x, y) = (R\cos\theta, R\sin\theta)$ is mapped to a curve with length

$$L = \int_0^{2\pi} d\theta \sqrt{(\partial_\theta \tilde{x})^2 + (\partial_\theta \tilde{y})^2} \, . \tag{5.54}$$

We now insert (5.52) into (5.54) and expand to second order in $\epsilon$, utilising the change of variables $\partial_x = \cos(\theta)\partial_r - r^{-1}\sin(\theta)\partial_\theta$ and $\partial_y = \sin(\theta)\partial_r + r^{-1}\cos(\theta)\partial_\theta$. The order $\epsilon$ correction is seen to be a total derivative in $\theta$ and integrates to zero, consistent with the disk being an extremum under VPDs. To second order, after some simple calculus we find

$$L = 2\pi R + \frac{\epsilon^2}{2R^3} \int_0^{2\pi} d\theta \, h(\theta) \left(\partial_\theta^2 + \partial_\theta^4\right) h(\theta) + \cdots , \tag{5.55}$$

where $h(\theta)$ is the pull-back of $h(x, y)$ to the boundary of the disc. This second order correction is furthermore seen to be positive, verifying that the circle indeed has the minimum length for a given fixed area.[16]

It remains to consider perturbations that do not generate new eigenvalues but are outside the regime of validity of (5.46). These are certain transformations with angular momentum above the cutoff $\Lambda$. We investigate such modes in Appendix D by considering perturbations numerically at finite $N$. The results of these investigations are shown in Figure 8 and support the conclusion that all of the modes are positive. However, we have not excluded

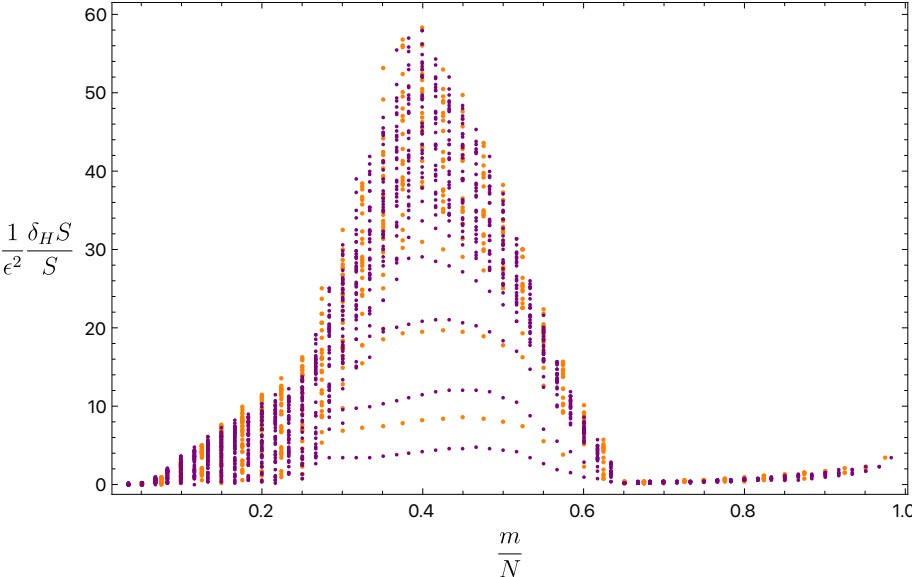

Figure 8: Relative change of the entropy $S$ under many matrix transformations $H$, labelled by azimuthal angular momentum $m$. Details are in Appendix D. Each dot is a different mode. Orange dots have $N = 40, M = 16, \Lambda = 10$ and purple dots have $N = 60, M = 24, \Lambda = 15$. The agreement between the orange and purple data points suggests approximate convergence to a large $N$ behaviour. All changes to the entropy are positive.

---

[16]If we set $h(\theta) = \sum_{m=0}^N [a_m \cos(m\theta) + b_m \sin(m\theta)]$ , then $\Delta L = \frac{\pi}{2R^3} \sum_{m=0}^N m^2(m^2 - 1)\left(a_m^2 + b_m^2\right) \geq 0$. Here we also see the $m = \pm 1$ zero modes from moving the subregion around.

the possibility that there is a negative mode which is not picked out by the perturbations we have considered or which emerges in a different parameter regime from the one we can access numerically.

One interesting point that is visible in Fig. 8 is that the change in entropy is suppressed for large values of the azimuthal angular momentum $m$. This may suggest that, at least close to weakly curved subregions, the cutoff $\Lambda$ in the state also regulates the area preserving transformations that contribute to the frame average.

## 5.4 The one-loop contribution

In this subsection we will estimate the magnitude of the one-loop contribution to the integral over frames. This magnitude is largely determined by the large number of directions in the flag manifold that move away from the saddle point, of order $M(N-M)/p^2$. The non-trivial work in this subsection will be to estimate a logarithmic factor multiplying this quantity.

The change of the action due to a fluctuation in the shape of a subregion was given in (5.44) as the sum of the changes of the Young diagram row lengths, $\delta_H(\ell_r)$. We will obtain the row lengths squared, $\ell_r^2$, as the eigenvalues of the matrix of expectation values $\langle Q_\Sigma^2 \rangle$. In general these are not equal to the expectation values of the eigenvalues of the matrix operator $Q_\Sigma^2$. However, repeating the argument around (4.5), without taking the trace, shows that at large $N$ and $\nu$ the eigenvalues of the expectation value of the matrix are the same as the expectation values of the eigenvalues.

It will be sufficient to work to order of magnitude, thus we may write

$$\langle Q_\Sigma^2 \rangle \sim X_{\mathrm{cl},\overline{\Sigma}\Sigma} X_{\mathrm{cl},\Sigma\overline{\Sigma}} \left\langle \Pi_{\overline{\Sigma}\Sigma} \Pi_{\Sigma\overline{\Sigma}} \right\rangle . \tag{5.56}$$

On the spherical cap subregion $X_{\mathrm{cl},\overline{\Sigma}\Sigma}$ is a rank one matrix with a single entry of order $\nu N$ while $\langle \Pi_{\overline{\Sigma}\Sigma} \Pi_{\Sigma\overline{\Sigma}} \rangle$ has rank $\Lambda$ with singular values of order $\nu\Lambda^3/N$. These latter statements follow from the fact that in the quantum state (3.10), see [36] for more details,

$$\left\langle \Pi_{\overline{\Sigma}\Sigma} \Pi_{\Sigma\overline{\Sigma}} \right\rangle \sim \sum_{l=1}^{\Lambda} l\nu\omega_l |Y_{lm}|^2 . \tag{5.57}$$

The factor of $l$ is from the degeneracy of the $m$ quantum number and $\omega_l \sim l$. The normal modes are normalised so that $\mathrm{Tr}\, |Y_{lm}|^2 = 1$, and therefore the singular values of $|Y_{lm}|^2$ are order $1/N$. Finally, the typical $Y_{lm}$ that contributes is a polynomial of degree $\Lambda$ in $X_{\mathrm{cl}}$ and therefore has rank $\Lambda$. The matrix (5.56) is thus a rank one matrix multiplied by a rank $\Lambda$ matrix. It therefore has a single non-zero eigenvalue, $\ell_1^2 \sim \nu^3 \Lambda^3 N$. This single non-zero row length of the cap subregion is responsible for the saddle point action (5.39).

We may now turn to perturbations of the spherical cap. In (5.56) we perturb $X_{\mathrm{cl}}$ using (5.53), and similarly $\Pi$. The spherical cap is invariant under azimuthal rotations, while the perturbations break this symmetry. Therefore (5.56) can only change at $\mathcal{O}(\epsilon^2)$. This means that either two of the $A \equiv X_{\mathrm{cl}}, \Pi$ terms in (5.56) must be replaced by $A \to \epsilon[H, A]$ or one of the terms must be replaced by $A \to \epsilon^2[H, [H, A]]$. To estimate the effects of these substitutions note that generically $H \propto X_{\mathrm{cl}}^N$, as the angular momentum of the wiggle modes

is not cut off by $\Lambda$, and therefore $[X_{\text{cl}}, H] \sim \nu N H \sim X_{\text{cl}} H$. These estimates are for the size of generic non-zero entries of the matrices. Similarly, because $\Pi \propto X_{\text{cl}}^{\Lambda}$ one finds $[\Pi, H] \sim \Lambda \Pi H$. The second order changes are: $[H, [H, \Pi]] \sim \nu \Lambda N \Pi H^2$ and $[H, [H, X_{\text{cl}}]] \sim \nu N X_{\text{cl}} H^2$. In addition to the magnitude of the entries, however, we must also keep track of the rank of the matrix.

We may first note that if either of the rank one $X_{\text{cl}, \overline{\Sigma}\Sigma}$ terms remain in (5.56) after perturbing, then $\langle Q_{\Sigma}^2 \rangle$ remains a rank one matrix, and hence such perturbations change the eigenvalue $\ell_1^2$ but do not introduce new eigenvalues. The dominant perturbation here is the one that involves a single $[H, [H, \Pi]]$, leading to

$$\delta_H(\ell_1) = \frac{\delta_H(\ell_1^2)}{2\ell_1} \sim (\nu\Lambda)^{5/2} N^{3/2} \epsilon^2 H^2 \,. \tag{5.58}$$

In order to introduce new eigenvalues, both of the $X_{\text{cl}, \overline{\Sigma}\Sigma}$ terms in (5.56) must be perturbed. These terms become full rank upon perturbation, but the fact that $\langle \Pi_{\overline{\Sigma}\Sigma} \Pi_{\Sigma\overline{\Sigma}} \rangle$ is rank $\Lambda$ means that only $\Lambda$ new eigenvalues of $\langle Q_{\Sigma}^2 \rangle$ are generated in this way. These have

$$\delta_H(\ell_r) = \sqrt{\delta_H(\ell_r^2)} \sim (\nu\Lambda)^{3/2} N^{1/2} |\epsilon| \sqrt{H^2} \qquad \text{for } 1 < r \le \Lambda \,. \tag{5.59}$$

To obtain the remaining eigenvalues it is necessary to perturb, in addition, both of the $\Pi$ factors in (5.56). This leads to an $\mathcal{O}(\epsilon^4)$ shift in $\langle Q_{\Sigma}^2 \rangle$ and hence

$$\delta_H(\ell_r) = \sqrt{\delta_H(\ell_r^2)} \sim \nu^{3/2} \Lambda^{5/2} N^{1/2} \epsilon^2 \sqrt{H^4} \qquad \text{for } \Lambda < r \le \frac{M}{2} \,. \tag{5.60}$$

The difference between the two sets of modes (5.59) and (5.60) may be related to the change in behaviour visible in Fig. 8 at large $m$.

Putting (5.58), (5.59) and (5.60) together, and setting $\epsilon = 1$, the fluctuations about the saddle point contribute, schematically, the following to the frame average

$$Z_{\text{loop}} \sim \frac{1}{\text{vol}_H \, \mathsf{F}} \int \mathrm{d}H e^{-\alpha H^2 - \beta\Lambda\sqrt{H^2} - \gamma\left(\frac{M}{2} - \Lambda\right)\sqrt{H^4}} \,. \tag{5.61}$$

Here $\int \mathrm{d}H$ is a $2M(N-M)/p^2$ dimensional integral over the independent directions in the flag manifold and $\alpha, \beta, \gamma$ are the prefactors in (5.58), (5.59) and (5.60). The normalisation factor is the volume of the flag manifold with the unnormalised Haar measure [88, 89]

$$\text{vol}_H \, \mathsf{F} \approx N^{-\frac{M(N-M)}{p^2}} \,, \tag{5.62}$$

at large $N$. This factor is necessary because the integral over frames is performed with the unit normalised Haar measure, whereas the measure $dH$ in (5.61), for Hermitian matrices such that $U = e^{iH}$, corresponds to the unnormalised Haar measure (i.e. the measure induced on the unitary matrices from flat space). We see that (5.61) is not quite a conventional Gaussian one-loop integral, due to the non-analytic dependence on $H$. This occurs because some of the row lengths vanish in the large $N$ saddle point and are constrained to be positive.

We can upper bound (5.61) as

$$Z_{\text{loop}} \lesssim \frac{1}{\text{vol}_H \, \mathsf{F}} \int dH e^{-\gamma\left(\frac{M}{2}-\Lambda\right)\sqrt{H^4}} \sim e^{-\frac{M(N-M)}{2p^2}\log N} \, . \tag{5.63}$$

The final expression comes from (5.62) combined with $2M(N-M)/p^2$ integrals each of order $N^{-3/4}$. This last fact follows from noting that $\gamma M \sim N^{3/2}$ and then rescaling variables. There may be factors of $\nu$ and $\Lambda$ inside the logarithm in (5.63) that we are not attempting to capture. The bound in (5.63) has been quoted in (5.10) above. We have now assembled all of the results quoted in the summary (5.10).

The most important fact about the one-loop contribution (5.63) is that it is sensitive to the coarse-graining parameter, $p$. This is in contrast to the saddle point value, (5.39), or the scaling of a generic action, (5.38). The fact that $Z_{\text{loop}}$ is parametrically small indicates that the one-loop region is a tiny portion of the total flag manifold. That is to say, the integral is very sharply peaked at the saddle point.

Finally, we are now in a position to estimate $\delta(0)$ and verify that, as claimed, it is subleading. As explained below (2.43), $\delta(0) \sim (\Delta V)^{-1}$ where $\Delta V$ is the uncertainty in the $|V\rangle$ basis states. The uncertainty in $V$ is related to the uncertainty in the quadratic Casimir. Consider configurations close to the saddle point, so that

$$\Delta_{C_2} \sim \frac{d^2 C_2}{dV^2}(\Delta V)^2 \, . \tag{5.64}$$

Close to the saddle $\Delta_{C_2} \sim C_2/\Lambda^{1/2} \sim \nu^3 \Lambda^{5/2} N$ [36]. From the results in this subsection: $\frac{d^2 C_2}{dV^2} \sim \frac{d^2}{d(\epsilon H)^2} \sum_r \ell_r^2 \sim N^a$ for a positive, order one power $a \geq 1$. Therefore

$$\log \delta(0) = -\log \Delta V \sim \log N \, , \tag{5.65}$$

and $\delta(0)$ is seen to produce a subleading contribution to the entropy.

# 6   Discussion

We have shown how a reference-frame-averaged partition of matrix degrees of freedom can exhibit a Ryu-Takayanagi-like minimal area formula for the entanglement. The areas emerge from the matrices through representation-theoretic combinatorics, while the minimisation emerges from the decoherence of the different frames and the subsequent saddle point computation of a 'classical' average over frames. Perhaps the most intriguing aspect of our computation is that for geometric subregions to dominate we found it necessary to coarse-grain the frame average. This fact introduced an element of arbitrariness in the construction, in the specific way we integrate over reference frames. While we have focussed on one particularly tractable manner of coarse-graining in this paper, we believe our results are robust to any coarse-graining that sufficiently reduces the number of independent 'wiggle modes.'

There are many open questions. The most burning ones involve connecting our framework more closely to the established semiclassical understanding of entanglement in theories of gravity and, likely related, extending the methods we have developed to more complicated

theories of large $N$ matrices.

## Rényi entropies and replica symmetry breaking

At the most technical level, there are two extensions of our calculations to include subleading effects that would deepen the analogy with semiclassical gravity. In §2.7 we explained that our calculation is at the level of accuracy of Fursaev's replica trick [74] for gravity: it works when we are only interested in the von Neumann entropy. Computing the higher Rényi entropies accurately, even to leading order, will require an understanding of the fluctuations in the irrep dimension $d_\mu$. Secondly, in §2.7 and Appendix A we have explained that in order to capture effects associated to replica symmetry breaking it will be necessary to quantify the cross terms between different QRFs.

## Beyond semiclassical matrices

We have been able to perform explicit computations because the large $N$ matrices in our model are in a semiclassical quantum state. This fact furthermore underpins the Moyal correspondence between matrices and geometry, which enabled us to move easily between the microscopic $U(N)$ gauge symmetry and the emergent symmetry of volume preserving diffeomorphisms. In richer theories of MQM — including maximally supersymmetric theories such as BFSS, BMN and $\mathcal{N} = 4$ super-Yang Mills (SYM) — the states of interest are likely to be significantly more quantum. The emergence of geometries and diffeomorphisms from the matrices in these theories is not understood in detail. Nonetheless, in all cases there remains a clear correspondence between the matrices and emergent directions of space. It is therefore plausible that matrix partitions similar to those that we have considered will again have geometric interpretations. In the case of $\mathcal{N} = 4$ SYM these should be related to geometric partitions of the internal space, cf. [90–94]. Other discussions of matrix entanglement in the context of D-brane theories include [95, 96].

The correspondence between $U(N)$ and volume preserving diffeomorphisms is best understood in two spatial dimensions. This correspondence may also be natural in higher dimensions too, insofar as the matrix trace, which is preserved by $U(N)$, is related to the emergent volume. In this regard it is interesting to note[17] that volume preserving diffeomorphisms are the gauge symmetry of higher dimensional gravity coupled to a dilaton field. Shifts of the dilaton can be used to compensate for the local volume changes of the metric. Said differently, the physical dilaton field ungauges the Weyl rescaling induced by diffeomorphisms.

## Diffeomorphisms and extremisation

Volume preserving diffeomorphisms play a central role in our minimisation formula. Is this a more general lesson for gravitational entanglement? Diffeomorphisms are known to be important in the calculation of entanglement in 2d or 3d gravity using the BF theory or Chern-Simons descriptions [97–101, 13]. In those calculations, one performs a standard

---

[17]We thank Diego Hofman for making this point.

replica trick in the topological QFT, and the entropy is given by the log of the dimension of an irrep of an internal gauge group. However, the gauge transformations in those theories are equivalent (on-shell) to spacetime diffeomorphisms, and these diffeomorphisms include those which move the position of the cut relative to the boundary, i.e. wiggle modes. Diffeomorphism edge modes, as well as wiggle modes, have been widely discussed from various other perspectives in e.g. [49, 102–107].

The structure we have found is similar in broad outline: We define a partition that includes wiggle modes, and find that the integral of $\operatorname{tr} \rho^n$ localises to an extremal surface. In our case time reparametrisations are not gauged and therefore extremality is the same as minimality. Our results may possibly, therefore, be taken in conjunction with those we have just described to point towards an important role for diffeomorphism invariance in extremisation formulas for gravitational entropy.

**Subregions as subsectors, and random tensor networks**

In our framework a Ryu-Takayanagi-like minimisation arises from a saddle point evaluation of traces of a reduced density matrix with many approximately orthogonal subsectors. It is interesting that this structure also arises in other models or gravitational entanglement, notably random tensor networks [108]. In random tensor networks, there is classical randomness that, thanks to properties of the Haar integral, becomes similar to our 'classical' uncertainty in the partition. It would be interesting to understand if the two classes of models could be treated in a unified way, and whether a similar structure exists in higher-dimensional theories. Some connections between MQM and tensor networks have recently been made in [109].

**Discontinuous diffeomorphisms and regularisation**

Our frame averaged matrix partition potentially receives contributions from highly disconnected and jagged subregions that have to be regulated in order to obtain a geometric answer for the entanglement. The contribution of disconnected regions to the average is consistent with the standard Ryu-Takayanagi formula [20]. It is plausible that the need to regulate the space of frames in the minimisation is a lesson that holds beyond the specific model we have considered — presumably, sufficiently jagged regions generically probe physics at scales where the gravitational effective field theory is not valid. A local partition should only be defined within the regime of validity of the notion of locality itself.

The need to regularise has appeared recently in the literature on edge modes in gravitational phase space [106] and also in work on non-perturbative corrections to the gravitational inner product [7–9].[18] Perhaps such a regularisation is also needed to regulate the divergence of the entropy due to the non-compactness of the gauge group found in [97, 62, 99, 100]. On the other hand, it may be that richer theories of quantum matrices are able to dynamically regulate themselves. For example, through cancellations induced by supersymmetric partners of the bosonic matrices or by having a larger geometric entanglement that is able to overcome the volume of the space of wiggle modes.

---

[18]We thank Adam Levine for drawing our attention to this connection.

**Towards a holography of quantum reference frames?**

The Ryu-Takayanagi formula in the AdS/CFT correspondence is anchored to a gauge-invariant boundary subregion. In spacetimes where there is no boundary, such as cosmologies, such a sturdy anchor does not exist. In such settings one is forced into a relational approach of the kind we have developed [50]. Interestingly, in the case where there is no boundary, the definition of entropy in [110] has been re-cast into the language of QRFs [111, 112]. Relational descriptions of geometric partitions have also been given in [113–115]. It would be extremely interesting to understand if our framework offers a microscopic counterpart to these developments.

# Acknowledgements

We thank Sumit Das, Elliot Gesteau, Diego Hofman, Philipp Höhn, Adam Levine, Pratik Rath, Douglas Stanford and Shreya Vardhan for discussions and comments. We thank Phillipp Höhn for detailed comments on a draft.

This work has been partially supported by STFC consolidated grant ST/T000694/1. RMS is supported by the Isaac Newton Trust grant "Quantum Cosmology and Emergent Time" and the (United States) Air Force Office of Scientific Research (AFOSR) grant "Tensor Networks and Holographic Spacetime". They also thank UC Berkeley for hospitality. SAH and JRF are partially supported by Simons Investigator award #620869. AF is partially supported by the NSF GRFP under grant no. DGE-165-651.

# A   The replica symmetry assumption

In going from (2.41) to (2.42) in the main text we have assumed that the partial trace ensures that the reduced density matrix does not mix different QRFs. This assumption can be thought of as a Hamiltonian version of replica symmetry: the form (2.42) of the density matrix ensures that $\operatorname{tr}\rho_{\text{in}}^n$ in (2.43) is a sum of terms where, for each $V$ (i.e. subregion) separately, the same function of $V$ appears for each of the $n$ factors of the reduced density matrix. Replica symmetry for all $V$ is stronger than the assumption used in derivations of the Ryu-Takayanagi formula [17], since those derivations only require that the saddle point of the replica path integral has replica symmetry. See §2.7 and §6 for further discussion.

Even for us, the replica symmetry assumptions on and off the saddle point, which we call on-shell and off-shell replica symmetry, have different justifications and physical content. We will discuss off-shell replica symmetry breaking first. We show that for off-shell replica symmetry breaking to have any effect, an exponentially large number of non-generic solutions to an overdetermined set of polynomial equations is required. Numerical investigations suggest that in fact no such solutions exist. On-shell replica symmetry is in general an assumption that can fail in interesting physical situations [71, 76, 77]. We show that for general MQM on-shell replica symmetry breaking would indicate a glassy degeneracy of minima. We furthermore establish that replica symmetry does hold for the cap subregion of the fuzzy sphere.

## Off-shell replica symmetry

In (2.39) the state is dominated by one classical configuration. Within this approximation, the presence of a mixed term in the reduced density matrix, so that a pair $V \neq V'$ have $\text{tr}_{\text{out}}\left[|\psi_V\rangle\langle\psi_{V'}|\right] \neq 0$, is equivalent to the statement that

$$\exists\, U, \widetilde{U} \quad \text{s.t.} \quad \left(U\widetilde{U}V'X_{\text{cl}}^a V'^\dagger \widetilde{U}^\dagger U^\dagger\right)_{\Sigma\overline{\Sigma},\overline{\Sigma}\overline{\Sigma}} = \left(VX_{\text{cl}}^a V^\dagger\right)_{\Sigma\overline{\Sigma},\overline{\Sigma}\overline{\Sigma}}. \tag{A.1}$$

That is to say that, on the 'out' blocks, the classical matrix transformed by $V$ and by $V'$, respectively, can be related by a $U(M) \times U(N-M)$ transformation. The existence of such transformations forms an equivalence relation among elements of $\mathsf{F}$. Denote the equivalence class of $V$ by $[V]$, and the saddle point of the integral as $V = V_0$.

Let us ask for the number of $U$, $\widetilde{U}$, and $V'$ satisfying (A.1), given $V$. The space of matrices $U\widetilde{U}V'$ is just $U(N)$, whose dimensionality is $N^2$. So (A.1) is, given $V$, $d(N^2 - M^2)$ equations for $N^2$ variables. The dimensionality of the solution space is, then, $N^2 - d(N^2 - M^2) = dM^2 - (d-1)N^2$. This is negative for $0 < M/N < \sqrt{(d-1)/d}$, and so solutions are non-generic in this range of $M/N$. For the fuzzy sphere model that we considered in detail, $d = 3$ and hence the upper bound here is $\sqrt{2/3} \approx .82$. Including $p > 1$ in the analysis does not change the result significantly. So, for a large range of $M$ we expect there to be at most isolated, non-generic solutions to (A.1) for generic $V$. Outside of this range equivalence classes might span a submanifold of $\mathsf{F}$. We will not consider this latter situation further.

Now let us investigate the corrected Rényi entropies after taking into account possible equivalence classes with more than one element. We will assume that $M$ is in the range, given above, where any solutions to (A.1) are isolated and $[V]$ is a discrete set. If there is a pair $V \neq V'$ satisfying (A.1) then, by definition, the outside blocks of $VX_{\text{cl}}^a V^\dagger$ and $V'X_{\text{cl}}^a V'^\dagger$ are in the same $U(M) \times U(N-M)$ orbit. This means that the 'out' components of both $|\psi_V\rangle$ and $|\psi_{V'}\rangle$ live in the same $U(M)_{\overline{\Sigma}}$ irrep, since each of these live in a single irrep. More generally, the distribution over irreps is the same in $|\psi_V\rangle$ and $|\psi_{V'}\rangle$. By Gauss's law, the 'in' components must be in the same irrep of $U(M)_\Sigma$. Thus, we find

$$\rho_{\text{in}} \approx \oint \mathrm{d}[V]\, p_{[V]}\rho_{[V],\text{in}} \otimes \frac{\mathbb{1}_{\mu_{[V]}}}{d_{\mu_{[V]}}}, \qquad \rho_{[V],\text{in}} \equiv \frac{1}{p_{[V]}} \sum_{V_1,V_2 \in [V]} \rho_{\text{in}}(V_1,V_2)\,|V_1\rangle\langle V_2| \tag{A.2}$$

Here, $\text{tr}\,\rho_{[V],\text{in}} = 1$. The assumption of off-shell replica symmetry sets the off-diagonal terms in $\rho_{[V],\text{in}}$ to zero; let us call the density matrix obtained by setting these cross-terms to zero $\sigma_{[V],\text{in}}$. $\text{tr}\,\sigma_{[V],\text{in}}^n \leq \text{tr}\,\rho_{[V],\text{in}}^n$ when $n > 1$; this can be proved by using the convexity of the function $\text{tr}(\cdot)^n$. Thus, the assumption of replica symmetry reduces the contribution of $[V]$ to $\text{tr}\,\rho_{\text{in}}^n$. This means that unless we can argue against the presence of these nontrivial equivalence classes, we have potentially underestimated the contribution of generic elements of the flag manifold to the frame average integral.

We have seen in the main text that generic elements of the flag manifold are exponentially suppressed relative to the saddle point. The dominance of the saddle can therefore only be affected if $\text{tr}\,\sigma_{[V],\text{in}}^n$ is exponentially smaller than $\text{tr}\,\rho_{[V],\text{in}}^n$. Since the $n$-purity of a density matrix on a $D$-dimensional Hilbert space ranges from $D^{1-n}$ to 1, an error large enough to

affect the dominance of the saddle point can only occur if $\|[V]\|$ is exponentially large. This would require an exponentially big conspiracy, that is, an exponentially large number of isolated solutions to the overdetermined quadratic equations in (A.1). We have performed numerical explorations of (A.1) that suggest that there are in fact no nontrivial solutions for $V'$ at generic $V$, as one would naïvely expect. We will therefore assume that this is the case, and hence the dominance of the saddle is not affected.

Beyond the semiclassical limit, as we have noted in the main text, there is a variance in the irrep for fixed $V$. This variance makes (A.1) only approximately true and allows for overlaps of $V$ and $V'$ not lying in the same equivalence class. For the fuzzy sphere state, our results above imply that these effects are subleading in the semiclassical expansion: The uncertainty in the irrep was shown to be small in §5.2, and in §5.4 this was shown to lead to a small width $\Delta V$. A small variance will also be present if we use a non-ideal QRFs where $V$ is not completely fixed.

## On-shell replica symmetry

Cross terms can also affect the saddle point contribution of $V = V_0$ to the integral. We will first make a general comment and then a stronger statement for the case of the cap subregion of the fuzzy sphere.

The general comment is that if $V_0$ is a global minimum then all the elements of $[V_0]$ must also be degenerate global minima. This follows, at leading order, from the fact that all elements of $[V_0]$ are in the same $U(M)_\Sigma$ irrep and therefore have the same dimension as $d_{\mu_{V_0}}$. A nontrivial effect would therefore require an exponentially large (in $N$) number of degenerate minima. This would be suggestive of glassy behaviour which, famously, is indeed associated to replica symmetry breaking. Our argument in the main text therefore assumes the absence of such physics. We may note that a small number of cross terms between minima can exist in general and also be physically interesting [71, 76, 77].

For the cap subregion of the fuzzy sphere, we can explicitly show that there are no cross-terms. Let us go back to (A.1) and set $V = \mathbb{1}$ for the cap subregion. Consider the $a = 3$ equation. Since $X_{\mathrm{cl}}^3$ is diagonal, the condition becomes

$$\left( \mathsf{U} X_{\mathrm{cl}}^3 \mathsf{U}^\dagger \right)_{\Sigma \overline{\Sigma}} = 0 \, , \qquad \left( \mathsf{U} X_{\mathrm{cl}}^3 \mathsf{U}^\dagger \right)_{\overline{\Sigma}\,\overline{\Sigma}} = \left( X_{\mathrm{cl}}^3 \right)_{\overline{\Sigma}\,\overline{\Sigma}} \, , \tag{A.3}$$

where $\mathsf{U} = U \widetilde{U} V'$. We will now show that, because $X_{\mathrm{cl}}^3$ has no coincident eigenvalues in this case, the only $\mathsf{U}$ that satisfy (A.3) are in $U(M) \times U(N - M)$ and therefore $V' = \mathbb{1}$. This proves on-shell replica symmetry in our setup.

The argument is as follows. Since the first equation in (A.3) implies that $\mathsf{U} X_{\mathrm{cl}}^3 \mathsf{U}^\dagger$ is a block diagonal matrix, it can be diagonalised by a $U(M) \times U(N - M)$ transformation. Since $X_{\mathrm{cl}}^3$ is diagonal, the second equation in (A.3) implies that $\mathsf{U} X_{\mathrm{cl}}^3 \mathsf{U}^\dagger$ can be diagonalised by a $U(M)$ transformation, let us call it $U'$. Furthermore, $\mathsf{U} X_{\mathrm{cl}}^3 \mathsf{U}^\dagger$ and $X_{\mathrm{cl}}^3$ have the same eigenvalues, as they are related by a unitary transformation, and so we must have $U' \mathsf{U} X_{\mathrm{cl}}^3 \mathsf{U}^\dagger U'^\dagger = X_{\mathrm{cl}}^3$. Note that any re-ordering of the eigenvalues can be achieved by $U'$ because (A.3) implies that $\mathsf{U} X_{\mathrm{cl}}^3 \mathsf{U}^\dagger$ agrees with $X_{\mathrm{cl}}^3$ on the outer $\overline{\Sigma}\,\overline{\Sigma}$ block, and hence $U(M)$ is sufficient to re-order any eigenvalues that got moved around by $\mathsf{U}$. We may therefore

conclude that $U'\mathsf{U}$ commutes with $X^3_{\text{cl}}$. Since $X^3_{\text{cl}}$ has no coincident eigenvalues, it follows that $U'\mathsf{U} \in U(1)^N \in U(M)$. Thus $\mathsf{U} \in U(M) \times U(N - M)$, as claimed.

# B   Normal modes

This Appendix collects some expressions concerning the normal modes for perturbations of the classical fuzzy sphere configuration with the Hamiltonian (3.8).

In order for the modes to be Hermitian matrices one must take linear combinations of the $m$ and $-m$ modes. Thus the modes are not individually eigenfunctions of $m$. We will instead label the modes by two families with $t = \{1, i\}$, one family will have $m \geq 0$ and the other $m > 0$. Furthermore, each family can have eigenvalues

$$\omega_+(l) = l + 1, \qquad \omega_-(l) = -l. \tag{B.1}$$

Thus for a given $l$ and positive $m$ there will be up to four modes altogether.

The matrix spherical harmonics $\hat{Y}_{lm}$ are constructed as described in [31], in the basis where the classical matrices take the form (3.2). We now write down, completely explicitly in terms of the spherical harmonics, the modes $Y^a_{lmt\pm}$. These modes obey the equations of motion given in [31]. Firstly, and recall that $t = \{1, i\}$,

$$Y^3_{lmt\pm} = \frac{\alpha_\pm}{\sqrt{N}} \left( t\hat{Y}_{lm} + t^*\hat{Y}^\dagger_{lm} \right), \tag{B.2}$$

where

$$\alpha_+^2 = \frac{(l + 1)^2 - m^2}{(2l + 2)(2l + 1)}, \qquad \alpha_-^2 = \frac{l^2 - m^2}{(2l + 1)2l}. \tag{B.3}$$

Secondly,

$$Y^+_{lmt+} = \frac{1}{\sqrt{N}} \left( \frac{\sqrt{(l - m + 1)(l - m)}}{\sqrt{(2l + 2)(2l + 1)}} t\hat{Y}_{lm+1} - \frac{\sqrt{(l + m + 1)(l + m)}}{\sqrt{(2l + 2)(2l + 1)}} t^*\hat{Y}^\dagger_{lm-1} \right), \tag{B.4}$$

and

$$Y^+_{lmt-} = \frac{1}{\sqrt{N}} \left( -\frac{\sqrt{(l + m + 1)(l + m)}}{\sqrt{(2l + 1)2l}} t\hat{Y}_{lm+1} + \frac{\sqrt{(l - m + 1)(l - m)}}{\sqrt{(2l + 1)2l}} t^*\hat{Y}^\dagger_{lm-1} \right), \tag{B.5}$$

so that

$$Y^1_{lmt\pm} = \frac{1}{2} \left( Y^+_{lmt\pm} + Y^{+\dagger}_{lmt\pm} \right), \qquad Y^2_{lmt\pm} = \frac{1}{2i} \left( Y^+_{lmt\pm} - Y^{+\dagger}_{lmt\pm} \right). \tag{B.6}$$

From the above formulae ones sees that the $\omega_+$ modes have $m = 0, \dots, l + 1$ while the $\omega_-$ modes have $m = 0, \dots, l - 1$. In particular, the values of $m = l$ and $m = l + 1$ are allowed in (B.4) because the coefficient of the $\hat{Y}_{lm+1}$ term vanishes for these cases. Furthermore, the coefficient $\alpha_+$ in (B.3) vanishes when $m = l + 1$.

The modes constructed above are orthonormal in the sense that

$$\sum_{a=1}^{3} \mathrm{Tr}\left[Y_{lmts}^{a\dagger} Y_{l'm't's'}^{a}\right] = \delta_{ll'} \delta_{mm'} \delta_{tt'} \delta_{ss'} . \tag{B.7}$$

The $Y_{lm}^{3}$ modes above have non-vanishing off-diagonal components and are therefore not in the same gauge as $X_{\mathrm{cl}}^{a}$. However, the modes can be rotated back into the original gauge by adding pure gauge zero modes to the $Y_{lm}^{a}$ that remove off-diagonal components. Doing this also requires us to modify the Hamiltonian by a BRST-exact term, in order for the rotated modes to be eigenstates. It is also fine to perform perturbative computations in a slightly different gauge given by the unrotated modes above.

## C   Coarse-grained second Casimir

Recall that in (5.30) we have decomposed the matrix as

$$M = \sum_{\alpha} M_{\alpha} \otimes \mathfrak{b}_{\alpha} . \tag{C.1}$$

We take the basis where each $\mathfrak{b}_{\alpha}$ has only one non-zero entry, equal to 1. As an example, for $p = 2$ we may take

$$\mathfrak{b}_{1} = \begin{bmatrix} 1 & 0 \\ 0 & 0 \end{bmatrix}, \quad \mathfrak{b}_{2} = \begin{bmatrix} 0 & 1 \\ 0 & 0 \end{bmatrix}, \quad \mathfrak{b}_{3} = \begin{bmatrix} 0 & 0 \\ 0 & 1 \end{bmatrix}, \quad \mathfrak{b}_{4} = \begin{bmatrix} 0 & 0 \\ 1 & 0 \end{bmatrix}. \tag{C.2}$$

The projection matrices take the simple form

$$\Theta_{M} = \Theta_{M'} \otimes \mathbb{1}_{p}. \tag{C.3}$$

The Haar integration over $U(N')$ obeys the exact same contractions as before, with an additional sum over the $\alpha_i$ labelling the $\mathrm{Mat}(\mathbb{C}^p)$ basis elements. Focussing on the dominant $\mathrm{Tr}(Y^2)\,\mathrm{Tr}(X^2)$ term:

$$\begin{aligned} \overline{\langle \mathrm{Tr}\, Q^2 \rangle} &\approx 4\nu^3 \sum_{lm} |\omega_{lm}| \frac{M'^2(N'-M')}{N'^4} \\ &\quad \times \sum_{\alpha_i} \mathrm{Tr}\left[Y_{lm,\alpha_1}^{a} Y_{lm,\alpha_2}^{b}\right] \mathrm{Tr}\left[X_{\mathrm{cl},\alpha_3}^{a} X_{\mathrm{cl},\alpha_4}^{b}\right] \mathrm{Tr}[\mathfrak{b}_{\alpha_1} \mathfrak{b}_{\alpha_2} \mathfrak{b}_{\alpha_3} \mathfrak{b}_{\alpha_4}]. \end{aligned} \tag{C.4}$$

The tensor decomposition (C.1) amounts to breaking the matrix up into $p \times p$ blocks, cf. (5.3). Recall that the $X_{\mathrm{cl}}$ are supported on the diagonal or one step off the diagonal. This means all except a fraction $1/p$ of the off-diagonal entries of $X_{\mathrm{cl}}$ end up in the diagonal $p \times p$ blocks. Therefore $X_{\mathrm{cl},\alpha_i}$ will be a diagonal matrix up to corrections suppressed by $\mathcal{O}(1/p)$. The $Y_{lm}$ have entries along two bands that are $\pm m$ steps away from the diagonal, see Appendix A of [36]. With a cutoff $\Lambda$ on $l$, and hence on $m$, this means that now all except a fraction $\Lambda/p$ of the off-diagonal entries end up in the diagonal $p \times p$ blocks. Therefore $Y_{jm,\alpha_i}^{a}$ is a

diagonal matrix up to corrections suppressed by $\mathcal{O}(\Lambda/p)$, taking $\Lambda \ll p$. Using these facts, and taking into account that $X^a_{\mathrm{cl},\alpha_i}$ and $Y^a_{lm,\alpha_i}$ are now $N' \times N'$ dimensional matrices, we may estimate that to leading order

$$\mathrm{Tr}\left[Y^a_{lm,\alpha_1} Y^b_{lm,\alpha_2}\right] = \mathcal{O}(N'/N) = \mathcal{O}(1/p) , \qquad \mathrm{Tr}\left[X^a_{\mathrm{cl},\alpha_3} X^b_{\mathrm{cl},\alpha_4}\right] = \mathcal{O}(N'N^2) . \qquad (C.5)$$

In fact most of these traces are zero. As we have just recalled, both the $X^a_{\mathrm{cl}}$ and $Y^a_{lm}$ matrices are band-diagonal, which means that the traces in (C.5) are non-zero for basis element labels $\alpha_i$ taking only $p$ of the possible $p^2$ values.

From the above discussion we obtain

$$\overline{\langle \mathrm{Tr}\, Q^2 \rangle} \sim \nu^3 \Lambda^3 N' N \sum_{\alpha_i} \mathrm{Tr}[\mathfrak{b}_{\alpha_1} \mathfrak{b}_{\alpha_2} \mathfrak{b}_{\alpha_3} \mathfrak{b}_{\alpha_4}] , \qquad (C.6)$$

where the sum over $\alpha_i$ is understood to include only those elements leading to non-zero traces in (C.5). This implies that there are only $\mathcal{O}(p)$ non-zero contributions to the sum — there are $p$ ways to choose $\alpha_1$ and this then determines the remainder up to an $\mathcal{O}(1)$ number of choices. For example, suppose that $m = 3$ and the first element is chosen to be $\mathfrak{b}_{\alpha_1} = |2)(5|$. Then for the trace in (C.6) to be non-zero we must take $\mathfrak{b}_{\alpha_2} = |5)(2|$ and one choice for the final two terms is $\mathfrak{b}_{\alpha_3} = \mathfrak{b}_{\alpha_4} = |2)(2|$. There are a small number of further options for the final two terms as the $X_{\mathrm{cl}}$ can also have non-zero entries on the band just above and below the diagonal. The fact that $\mathfrak{b}_{\alpha_1}$ and $\mathfrak{b}_{\alpha_2}$ are transposes of each other also ensures that $Y^a_{lm,\alpha_1} Y^b_{lm,\alpha_2} = |Y^a_{lm,\alpha_1}|^2 \geq 0$ so there can be no cancellations between different elements. Thus we conclude that

$$\overline{\langle \mathrm{Tr}\, Q^2 \rangle} \sim \nu^3 \Lambda^3 N^2 . \qquad (C.7)$$

It may seem strange that the $N$ scaling of the typical second Casimir is independent of $p$ (at least at large $p$). However the point is that conjugation by a typical element of $U(N') \otimes \mathbb{1}_p$ is enough to generate $\mathcal{O}(N)$ eigenvalues of magnitude $\mathcal{O}(N)$. For example, this can be checked explicitly even in the simple case of $N' = 2$. While the matrices $X^a_{\mathrm{cl}}$ are nearly diagonal, adjoint action by the matrix

$$\begin{bmatrix} 1/\sqrt{2} & 1/\sqrt{2} \\ 1/\sqrt{2} & -1/\sqrt{2} \end{bmatrix} \otimes \mathbb{1}_{N/2} \in U(2) \subset U(N), \qquad (C.8)$$

acting on any of the $X^a_{\mathrm{cl}}$ will create large off-diagonal blocks that lead to $\mathcal{O}(N)$ eigenvalues of magnitude $\mathcal{O}(N)$.

# D  Non-geometric perturbations of the saddle point

This Appendix considers the change of the entropy when the spherical cap configuration is transformed by $V = e^{i\epsilon H}$. The results of this Appendix are shown in Fig. 8 in the main text. We work numerically at finite $N$ as follows:

1. Consider the fluctuations $H_{lm} \equiv \hat{Y}_{lm} + \hat{Y}^\dagger_{lm}$. Here $\hat{Y}_{lm}$ is an $N \times N$ dimensional matrix

spherical harmonic, see footnote 12. For simplicity we take $p = 1$ here.

2. Set $V_{lm} \equiv e^{i\epsilon H_{lm}}$ for some small fixed $\epsilon$. We normalise $\text{Tr}\left(\hat{Y}_{lm}\hat{Y}_{lm}^{\dagger}\right) = 1$.

3. Use this $V_{lm}$ to construct the corresponding $Q_\Sigma$ from (4.4).

4. Compute the expectation value of the matrix $\langle Q_\Sigma^2 \rangle$, i.e. the matrix of expectation values of the components of $Q_\Sigma^2$, using (5.20) to compute the $\langle \delta\pi^2 \rangle$.

5. The arguments in §4.1 — in particular, repeating the argument around (4.5) without taking the trace — imply that at large $N$ and $\nu$ the eigenvalues of the matrix of expectation values $\langle Q_\Sigma^2 \rangle$ are just the $\ell_r^2$.

6. To leading order, i.e. up to multiplicative logarithms, the relative change in the entropy:

$$\frac{1}{\epsilon^2}\frac{\delta_H S}{S} = \frac{\sum_r \left(\ell_r^{(\epsilon)} - \ell_r^{(0)}\right)}{\epsilon^2 \sum_r \ell_r^{(0)}}. \tag{D.1}$$

A change in the entropy of $\mathcal{O}(\epsilon^2)$, as we find numerically and as assumed in (D.1), is consistent with the spherical cap being a saddle point. We have seen in §5.4 how in the strict large $N$ limit the change in some eigenvalues is instead $\mathcal{O}(|\epsilon|)$. This is due to the fact that most of the row lengths are zero on the spherical cap at infinite $N$, while at any finite $N$ they are finite. This is one sense in which the finite $N$ numerics we perform here are not in the same regime as the large $N$ considerations of the main text. Nonetheless, in Fig. 8 we see that the stability of the spherical cap saddle is robust against increasing $N$, supporting the stability of the large $N$ limit.

Fig. 8 shows the relative change in entropy under perturbations $H_{lm}$ for all allowed values for $lm$ with $m \geq 2$. To speed up the numerics we used an approximation for the matrix spherical harmonics described in [36] — this becomes more accurate at large $N$. The figure is consistent with the results being in the large $N$ regime, as they are stable against changing $N$. The most important result in Fig. 8 is that all changes in the entropy are positive, consistent with the cap subregion being a local minimum. We should note that we have found that numerical runs with certain choices of parameter values show negative changes for the entropy at large values of $m$. However, these negative modes are always found to go away as $N$ is increased and are therefore finite $N$ artefacts. The computation described above is only strictly valid in the large $N$ limit as otherwise there is a variance in the eigenvalues of $Q_\Sigma$ that is not being accounted for.

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
