# Peer review of "Minimal Areas from Entangled Matrices"

_SciPost Physics_

## Round 1 · Referee Report · Anonymous (Referee 1) · 2024-11-18

Report

Summary:
This paper is a continuation of previous efforts to define a Hilbert space factorization in matrix models that leads to an emergent area law entanglement entropy. The matrix model consists of NxN Hermitian matrices subject to a gauge symmetry given by conjugation by U(N) matrices. It is known that factorization in gauge theories requires a lifting of the Gauss law constraint, leading to an extended Hilbert space on which certain gauge symmetries are promoted to physical ``edge mode" symmetries. Physical states satisfying the Gauss law constraint are singlets under these edge mode symmetries, and their presence leads to a Gauss law entanglement entropy that counts the corresponding edge modes. These transforms under different representations of the symmetry, and the entropy involves dimensions of these representations.

For a generic matrix model, a preliminary (partially gauge fixed) notion of a subsystem can be given by a fixed MxM block, in which case the edge mode symmetry is U(M)xU(N-M). It was explained in previous work that for semi-classical states Fuzzy sphere state of the mini-BMN model, the U(N) symmetry has a low energy description in terms of area preserving diffeomorphisms of a two-sphere, and the MxM block can be viewed as a subregion of a two sphere. In the low energy description, the edge mode symmetry U(M)xU(N-M) are volume preserving diffeos that fix the subregion. The saddle point approximation to the entropy produces an area law on the sphere ( in this case area= perimeter) , which counts the dimensions of the dominant irrep.

In this work, the authors incorporate the edge mode symmetries U(N)/U(M)xU(N-m) that do not fix the MxM block: these ``wiggle" modes move the subregion on the sphere around in an volume preserving away. A gauge fixed subsystem then corresponds to any region of fixed volume. The authors show that reduced density matrix
involves direct sum over such subregions of fixed volume, and a saddle point approximation pics out the region with minmimal area. This is then compared to the area minimization of the Ryu-Takayanagi formula in holographic theories.

Recommendation:
The paper provides a compelling proposal for the factorization map and extended Hilbert space construction for matrix models, and illustrates in detail how this works in the mini-BMN model. I am happy to recommend its publication, provided the questions below are answered.

1.) Section 2.3 on the interpretation discussion of quantum reference frames is interesting but I would like to clarify how it goes beyond the usual paradigm of edge modes as ``Stuckelberg" degrees of freedom

For example consider the discussion around eq 2.16, that is supposed to make the matrix elements of X a gauge invariant quantity, despite that fact it manifestly transforms under conjugation by a U(N) matrix U. In the "Stuckelberg" edge mode paradigm, we would say that to make X gauge invariant, we need to introduce edge modes g in U(N), which is a choice of gauge promoted to a physical degree of freedom. Then we dress X by these edge modes by introducing gXg^{-1} as the new gauge invariant degree of freedom, where the gauge transformations map X to UXU^{-1} and g to gU^{-1}. Then gXg^{-1} is gauge invariant, but only after introducing g as a physical edge modes. Do the authors agree that what I said above is equivalent to the construction below 2.16?

If so, is the quantum reference frame the Stuckelberg formalism ?

2.) I am puzzeld by the definition of F in 2.38. According to the first two definitions, H and H' are both in G. What does it mean to take there direct product and then delete G?

Or did you mean by G\cap U(M) something different than the intersection of two sets? Like taking elements in G and restricting to their action on the first M indices?

Also Flag manifolds are quotients, did you mean HxH'/G instead of HxH'\G?
3.) Equation 2.39 is difficult to parse. Can an explicit definition of |\psi_{V}> be given?

Recommendation

Publish (easily meets expectations and criteria for this Journal; among top 50%)

  • validity: high
  • significance: high
  • originality: high
  • clarity: good
  • formatting: excellent
  • grammar: perfect

Author:  Ronak Soni  on 2025-04-22  [id 5396]

(in reply to Report 1 on 2024-11-18)

We thank the referee for their excellent summary of our work and their kind review and engagement. We have addressed their questions as follows.

  1. The referee is correct that there is a close connection between Stuckelberg modes and quantum reference frames. The Stuckelberg formalism is one way of dealing with edge modes, often convenient, but it is not necessary. If one just considers the degrees of freedom of the gauge theory restricted to the subregion, then one can find modes localised to the boundary within the bulk fields. This can be thought of as the "unitary" gauge of a Stuckelberg field living on the boundary, as explained in 2202.00133. But this is not all one can do. One can also set the Stuckelberg field to a fixed value (effectively getting rid of it). This is equivalent to gauge-fixing a boundary gauge freedom.

    In all approaches, one is writing down a prescription for factorising the Hilbert space. These different choices are different choices of factorisation map. This language of factorisation maps is a more useful one in that it gives all these choices at the boundary a definite place in our equations. Further, it also incorporates both gauge-theoretic edge modes as well as edge modes for scalar field theories into one framework. Another reason it is useful is that there is a clear notion of compatibility between a factorisation map and an algebra (mentioned in the first full paragraph on page 14) which we can apply to think about which factorisation prescription (any of the options in the previous paragraph) is appropriate for one's physical setup.

    Now we come to the question about the relation between our QRF discussion and edge modes: we use QRFs to delineate the set of operators whose entanglement we want to calculate. Thus, our discussion is meant to provide an operational meaning to our factorisation map. One doesn't need to take this detour in usual gauge theories, because in that case there are no gauge transformations that can literally move degrees of freedom across the boundary, and so one could (almost) always write down a non-relational algebra compatible with one's factorisation map.

    We are in a more complex situation where there are no non-relational observables local to a matrix sub-block, and thus this language is necessary to clarify what meaning our prescription has.

    Summarising, the QRF discussion is not equivalent to any particular formalism for edge modes; it only serves to clarify the meaning of the factorisation map.

  2. This question is due to a notational confusion, which we have now clarified by adding a sentence below equation 2.38 which explicitly defines the left quotient: "$\mathsf{F}$ is a left-quotient, which identifies $g \sim h h' g$ for $h \in H, h' \in H'$". In our notation, we are differentiating between right-quotients $G/H \times H'$ and left-quotients $H \times H'\backslash G$.

  3. We agree with the referee that an explicit definition of $|\psi_V\rangle$ would be useful, so we have added a new equation 2.40 that gives this definition, together with a couple of sentences immediately below it.

---

## Round 1 · Referee Report · Anonymous (Referee 2) · 2025-3-28

Report

In this article, the authors study a notion of entanglement entropy within the internal degrees of freedom in a large N gauged matrix model. Various proposals for such a notion of entanglement have been discussed previously in the literature, and this paper builds on some of this previous work to propose a Ryu-Takayanagi like minimization formula for the entanglement entropy. The essential subtlety in defining entanglement entropy in such a model is that the gauge-invariant Hilbert space of the matrix model does not admit any canonical factorization with respect to which the entropy can be computed, and thus the entropy is naturally associated to a choice of a factorization map, i.e., a map of the gauge-invariant Hilbert space into an extended Hilbert space which does admit factorization. The main result of this paper is a proposal for the factorization map that involves an integration over gauge transformations that "move the subregion" around while preserving its volume -- the resulting entropy in the saddle point approximation then gives an RT-like minimization formula for the entropy. In my opinion, the paper is extremely clear and very well-written. The results are very nice, and of broad potential interest to the AdS/CFT and quantum gravity community. I strongly recommend publication of the article.

One or two minor things that I personally found confusing: in AdS/CFT, there is a canonical UV definition of the entropy in terms of spatial subregions in the boundary CFT, which can then be computed in the semi-classical limit by the RT formula. In the matrix model, however, it seems like there is no canonical choice -- one must compute the entropy w.r.t to a choice of factorization map. Is there a sense in which the factorization map defined by the authors a canonical one? Or is it the case that this is a choice which gives a pleasing answer involving the minimization formula? Another general confusion is that in these matrix models, U(N) gauge transformations on the matrix side end up becoming area-preserving diffeomorphisms in the effective geometric description, and this was important in the interpretation of the entropy formula. How general is this fact? For instance, do we expect the U(N) gauge invariance of a general holographic field theory (say, N=4 SYM) to be related to diffeomorphism invariance in the dual gravity description? I feel some discussion on these issues could be useful, but as I said above, the paper is already very well-written and very readable, so this is more of a suggestion, not a requirement.

Recommendation

Publish (easily meets expectations and criteria for this Journal; among top 50%)

  • validity: -
  • significance: -
  • originality: -
  • clarity: -
  • formatting: -
  • grammar: -

Author:  Ronak Soni  on 2025-04-22  [id 5395]

(in reply to Report 2 on 2025-03-28)

We also thank the second referee for their excellent summary of our work and their kind review and engagement. The referee has recommended publication of the article in its current form. We agree with the two points they have raised, as we now explain.

The referee's observation about the lack of a canonical choice of UV factorisation is exactly correct. In fact, as we attempt to make clear in the introduction, we are effectively *hunting* for a reasonable factorisation map that gives us an area-minimization formula. Our main result is that it exists, and the reason we find it interesting is that it is completely explicit.

An additional comment: while it is true that AdS/CFT has a UV factorisation map, one might want to consider e.g. the entropy of an algebra of observables dressed to an observer. In that case also, the observables should not close to an algebra in the UV (e.g. large backreaction can cause the disintegration of the observer) and yet we expect to assign an entropy to the (almost-)algebra. Perhaps our type of approach will work there as well.

We also agree with the referee that it is a very important question to understand in conventional AdS/CFT whether there is any connection between the U(N) gauge symmetry of the boundary theory and bulk diffeomorphisms. We have a brief discussion on this point in the part of the discussion section entitled "beyond semiclassical matrices."

---

## Editorial Decision

resubmitted